# U-Cast: Learning Hierarchical Structures for High-Dimensional Time Series Forecasting

## Abstract

Time series forecasting (TSF) is a central problem in time series analysis. However, as the number of channels in time series datasets scales to the thousands or more, a scenario we define as **High-Dimensional Time Series Forecasting (HDTSF)**, it introduces significant new modeling challenges that are often not the primary focus of traditional TSF research. HDTSF is challenging because the channel correlation often forms complex and hierarchical patterns. Existing TSF models either ignore these interactions or fail to scale as dimensionality grows. To address this issue, we propose U-Cast, a channel-dependent forecasting architecture that learns latent hierarchical channel structures with an innovative query-based attention. To disentangle highly correlated channel representation, U-Cast adds a full-rank regularization during training. We also release Time-HD, the first benchmark of large, diverse, high-dimensional datasets. Our theory shows that exploiting cross-channel information lowers forecasting risk, and experiments on Time-HD demonstrate that U-Cast surpasses strong baselines in both accuracy and efficiency. Together, U-Cast and Time-HD provide a solid basis for future HDTSF research. Our code and benchmark are available to ensure reproducibility.

## 1 Introduction

Time series data involving multiple variables (a.k.a. channels) that change over time are fundamental to numerous real-world applications. Consequently, time series forecasting (TSF) has become a central challenge in the time series analysis community, aiming to predict future values based on historical observations (Chatfield, 2000; De Gooijer & Hyndman, 2006). Recent advances in machine learning have catalyzed significant methodological progress in this field (Wang et al., 2024b; Lim & Zohren, 2021), leading to a variety of novel architectures (Wu et al., 2021; Zhou et al., 2022; 2021; Liu et al., 2024a; Oreshkin et al., 2020; Nie et al., 2023; Wang et al., 2024a; Wu et al., 2022; Ni et al., 2025).

Table 1: Comparison of existing datasets by channel scale.

| Datasets | #Channels |
|---|---|
| ETT | 7 |
| Weather | 21 |
| Solar | 137 |
| ECL | 321 |
| Traffic | 862 |
| **Time-HD** | **1k–20k** |

Despite this progress, most existing TSF datasets, as shown in Table 1 and Table 6, remain relatively low-dimensional [1], with only a handful of channels (e.g., ETT, Weather) to at most a few hundred (e.g., Traffic, ECL). However, in many real-world applications, **the number of channels can easily scale into the thousands or more**. For instance, financial markets involve thousands of stocks (Granger & Newbold, 2014), urban traffic systems rely on tens of thousands of sensors (Yin et al., 2021), smart grids generate massive streams from large-scale meters (Wan et al., 2015), and climate reanalysis contains hundreds of thousands of variables defined on global spatial grids (Wu et al., 2023). We define this scenario as **High-Dimensional Time Series Forecasting (HDTSF)**, where the scale of channels introduces significant new modeling challenges that are often not the primary focus of traditional TSF research.

Two difficulties are especially pressing. First, scaling to thousands of channels with complicated dependencies makes many channel-dependent models, such as iTransformer (Liu et al., 2024a) and

---

[1]We use the term "dimensionality" to denote the number of variables or channels, either from a single object or aggregated across multiple related objects such as sensors (see Appendix D for a detailed definition).

TSMixer (Chen et al., 2023), computationally inefficient or unreliable. Second, high dimensionality often gives rise to **latent hierarchical structures** among the channels. These structures are common in large real-world systems (Appendix P), where channels exhibit implicit groupings and multi-level correlations based on factors such as spatial proximity (e.g., nested geographical regions in climate data) or semantic relationships (e.g., stocks within related economic sub-sectors). While such multi-scale correlations are pervasive, most existing models are not designed to explicitly discover or leverage them, which limits generalizability in high-dimensional contexts. This motivates two central research questions: (1) How can we build *forecasting architectures that explicitly capture latent hierarchical structures* to scale across thousands of channels? (2) How can we systematically *evaluate such* models when existing benchmarks rarely exceed a few hundred channels?

To address the first question, we propose **U-CAST**, a scalable forecasting architecture that introduces a Hierarchical Latent Query Network and a Hierarchical Upsampling Network to efficiently uncover latent multi-scale channel organizations. To disentangle highly correlated inputs, U-CAST also employs a full-rank regularization objective that encourages diverse, non-redundant channel representations. Answering the second question requires a *new empirical foundation*. Existing TSF benchmarks contain only limited channels and may not possess the scale or structural richness representative of high-dimensional applications, making them inadequate for evaluating scalability and advanced dependency modeling. To this end, we curate **TIME-HD**, the first comprehensive benchmark suite for HDTSF. TIME-HD spans 16 datasets with 1k–20k channels across diverse domains, providing a much-needed testbed for evaluating scalability and dependency modeling. While this paper does not claim to exhaustively analyze all benchmarking results, we release TIME-HD as a community resource to facilitate systematic study of HDTSF in future work. Our experiments on TIME-HD validate U-CAST's efficacy in learning hierarchical structures and achieving superior forecasting accuracy in these demanding high-dimensional settings. Our main contributions are summarized as:

**C1: Theoretical and empirical analysis of high-dimensional channel dependency.** We prove that channel-dependent (CD) models have lower Bayes risk than channel-independent (CI) models whenever non-redundant channels exist, with the advantage increasing in higher dimensions (Section 3). Our controlled experiments on synthetic and real-world datasets further corroborate these findings.

**C2: U-CAST: A scalable architecture for HDTSF.** We propose U-CAST (Section 4), a new CD forecasting model that introduces an innovative query-based attention mechanism to efficiently learn latent hierarchical channel structures and enables scalable modeling of inter-channel correlations. U-CAST combines strong performance across TIME-HD with the best efficiency among baselines, making it a strong reference design for future HDTSF models.

**C3: TIME-HD: the first benchmark suite for HDTSF.** We release TIME-HD (Section 5), the first comprehensive benchmark suite curated for HDTSF. TIME-HD spans 16 datasets (1k–20k channels) across diverse domains, providing the necessary foundation for evaluating scalability and dependency modeling in realistic high-dimensional contexts. While not intended as an exhaustive benchmarking study, TIME-HD establishes a standardized and challenging testbed that enables rigorous evaluation and comparison of future models.

**C4: Open-source library for reproducibility.** We provide TIME-HD-LIB, an open-source library supporting the benchmark. It offers standardized preprocessing, unified evaluation protocols, and automated hyperparameter optimization. This toolkit ensures reproducibility of our results and lowers the barrier for future work on HDTSF.

## 2 RELATED WORK

**High-Dimensional Time Series Datasets.** Some high-dimensional datasets have already been used in recent work on building time series foundation models (Woo et al., 2024; Ansari et al., 2024; Shi et al., 2024b; Goswami et al., 2024). These studies primarily focus on leveraging high-dimensional datasets for pretraining. However, several limitations exist in how they handle such datasets. **First**, during pretraining, channels are treated independently, capturing only temporal dependencies without modeling inter-channel correlations. **Second**, despite pretraining on high-dimensional datasets, fine-tuning and evaluation are still performed on low-dimensional ones. **Third**, some pretraining datasets have unaligned timestamps, complicating their use in evaluation. These limitations highlight the importance of developing high-dimensional time series datasets specifically tailored for benchmarking

and evaluation in time series forecasting research. Table 6 in Appendix A provides a more detailed summarization of the differences between TIME-HD and existing works.

**Multivariate Time Series Forecasting.** Previous methods for modeling multivariate time series can be broadly categorized into **channel-independent (CI)** and **channel-dependent (CD)** strategies. The CI modeling strategy employs a shared model backbone across all channels and processes each channel independently during the forward pass (Nie et al., 2023; Zeng et al., 2023; Xu et al., 2023; Wang et al., 2024a; Tan et al., 2024; Dai et al., 2024; Lin et al., 2024; Li et al., 2023). This design typically results in lower model capacity but offers greater robustness. In contrast, the CD modeling strategy introduces dedicated modules to capture inter-channel dependencies. Based on the granularity at which these correlations are modeled, CD can be further divided into position-wise(Chen et al., 2023; Wu et al., 2022; 2021; Zhou et al., 2021; 2022; Zhang & Yan, 2023), token-wise(Liu et al., 2024a; Wang et al., 2024c), and cluster-wise (Chen et al., 2024; Qiu et al., 2024b; Li & Anastasiu, 2025) approaches. These methods are expected to provide higher capacity and make better use of cross-channel information. However, due to the constraints of existing low-dimensional time series forecasting benchmarks, most methods have not been validated at scale.

## 3 PRELIMINARY STUDY

Prior studies on existing time series benchmarks, which are often of low dimensionality, do not consistently show a clear advantage of CD over CI models (Zhao & Shen, 2024; Nie et al., 2023; Zeng et al., 2023; Dai et al., 2024). This observation might lead to skepticism regarding the practical benefits of explicitly modeling inter-channel correlations. However, we hypothesize that the limited dimensionality of these benchmarks inherently restricts the potential gains from CD approaches. This preliminary study aims to theoretically and empirically investigate the impact of data dimensionality on the forecasting performance achievable by leveraging channel correlations. We begin with a theoretical analysis of risk reduction.

**Task Formulation.** For time series forecasting, given an input time series $\mathbf{X} \in \mathbb{R}^{C \times T}$, where $T$ represents the length of the look-back window and $C$ represents the number of variables, the goal is to predict the future $S$ time steps $\mathbf{Y} \in \mathbb{R}^{C \times S}$.

### 3.1 THEORETICAL ANALYSIS OF RISK REDUCTION FROM CD MODELS

To formally analyze the benefits of CD, we consider a simplified setting. Assume a **bivariate** time series ($C = 2$) generated by a Vector Autoregression of order 1 (VAR(1)) (Stock & Watson, 2001):

$$z_{t+1} = \mathbf{A}z_t + \varepsilon_{t+1}, \quad \mathbf{A} = \begin{pmatrix} a_{11} & a_{12} \\ a_{21} & a_{22} \end{pmatrix}, \quad \varepsilon_t \overset{i.i.d.}{\sim} \mathcal{N}(\mathbf{0}, \Sigma), \quad \Sigma = \begin{pmatrix} \sigma_{11} & 0 \\ 0 & \sigma_{22} \end{pmatrix}. \quad (1)$$

Here, $z_t = (z_t^{(1)}, z_t^{(2)})^\top$ is the bivariate time series value at time $t$, with $z_0 \sim \mathcal{N}(\mathbf{0}, \mathbf{I})$, where $\mathbf{I}$ is the identity matrix. The matrix $\mathbf{A}$ captures the linear dependencies, with coefficients $a_{ij}$ describing self-dependencies ($i = j$) and cross-dependencies ($i \neq j$). The noise $\varepsilon_t$ is zero-mean Gaussian with independent components. Our goal is to predict $z_{t+1}$ given $z_t$. All time series are assumed to be normalized to have zero mean. We compare two modeling strategies under squared-error loss: *(1) CI Modeling:* Each channel $z_{t+1}^{(i)}$ is predicted using only its own past, $z_t^{(i)}$. The Bayes optimal forecast for $z_{t+1}^{(i)}$ is $\mathbb{E}[z_{t+1}^{(i)}|z_t^{(i)}]$ and the total risk is $R_{\text{CI}} = \sum_{i=1}^{2} \mathbb{E}[(z_{t+1}^{(i)} - \mathbb{E}(z_{t+1}^{(i)}|z_t^{(i)})^2]$. *(2) CD Modeling:* Each channel $z_{t+1}^{(i)}$ is predicted using the past of all channels. The Bayes optimal forecast for $z_{t+1}^{(i)}$ is $\mathbb{E}[z_{t+1}^{(i)}|z_t]$ and the total risk is $R_{\text{CD}} = \sum_{i=1}^{2} \mathbb{E}[(z_{t+1}^{(i)} - E(z_{t+1}^{(i)}|z_t)^2]$.

**Theorem 1** (Risk Reduction from CD). *For the time series described in Eq. equation 1, the Bayes risks of CI and CD models under squared-error loss satisfy:*

$$R_{\text{CI}} - R_{\text{CD}} = a_{12}^2 \operatorname{Var}\left(z_t^{(2)} \mid z_t^{(1)}\right) + a_{21}^2 \operatorname{Var}\left(z_t^{(1)} \mid z_t^{(2)}\right) \geq 0$$

*This inequality is strict if and only if $a_{12} \neq 0$ and $a_{21} \neq 0$, and the conditional variances $\operatorname{Var}(z_t^{(1)} \mid z_t^{(2)})$ (representing the variance of $z_t^{(1)}$ that remains after $z_t^{(2)}$ is known) and $\operatorname{Var}(z_t^{(2)} \mid z_t^{(1)})$ are positive.*

We provide the proof of this theorem in Appendix J. Theorem 1 establishes that CD modeling is at least as good as CI modeling. A strict advantage for CD arises if: (1) there is genuine predictive information flowing between channels (non-zero $a_{12}$ or $a_{21}$ means one channel directly influences the future of the other), and (2) each channel contains some unique information not present in the other (positive conditional variances). If channels are truly independent in their evolution ($a_{12} = a_{21} = 0$) or one is a deterministic function of the other, then $R_{\text{CI}} = R_{\text{CD}}$.

To investigate the impact of increasing dimensionality, we now consider a general $P$-channel VAR(1) process: $z_{t+1} = \mathbf{A}z_t + \varepsilon_{t+1}$, where $z_t \in \mathbb{R}^P$. We focus on forecasting a single target channel, say $Y = z_{t+1}^{(1)}$, which is given by $z_{t+1}^{(1)} = \sum_{j=1}^{P} a_{1j} z_t^{(j)} + \varepsilon_{t+1}^{(1)}$, with $\text{Var}(\varepsilon_{t+1}^{(1)}) = \sigma_{11}$. Let $R_p$ denote the Bayes risk (minimum MSE) for forecasting $Y = z_{t+1}^{(1)}$ using the information from the first $p$ channels at time $t$, i.e., $\{z_t^{(1)}, \ldots, z_t^{(p)}\}$.

**Theorem 2** (Bayes Risk Monotonicity with Increasing Channel Information). *For forecasting the target channel $z_{t+1}^{(1)}$, the Bayes risks obtained by conditioning on an increasing number of predictor channels $\{z_t^{(1)}, \ldots, z_t^{(p)}\}$ exhibit monotonicity: $R_1 \geq R_2 \geq \cdots \geq R_P = \sigma_{11}$. Note that $R_1$ is the risk of CI Modeling (using only $z_t^{(1)}$ to predict $z_{t+1}^{(1)}$), and $R_p(p > 1)$ is the risk of CD Modeling. Furthermore, the risk difference between CI and CD, $\Delta_p = R_1 - R_p$, is non-decreasing in $p$:*

$$0 = \Delta_1 \leq \Delta_2 \leq \cdots \leq \Delta_P = R_1 - \sigma_{11},$$

*This gap strictly increases (i.e., $\Delta_p > \Delta_{p-1}$) if and only if the $p$-th channel, $z_t^{(p)}$, provides new information for predicting $Y$ that is not already contained in channels $z_t^{(1)}, \ldots, z_t^{(p-1)}$.*

The proof is provided in Appendix K. Theorem 2 formally shows that incorporating additional informative and non-redundant channels progressively reduces the forecasting risk for a target variable. This underscores the theoretical benefit of using more relevant dimensions (channels), supporting our hypothesis that higher-dimensional data provides greater opportunity for CD models to outperform CI approaches.

## 3.2 EMPIRICAL ANALYSIS

Table 2: MSE for CI and CD models under different settings.

| Model | Independent 100 | Anti-Self 100 | Anti-Self 250 | Anti-Self 2000 |
|---|---|---|---|---|
| CI | 0.0043 | 0.0052 | 0.0054 | 0.0054 |
| CD | 0.0066 | 0.0014 | 0.0012 | 0.0011 |

To empirically validate our theoretical insights above, we conduct controlled experiments on synthetic data for evaluating the performance difference between CI and CD modeling under varying channel-dependency structures and dimensionalities.

**Data Generation.** We generate synthetic multivariate time series using a $C$-dimensional Vector Autoregression process in Equation 1, where the coefficient matrix $\mathbf{A} \in \mathbb{R}^{C \times C}$ encodes the dependency structure of channels. We focus on two distinct structures for $\mathbf{A}$: (1) **Independent**: Channels evolve independently of each other i.e., $\mathbf{A}$ is a diagonal matrix. For this setting, we use $C = 100$ channels. (2) **Anti-Self**: Channels depend significantly on other channels but not on their own immediate past, i.e., $\mathbf{A}$ has zero diagonal entries and non-zero off-diagonal entries. To examine the impact of dimensionality, we generate Anti-Self datasets with $C \in \{100, 250, 2000\}$. More experimental details are provided in Appendix N.

**Forecasting Models.** To isolate the core impact of utilizing versus disregarding channel dependencies, we employ two simple linear forecasting architectures: (1) **CI**: A single univariate linear model is fitted for each channel using only its own past values. (2) **CD**: A linear model is applied along the channel dimension to incorporate inter-channel dependencies.

**Results and Discussion.** Table 2 shows that CI models perform better in the Independent setting without inter-channel dependencies, while CD models achieve markedly lower MSE in Anti-Self settings, with larger gains as the number of interacting channels grows. These results align with our theory that *CD modeling becomes increasingly effective when significant inter-channel dependencies exist,*

Figure 1: Overall framework of U-CAST, which consists of five main modules: channel embedding, hierarchical latent query, temporal alignment, hierarchical upsampling, and output projection.

*especially in higher-dimensional systems*. We further confirm this on real-world data (Appendix S). These findings underscore two critical needs: **(1) novel CD models** that efficiently and effectively navigate the intricacies of high-dimensional time series, and **(2) dedicated high-dimensional forecasting benchmarks** with complex channel dependencies for rigorous evaluation. Accordingly, in the subsequent sections, we address these needs by first introducing our U-CAST framework specifically tailored for HDTSF and then presenting TIME-HD, a comprehensive benchmark designed to support and galvanize further research in this evolving domain.

## 4 U-CAST FRAMEWORK

Our theoretical and empirical analyses in Section 3 highlight the need for novel CD models that can efficiently handle high-dimensional time series. Traditional models often struggle with scalability and overlook the structured nature of such data, especially the *hierarchical* correlations among variable groups that frequently occur in real-world datasets (see Appendix P). To address this, we propose U-CAST (see Figure 1), an efficient model that captures channel correlations via learning latent hierarchical structures. We also introduce a full-rank regularization term to encourage disentanglement and improve the learning of structured representations.

### 4.1 MODEL ARCHITECTURE

**Channel Embedding.** The input sequence is first normalized. A linear projection $\mathbf{W}_{\text{in}} \in \mathbb{R}^{T \times d}$ converts the temporal dimension to a hidden dimension $d$:

$$\mathbf{H}^{(0)} = \mathbf{X}\mathbf{W}_{\text{in}} \in \mathbb{R}^{C \times d}. \tag{2}$$

The channel embedding aggregates the global representations of series that can be more variate-centric. This is critical as it allows the subsequent attention layers to focus exclusively on learning the complex inter-channel correlations, which is the central challenge of HDTSF.

**Hierarchical Latent Query Network.** Attention is order-invariant and well suited to modelling channel-wise dependencies because every variable can attend to every other variable without assuming locality and attention is order-invariant. However, applying full attention to $C$ channels incurs quadratic cost, which is prohibitive when $C$ is large. Hierarchical latent query network addresses this by introducing a set of *latent queries* that serve as an information bottleneck. At layer $\ell \in \{1, \ldots, L\}$ the number of latent queries is $C_\ell = \lfloor C/r^\ell \rfloor$, where $r > 1$ is a pre-defined *reduction ratio*. The queries, which are learnable, are shared across samples: $\mathbf{Q}_\ell \in \mathbb{R}^{C_\ell \times d}$. The Latent Query Attention at layer $\ell$ is

$$\mathbf{Q} = W_q\mathbf{Q}_\ell, \quad \mathbf{K} = W_k\mathbf{H}^{(\ell-1)}, \quad \mathbf{V} = W_v\mathbf{H}^{(\ell-1)},$$

$$\mathbf{H}^{(\ell)} = W_o\Big(\text{softmax}\big(\tfrac{\mathbf{Q}\mathbf{K}^\top}{\sqrt{d_h}}\big)\mathbf{V}\Big) \in \mathbb{R}^{C_\ell \times d}, \tag{3}$$

followed by layer normalisation. Successive layers hierarchical latent query network builds a hierarchy in which higher-level queries summarise wider channel groups.

**Temporal Alignment.** The deepest latent representation $\mathbf{H}^{(L)} \in \mathbb{R}^{C_L \times d}$ stores $C_L$ latent channel tokens whose $d$-dimensional feature vectors encode the *temporal* dynamics extracted from the look-back window. To keep these temporal features coherent when the model switches from down-sampling to up-sampling, we apply a shared linear predictor $f_{\text{pred}}$ along the temporal dimension:

$$\mathbf{U}^{(L)} = f_{\text{pred}}\big(\mathbf{H}^{(L)}\big) \in \mathbb{R}^{C_L \times d}. \tag{4}$$

**Hierarchical Upsampling Network.** Forecasting demands channel-wise outputs of size $C$, so the hierarchy is traversed in reverse to restore resolution. At layer $\ell = L, \dots, 1$ the Up-Latent Query Attention uses the representation from the encoder as queries and the current decoder representation as keys and values:

$$\mathbf{Q} = \mathbf{W}_q \mathbf{H}^{(\ell-1)}, \quad \mathbf{K} = \mathbf{W}_k \mathbf{U}^{(\ell)}, \quad \mathbf{V} = \mathbf{W}_v \mathbf{U}^{(\ell)},$$
$$\mathbf{U}^{(\ell-1)} = \mathbf{W}_o\Big(\text{softmax}\big(\tfrac{\mathbf{Q}\mathbf{K}^\top}{\sqrt{d_h}}\big)\mathbf{V}\Big) + \mathbf{H}^{(\ell-1)} \in \mathbb{R}^{C_{\ell-1} \times d}. \tag{5}$$

The skip connection $\mathbf{H}^{(\ell-1)}$ guides the reconstruction, ensuring that the original channel information is recovered with minimal distortion.

**Output Projection.** After upsampling $\mathbf{U}^{(0)} \in \mathbb{R}^{C \times d}$ has the same channel dimension as the encoder output $\mathbf{H}^{(0)}$. A residual link followed by a projection yields the horizon-length prediction:

$$\hat{\mathbf{Y}} = (\mathbf{U}^{(0)} + \mathbf{H}^{(0)})\mathbf{W}_{\text{out}}, \qquad \mathbf{W}_{\text{out}} \in \mathbb{R}^{d \times S}. \tag{6}$$

Finally, we apply the inverse of the initial normalisation to obtain the forecast in the original scale.

## 4.2 Optimization Objective

**Full-Rank Regularization.** Representation learning is fundamentally about *disentanglement*: channels are highly correlated in the high-dimensional time series data (see Table 3). This entanglement means that the latent matrix $\mathbf{H}^{(\ell-1)} \in \mathbb{R}^{C \times d}$ often has rank $r = \text{rank}(\mathbf{H}^{(\ell-1)}) \ll C$. The resulting rank deficiency signals redundancy among channels and obscures the latent hierarchical channel structure. Achieving disentanglement is therefore required to learn this structure. To resolve this issue, we introduce a full-rank regularization, whose effect is formalised in Theorem 3 below.

**Theorem 3** (Full-Rank Regularisation). *Let $\mathbf{H}^{(\ell-1)} \in \mathbb{R}^{C \times d}$ be of rank $r < \min(C, d)$. Let $\mathbf{Q} \in \mathbb{R}^{C' \times C}$ be learnable, full row rank, and $C' \leq r$. Define $\mathbf{H}^{(\ell)} = \mathbf{Q}\mathbf{H}^{(\ell-1)} \in \mathbb{R}^{C' \times d}$. There exists a choice of $\mathbf{Q}$ such that $\text{rank}(\mathbf{H}^{(\ell)}) = C'$. Further, if $r \geq d$ and $C' \geq d$, adding a full-rank regulariser (e.g. the log-determinant of $\mathbf{H}^{(\ell)}\mathbf{H}^{(\ell)\top}$) drives $\mathbf{H}^{(\ell)}$ toward row rank $\min(C', d)$.*

The proof is given in Appendix L. Theorem 3 states that enforcing full rank on $\mathbf{H}^{(\ell)}$ is sufficient to remove linear redundancy among channels and reveal a clear hierarchical latent channel structure. Define the row-covariance matrix $\mathbf{\Sigma}^{(\ell)} = \frac{1}{d}\mathbf{H}^{(\ell)}\mathbf{H}^{(\ell)\top} \in \mathbb{R}^{C' \times C'}$. The log-determinant $\log \det\big(\mathbf{\Sigma}^{(\ell)} + \varepsilon I_{C'}\big)$ is proportional to the generalised variance[2]. Maximising this value keeps every eigenvalue bounded away from zero, so the channel vectors occupy a larger subspace and share less redundant information. We thus define the full-rank regularization loss as

$$\mathcal{L}_{\text{cov}}^{(\ell)} = -\frac{1}{C'} \log \det\big(\mathbf{\Sigma}^{(\ell)} + \varepsilon I_{C'}\big),$$

where $\varepsilon I_{C'}$ ensures positive definiteness and stabilizes early training. The $1/C'$ factor removes scale dependence, simplifying the weighting of this term. The overall loss is the average over all $L$ layers. Minimizing $\mathcal{L}_{\text{cov}}$ increases the Shannon differential entropy (Theorem 4), reducing redundancy and promoting disentanglement by encouraging each channel to carry distinct information.

**Overall Objective.** The model parameters are learned by minimising the combined loss

$$\mathcal{L} = \mathcal{L}_{\text{mse}} + \alpha\, \mathcal{L}_{\text{cov}}, \tag{7}$$

where $\mathcal{L}_{\text{mse}}$ is the supervised mean-squared forecasting error, $\mathcal{L}_{\text{cov}}$ is the layer-averaged covariance penalty, and $\alpha$ controls the strength of this regulariser.

Table 3: TIME-HD descriptions. Details on dataset generation, collection, cleaning, and prediction length determination are provided in Appendix G.

| Dataset | Dimensions | Dataset Size | Frequency | Pred Length | Storage | Domains | Correlation |
|---------|-----------|-------------|-----------|-------------|---------|---------|-------------|
| Neurolib | 2,000 | 60,000 | 1 ms | 336 | 2.44 GB | Neural Science | 0.926±0.007 |
| Solar | 5,162 | 105,120 | 5 Mins | 336 | 2.33 GB | Energy | 0.998±0.000 |
| Atec | 1,569 | 8,928 | 10 Mins | 336 | 158.8 MB | Cloud | 0.851±0.036 |
| Meter | 2,898 | 28,512 | 30 Mins | 336 | 651 MB | Energy | 0.864±0.023 |
| Temp | 3,850 | 17,544 | 1 Hour | 168 | 383.6 MB | Weather | 0.833±0.039 |
| Wind | 3,850 | 17,544 | 1 Hour | 168 | 331.4 MB | Weather | 0.937±0.011 |
| Traffic-CA | 7,491 | 43,824 | 1 Hour | 168 | 2.48 GB | Traffic | 0.962±0.009 |
| Traffic-GLA | 3,376 | 43,824 | 1 Hour | 168 | 1.12 GB | Traffic | 0.947±0.012 |
| Traffic-GBA | 2,229 | 43,824 | 1 Hour | 168 | 689.1 MB | Traffic | 0.978±0.003 |
| Air Quality | 1,105 | 15,461 | 6 Hours | 28 | 94.6 MB | Environment | 0.854±0.030 |
| SIRS | 2,994 | 9,000 | 1 Day | 7 | 675 MB | Epidemiology | 0.991±0.002 |
| SP500 | 1,475 | 7,553 | 1 Day | 7 | 184.1 MB | Finance | 0.738±0.046 |
| M5 | 3,049 | 1,941 | 1 Day | 7 | 13.7 MB | Sale | 0.726±0.073 |
| Measles | 1,161 | 1,330 | 1 Day | 7 | 12.1 MB | Epidemiology | 0.724±0.048 |
| Wiki-20k | 20,000 | 2,557 | 1 Day | 7 | 289 MB | Web | 0.923±0.017 |
| Mobility | 5,826 | 974 | 1 Day | 7 | 29.7 MB | Social | 0.847±0.027 |

## 5 TIME-HD: HIGH-DIMENSIONAL TIME SERIES FORECASTING BENCHMARK

The goal of TIME-HD is to support and advance research in TSF, which is a rapidly growing and increasingly important area. As shown in Table 3, TIME-HD has several key characteristics:

- **High Dimensionality.** TIME-HD includes 16 high-dimensional time series forecasting datasets. The number of variables (dimensions) in these datasets ranges from 1,161 to 20,000, which is significantly larger than in commonly used benchmarks such as ETT, Weather, ECL, Solar, and Traffic (which typically have 7–862 channels). These high-dimensional datasets provide opportunities for developing scalable methods, enable exploration of channel-dependent models, and allow for more robust performance evaluations.
- **Diverse Sources.** TIME-HD comprises both simulated and real-world datasets. *Neurolib* and *SIRS* are simulated using domain-informed differential equations and are valuable for scientific modeling and hypothesis testing. The remaining datasets are based on real-world observations, making them suitable for evaluating the generalization ability of forecasting models in practical settings. The inclusion of both types enables comprehensive performance assessments.
- **Varied Scales.** TIME-HD offers datasets of different sizes, defined by the number of variables and temporal length. Disk memory usage reflects this variability. As indicated in Table 3, there are 4 large-scale (gigabyte-level), 8 medium-scale (hundreds of megabytes), and 4 small-scale (tens of megabytes) datasets. The small and medium-scale datasets fit in the memory of a single GPU, making them suitable for evaluating resource-intensive models. The large-scale datasets support research on scalable methods using mini-batching and distributed training.
- **Different Sampling Frequencies.** TIME-HD covers a range of sampling frequencies, including milliseconds, minutes, hours, and days (see Table 3). This variety reflects real-world conditions across application domains and enables evaluation across different temporal resolutions. Furthermore, TIME-HD uses frequency-specific prediction lengths that are more realistic than fixed-length horizons commonly used in previous benchmarks.
- **Broad Domain Coverage.** TIME-HD includes datasets from 10 diverse domains, such as neural science, energy, cloud computing, weather, traffic, epidemiology, finance, and social behavior. This diversity supports the development of general-purpose forecasting models and enables comparison against domain-specific approaches.

*Channel Correlation Statistic.* We compute channel correlation following TFB (Qiu et al., 2024a), with results reported in the *correlation* column of Table 3. All datasets exhibit strong channel dependencies, which are not intentionally curated but naturally arise in large-scale real-world systems.

---

[2]Product of the eigenvalues of $\Sigma^{(\ell)}$.

Table 4: Forecasting results on various datasets. Input length $T$ is tuned per model for best performance (see Appendix I), and prediction length $S$ is given in Table 3. Lower MSE/MAE indicates better performance; best results are in **red** and second-best are underlined.

| Models | Channel-Independent | | | | | | Channel-Dependent | | | | | | | | | | | |
|---|---|---|---|---|---|---|---|---|---|---|---|---|---|---|---|---|---|---|
| | DLinear | | PAttn | | PatchTST | | iTransformer | | TSMixer | | TimesNet | | CCM | | DUET | | U-Cast | |
| Metrics | MSE | MAE | MSE | MAE | MSE | MAE | MSE | MAE | MSE | MAE | MSE | MAE | MSE | MAE | MSE | MAE | MSE | MAE |
| Atec | 0.318 | 0.314 | 0.299 | **0.275** | 0.298 | 0.298 | 0.345 | 0.319 | 0.398 | 0.387 | 0.493 | 0.429 | 0.362 | 0.346 | 0.330 | 0.339 | **0.287** | 0.280 |
| Air Quality | 0.449 | 0.446 | 0.449 | 0.432 | 0.448 | 0.432 | 0.447 | 0.431 | 0.447 | 0.438 | 0.457 | 0.438 | 0.458 | 0.451 | 0.452 | 0.444 | **0.446** | **0.430** |
| Temp | 0.272 | 0.391 | 0.278 | 0.395 | 0.279 | 0.396 | 0.265 | 0.386 | 0.266 | 0.389 | 0.287 | 0.408 | 0.279 | 0.398 | 0.435 | 0.511 | **0.262** | **0.383** |
| Wind | 1.128 | 0.697 | 1.256 | 0.758 | 1.254 | 0.757 | 1.116 | 0.699 | 1.346 | 0.742 | 1.161 | 0.708 | 1.165 | 0.698 | 1.227 | 0.746 | **1.104** | **0.692** |
| Mobility | 0.344 | 0.359 | 0.337 | 0.336 | 0.344 | 0.341 | **0.312** | **0.314** | 1.165 | 0.787 | 0.410 | 0.388 | 0.523 | 0.468 | 0.439 | 0.410 | 0.315 | 0.317 |
| Traffic-CA | 0.063 | 0.141 | 0.491 | 0.554 | 0.295 | 0.417 | 0.271 | 0.391 | 0.082 | 0.186 | 0.101 | 0.205 | 0.067 | 0.153 | — | — | **0.061** | **0.131** |
| Traffic-GBA | 0.062 | 0.137 | 0.059 | 0.132 | 0.060 | 0.139 | 0.063 | 0.141 | 0.074 | 0.168 | 0.098 | 0.204 | 0.067 | 0.151 | 0.650 | 0.638 | **0.059** | **0.126** |
| Traffic-GLA | 0.062 | 0.142 | 0.060 | 0.136 | 0.060 | 0.136 | 0.065 | 0.145 | 0.071 | 0.168 | 0.094 | 0.199 | 0.066 | 0.152 | 0.810 | 0.738 | **0.060** | **0.132** |
| M5 | 3.688 | 0.870 | 3.650 | 0.867 | 3.655 | 0.872 | 3.549 | 0.853 | 6.863 | 1.623 | 4.490 | 0.919 | 4.916 | 0.941 | 3.768 | 0.880 | **3.501** | **0.849** |
| Measles | 0.128 | 0.252 | 0.011 | 0.048 | 0.013 | 0.058 | 0.010 | 0.048 | 0.569 | 0.547 | 0.018 | 0.060 | 0.244 | 0.323 | 0.015 | 0.064 | **0.010** | **0.042** |
| Neurolib | 1.793 | 0.381 | 2.458 | 0.445 | 2.395 | 0.438 | **1.718** | **0.347** | 2.240 | 0.532 | 2.475 | 0.458 | 1.774 | 0.403 | 2.519 | 0.451 | 1.750 | 0.350 |
| Solar | 0.174 | 0.255 | 0.604 | 0.582 | 0.416 | 0.469 | 0.343 | 0.427 | **0.155** | **0.216** | 0.157 | 0.224 | 0.177 | 0.258 | — | — | 0.172 | 0.246 |
| SIRS | 0.058 | 0.168 | 0.025 | 0.109 | 0.033 | 0.129 | 0.028 | 0.113 | 0.016 | 0.078 | 0.162 | 0.327 | 0.048 | 0.156 | 0.095 | 0.236 | **0.007** | **0.052** |
| Meters | 0.944 | **0.549** | 0.941 | 0.552 | 1.254 | 0.706 | 0.949 | 0.556 | 1.034 | 0.586 | 0.946 | 0.551 | 1.308 | 0.731 | | | 0.943 | 0.551 |
| SP500 | 0.630 | 0.367 | 0.516 | 0.309 | 0.523 | 0.313 | 0.511 | 0.306 | 2.674 | 1.120 | 0.611 | 0.343 | 0.727 | 0.414 | 0.568 | 0.335 | **0.502** | **0.301** |
| Wiki-20k | 10.740 | 0.394 | 10.290 | 0.306 | 10.291 | 0.305 | 10.933 | 0.405 | 10.446 | 0.332 | 10.586 | 0.325 | 11.413 | 0.373 | 10.278 | 0.304 | **10.273** | **0.302** |
| 1st Count | 0 | 1 | 1 | 1 | 0 | 0 | 2 | 2 | 1 | 1 | 0 | 0 | 0 | 0 | 0 | 0 | **12** | **11** |

The high correlation reflects complex inter-channel dependencies and a shared global trend across channels. Further analysis on channel correlation of TIME-HD is provided in Appendix H.

## 6 EXPERIMENT RESULTS AND FRAMEWORK ANALYSIS

**Baselines.** We consider the following representative models as our baselines for HDTSF:

(a) **Channel-Independent.** We include three methods that employ a shared model backbone across all channels: **DLinear** (Zeng et al., 2023), **PAttn** (Tan et al., 2024), and **PatchTST** (Nie et al., 2023). These methods use linear projection, attention mechanisms, and the Transformer architecture, respectively, to model temporal dependencies.

(b) **Channel-Dependent.** We adopt five methods that incorporate dedicated modules to capture inter-channel dependencies. **TimesNet** (Wu et al., 2022) and **TSMixer** (Chen et al., 2023) are position-wise methods that capture channel correlations at each time step by embedding. **iTransformer** (Liu et al., 2024a) is a token-wise method that treats the entire time series of each channel as a token and models inter-channel dependencies using a Transformer. Additionally, we consider two clustering-based methods: **CCM** (Chen et al., 2024) and **DUET** (Qiu et al., 2024b), which explicitly group highly correlated channels into clusters and model dependencies within each cluster.

**Benchmark Results.** The experimental setup is detailed in Appendix I. The results are shown in Table 4. We summarize the following observations from our benchmark results: **First**, although CI methods cannot explicitly model channel correlations, they are designed to learn shared temporal patterns across channels. This can help reduce the risk of overfitting, enabling CI methods to achieve competitive performance. Among them, PAttn generally achieves better results, indicating the effectiveness of attention mechanisms for temporal modeling. **Second**, among CD methods, iTransformer significantly outperforms others, suggesting that treating each channel as a token and using attention to model inter-channel dependencies is more effective than position-wise or cluster-based approaches in high-dimensional settings. We argue that the token-wise design helps mitigate noise in the temporal dimension and fully utilizes the benefits of the attention mechanism.

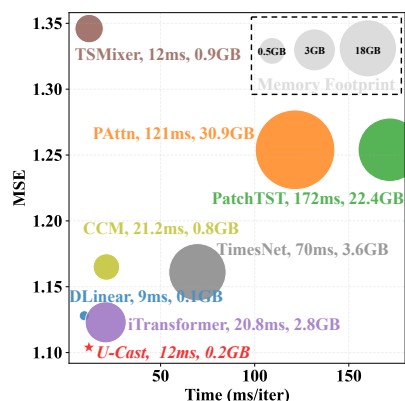

Figure 2: MSE vs. average training time per batch on Wind dataset ($C = 3{,}850$); marker size reflects GPU memory usage.

**Third**, CD methods do not show a consistent or significant performance advantage over CI methods. However, our theoretical analysis in Section 3 suggests that CD can reduce the MSE risk compared to CI in multivariate time series forecasting. We believe this is due to current CD baselines either failing to effectively exploit complex channel dependencies or being unable not scale to high-dimensional time series.

**Performance Gains of U-CAST.** As shown in Table 4, U-CAST achieves top performance on the high-dimensional TIME-HD benchmark, ranking **first** on 12 datasets for MSE and 11 for MAE,

and further achieves an average forecasting error reduction of 15% compared to iTransformer (t-test p-value=$1.34*10^{-5}$). While iTransformer, the representative CD model that explicitly models multivariate correlations via Transformer, serves as a strong baseline, it underperforms on many datasets compared to U-CAST. This highlights the advantage of learning hierarchical latent channel structures for capturing complex inter-channel dependencies. The limitation of iTransformer may stem from the extreme complexity of channel dependencies in high-dimensional settings, where the absence of explicitly hierarchical structure learning in its attention mechanism constrains its effectiveness. In contrast, by incorporating hierarchical latent queries and full-rank regularization, U-CAST is better equipped to handle such complexity. We also report U-CAST's performance on traditional low-dimensional datasets in Table 7, with further discussion in Appendix B.

**Model Efficiency.** Beyond its strong forecasting performance, another notable advantage of U-CAST is its lightweight architecture. Figure 2 shows U-CAST achieves the best trade-off between training speed, memory footprint, and performance. We also find that U-CAST consistently achieves favorable training efficiency across varying dimensionalities and as dimensionality increases, the efficiency advantage of U-CAST over other methods becomes more pronounced (see Appendix V). Additionally, we conduct computational complexity analysis, which indicates that U-CAST lowers both time and memory by a factor of $r$ compared with iTransformer, yet still retains the expressive power of attention through its latent-query hierarchy (see Appendix O).

**Ablation Study.** We evaluate the impact of: (1) **w/o hierarchical** by retaining only a single layer for dimensionality reduction, (2) **w/o latent query** by setting $\mathbf{Q}_\ell$ requires_grad=False, and (3)**w/o upsampling** by using a simple linear projection to restore channel dimension. In addition, we examine the role of the covariance full-rank regularisation by **w/o** $\mathcal{L}_{\mathbf{cov}}$, i.e., setting $\alpha = 0$. Table 5 shows that removing any component degrades U-CAST's averaged performance, confirming their necessity. To conduct a deeper analysis and isolate the contribution of each architectural component, we conduct a stepwise ablation study from MLP to U-CAST by progressively adding the proposed modules. The results show that each added component leads to a measurable improvement. For the full results on all datasets and deeper analysis, we refer the reader to Appendix U.

Table 5: Ablation studies on U-CAST. All results are averaged over 16 datasets. Full per-dataset results are provided in Tables 16 and U.

| Variant | MSE | MAE |
|---|---|---|
| **U-CAST** | **1.243** | **0.326** |
| w/o $\mathcal{L}_{cov}$ | 1.267 | 0.341 |
| w/o Hierarchical | 1.263 | 0.332 |
| w/o Latent Query | 1.260 | 0.331 |
| w/o Upsampling | 1.269 | 0.336 |
| **MLP+Hierarchical Query Attn+$\mathcal{L}_{cov}$** | **1.243** | **0.326** |
| MLP+Query Attn+$\mathcal{L}_{cov}$ | 1.263 | 0.332 |
| MLP+Hierarchical Query Attn | 1.267 | 0.341 |
| MLP+Query Attn | 1.293 | 0.348 |
| MLP+Attn | 1.305 | 0.369 |
| MLP | 1.317 | 0.372 |

**Does full-rank regularization $\mathcal{L}_{\mathbf{cov}}$ disentangle channels correlation effectively?** As shown in Figure 3 (a), the top four subplots visualize the evolution of the covariance matrix $\boldsymbol{\Sigma}$ from a randomly initialized state (Epoch 0) to a well-optimized one (Epoch 10). The structure of the covariance matrix changes significantly across epochs, transitioning from dense to more sparse. This indicates that $\mathcal{L}_{cov}$ is effectively promoting disentanglement by reducing redundancy among channels.

**Can U-CAST learn latent hierarchical structure among channels?** As shown in Figure 3 (b), we visualize the attention maps at different layers. The attention focuses vary across layers, reflecting a latent hierarchical structure in the learned representations. Furthermore, assigning dimension reduction and multivariate correlation modeling to the attention mechanism improves the interpretability of attention maps. Figure 3 (c) shows attention maps from the hierarchical latent query network on the Measles dataset ($C =$

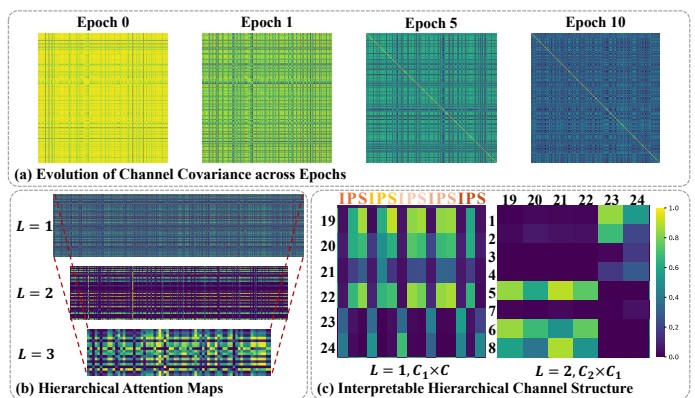

Figure 3: A showcase on Measles dataset of full-rank regularization and latent hierarchical channel structure learning.

1161, 387 regions × 3 features: **I**–Inferred Infections, **P**–Population, **S**–Suspected Cases). Five regions are selected (colors indicate regions). At $L = 1$, the model reduces to $C_1 = 32$ latent dimensions (visualizing 19–24), and at $L = 2$, to $C_2 = 8$. At $L = 1$, dimensions 19–22 mainly attend to P and S, while 23–24 focus on I. At $L = 2$, dimensions 1–4 attend to outputs from 19–22, and 5–8 to those from 23–24. This pattern reflects a learned hierarchy: the model first separates features within regions, then integrates related features across regions, revealing a meaningful latent channel structure.

## 7 CONCLUSION

In this paper, we propose U-CAST to address the unique challenges of effectively modeling these latent hierarchical channel structures within High-Dimensional Time Series Forecasting (HDTSF). To facilitate research in HDTSF and provide a testbed for future innovative approaches, we curate the TIME-HD benchmark. Through extensive experiments, we identify and highlight key research directions and opportunities in Appendix E. It is our aspiration that U-CAST, TIME-HD, TIME-HD-LIB, and their accompanying resources will serve as a significant catalyst for innovation and progress in the time series community (see Appendix F for broader impact).

ETHICS STATEMENT

**We have adhered to the ICLR Code of Ethics in preparing this submission.** This work does not involve human subjects, personally identifiable data, or sensitive information. All datasets used are publicly available benchmark datasets, and we follow their respective usage and licensing guidelines. The proposed methods are designed for advancing research in high-dimensional time series forecasting and do not raise foreseeable risks of harm.

REPRODUCIBILITY STATEMENT

We provide an anonymous repository containing the full source code and implementation details of our proposed U-CAST and TIME-HD-LIB pipeline at `https://anonymous.4open.science/r/Time-HD-Lib-1A71`. All datasets of TIME-HD used in our experiments are available at an anonymous HuggingFace repo: `https://huggingface.co/datasets/Time-HD-Anonymous/High_Dimensional_Time_Series`. Detailed descriptions of model architectures, training protocols, and hyperparameters are included in the main text and appendix. These resources are intended to ensure that all reported results can be independently reproduced.

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

# THE USE OF LARGE LANGUAGE MODELS (LLMs)

LLMs were used solely for language polishing and had no role in research or experiments.

## A    COMPARISON WITH EXISTING WORKS

Table 6: Comparison of datasets in benchmarks and foundation models on four properties: (1) **Multivariate** indicates whether the datasets involve multiple variables (more than one channel). (2) **High-Dimensional (≥1000)** refers to the presence of datasets with at least 1000 channels, either for pretraining or evaluation. (3) **Temporal Alignment** denotes whether the datasets are temporally aligned. (4) **Evaluation** assesses whether high-dimensional datasets are used specifically for evaluation purposes.

| Property | Multivariate | High-Dimensional (≥1000) | Temporal Alignment | Evaluation |
|---|---|---|---|---|
| **Benchmark** | | | | |
| M3 (Makridakis & Hibon, 2000) | ✗ | ✗ | ✗ | ✗ |
| M4 (Makridakis et al., 2018) | ✗ | ✗ | ✗ | ✗ |
| LTSF-Linear (Zeng et al., 2023) | ✓ | ✗ | ✓ | ✗ |
| TSlib (Wu et al., 2022) | ✓ | ✗ | ✓ | ✗ |
| BasicTS (Liang et al., 2022) | ✓ | ✗ | ✓ | ✗ |
| BasicTS+ (Liang et al., 2022) | ✓ | ✗ | ✓ | ✗ |
| Monash (Godahewa et al., 2021) | ✗ | ✗ | ✗ | ✗ |
| Libra (Bauer et al., 2021) | ✗ | ✗ | ✗ | ✗ |
| TFB (Qiu et al., 2024a) | ✓ | ✓ (only one) | ✓ (only one) | ✓ (only one) |
| **Foundation Models** | | | | |
| Chronos (Ansari et al., 2024) | ✓ | ✓ | ✗ | ✗ |
| Moment (Goswami et al., 2024) | ✓ | ✓ | ✗ | ✗ |
| Moirai (Woo et al., 2024) | ✓ | ✓ | ✗ | ✗ |
| TimesFM (Das et al., 2024) | ✓ | ✓ | ✗ | ✗ |
| Time-MoE (Shi et al., 2024b) | ✓ | ✓ (only one) | ✓ (only one) | ✓ (only one) |
| **Ours** | | | | |
| TIME-HD | ✓ | ✓ | ✓ | ✓ |

**High-Dimensional Time Series Datasets.** As shown in Table 6, most existing benchmarks do not include any high-dimensional datasets. The only exception is TFB (Qiu et al., 2024a), which incorporates only one dataset with 2000 channels (Wiki2000) for evaluation. Among foundation models, although all of them include high-dimensional time series datasets, these datasets are primarily used for pretraining rather than evaluation. Time-MoE (Shi et al., 2024b) is the only one that evaluates on a high-dimensional dataset (Global Temp) with 1000 channels. Both TFB and Time-MoE attempt to explore the high-dimensional regime, but they each include **only one** such dataset. This limited coverage constrains their effectiveness as comprehensive benchmarks.

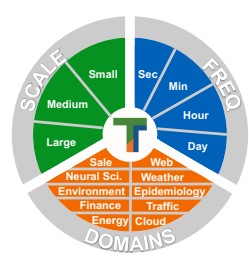

Figure 4: TIME-HD provides diverse high-dimensional datasets.

**Multivariate Time Series Forecasting.** *Channel-independent (CI)*: The CI modeling strategy uses a shared model backbone for all channels and processes each channel independently during the forward pass (Nie et al., 2023; Zeng et al., 2023; Xu et al., 2023; Wang et al., 2024a; Tan et al., 2024; Dai et al., 2024; Lin et al., 2024). This design generally results in lower model capacity but offers greater robustness. However, since CI models treat each channel independently, they are unable to capture inter-channel correlations explicitly.

*Channel-dependent (CD)*: In contrast, the CD modeling strategy incorporates modules that are specifically designed to capture dependencies across channels. These methods tend to offer higher model capacity and can leverage cross-channel information more effectively. Based on how the inter-channel correlations are handled, CD approaches can be categorized as follows: **(1) Position-wise** (Chen et al., 2023; Wu et al., 2022; 2021; Zhou et al., 2021; 2022; Zhang & Yan, 2023): These methods first project the channel dimension of each time step into a hidden embedding space, resulting in one embedding per time step. This approach may ignore correlations across channels at different time steps and could introduce noise. After generating the embeddings, these methods mainly focus

Table 7: Time series forecasting results on traditional low-dimensional datasets with prediction lengths $S = 96$. A lower MSE or MAE indicates a better prediction. The best performance is highlighted in **red**, and the second-best is underlined.

| Models | UCast | | iTransformer | | ModernTCN | | TimeMixer | | PatchTST | | MICN | | FEDformer | | TimesNet | | Autoformer | |
|--------|-------|-------|-------|-------|-------|-------|-------|-------|-------|-------|-------|-------|-------|-------|-------|-------|-------|-------|
| Metric | MSE | MAE | MSE | MAE | MSE | MAE | MSE | MAE | MSE | MAE | MSE | MAE | MSE | MAE | MSE | MAE | MSE | MAE |
| ETTh1 | **0.383** | **0.405** | 0.384 | 0.405 | 0.389 | 0.412 | 0.389 | 0.417 | 0.428 | 0.438 | 0.746 | 0.642 | 0.486 | 0.502 | 0.451 | 0.461 | 0.590 | 0.564 |
| ETTh2 | **0.278** | **0.337** | 0.286 | 0.342 | 0.288 | 0.350 | 0.323 | 0.384 | 0.338 | 0.386 | 0.395 | 0.427 | 0.410 | 0.457 | 0.415 | 0.446 | 1.173 | 0.824 |
| ETTm1 | **0.300** | **0.347** | 0.301 | 0.350 | 0.325 | 0.369 | 0.309 | 0.357 | 0.308 | 0.368 | 0.356 | 0.404 | 0.363 | 0.422 | 0.333 | 0.374 | 0.475 | 0.485 |
| ETTm2 | **0.159** | **0.250** | 0.165 | 0.256 | 0.180 | 0.269 | 0.197 | 0.292 | 0.181 | 0.273 | 0.215 | 0.311 | 0.298 | 0.362 | 0.202 | 0.289 | 0.309 | 0.374 |
| Weather | 0.159 | 0.210 | 0.161 | 0.210 | 0.152 | 0.208 | **0.147** | **0.202** | 0.158 | 0.214 | 0.171 | 0.231 | 0.333 | 0.395 | 0.174 | 0.234 | 0.372 | 0.419 |
| ECL | **0.127** | **0.221** | 0.134 | 0.230 | 0.140 | 0.239 | 0.132 | 0.227 | 0.135 | 0.239 | 0.169 | 0.283 | 0.238 | 0.347 | 0.218 | 0.324 | 0.224 | 0.334 |
| Traffic | **0.345** | **0.254** | 0.364 | 0.265 | 0.392 | 0.276 | 0.366 | 0.272 | 0.374 | 0.272 | 0.485 | 0.313 | 0.608 | 0.388 | 0.677 | 0.391 | 0.691 | 0.422 |

on modeling temporal dependencies. **(2) Token-wise** (Liu et al., 2024a; Wang et al., 2024c): These methods treat each channel as a token and input the sequence of tokens into a Transformer model. This helps reduce irrelevant noise for forecasting. Although multi-layer Transformers have the potential to capture complex hierarchical inter-channel structures, they are extremely time-consuming and do not scale well to high-dimensional time series. Moreover, the lack of an explicit mechanism for learning hierarchical channel structure may limit their effectiveness. **(3) Cluster-wise** (Chen et al., 2024; Qiu et al., 2024b): These approaches divide the multivariate time series into disjoint channel clusters using clustering algorithms. However, using a single-layer clustering step may overlook the hierarchical organization present in high-dimensional time series, which can lead to suboptimal performance. Furthermore, the challenge of modeling channel dependencies is often framed as a Multi-Task Learning (MTL) problem, where each channel is a task. This has led to strategies like explicit clustering, such as MTLinear (Nochumsohn et al., 2025), which groups variables by correlation and assigns separate linear heads to each group. Other MTL strategies use regularization to manage task relationships. For example, Ilbert (Ilbert et al., 2024) proposes a framework to decompose models into shared and task-specific components, allowing univariate models like PatchTST (Nie et al., 2023) to leverage multivariate information. While valuable, these methods often rely on predefined (e.g., correlation-based) or single-level groupings.

# B  U-CAST'S PERFORMANCE ON TRADITIONAL LOW-DIMENSIONAL DATASETS

To further evaluate U-CAST on widely used low-dimensional datasets, we include results on ETTh1, ETTh2, ETTm1, ETTm2, Weather, ECL, and Traffic with a prediction length of $S = 96$. These datasets contain far fewer channels than our benchmark and therefore serve as a useful check on whether $S = 96$ behaves reasonably outside the high-dimensional regime.

The results in Table 7 show that UCast remains competitive even though it is not designed for these settings. U-CAST achieves the best performance on six of the seven datasets. The only exception is Weather, where TimeMixer reaches the lowest error. This pattern suggests that the model generalizes well and is not tuned specifically to favor the proposed benchmark.

These findings help confirm that the new benchmark does not produce an artificial advantage. Instead, U-CAST performs well on both traditional low-dimensional datasets and the proposed high-channel datasets, which strengthens the reliability of the overall evaluation.

# C  FOUNDATION MODELS' PERFORMANCE ON TIME-HD

We conduct experiments with Time-MoE-base (50M) (Shi et al., 2024a) and Sundial-base (128M) (Liu et al., 2025a) as time series foundation model baselines. To keep the evaluation feasible, we select several moderate-sized datasets from TIME-HD, because deploying foundation models on the large datasets is extremely time-consuming.

As shown in Table 8, we mark datasets that appear in the training sets of the foundation models as **IID**, and datasets that the foundation models never observed as **OOD**. The results show that both Time-MoE and Sundial perform better on IID datasets than on OOD datasets, which is expected.

Table 8: Comparison among U-Cast and time series foundation models on Time-HD. Lower MSE/MAE indicates better performance.

| Datasets | | U-Cast | | Time-MoE | | Imp. | | Sundial | | Imp. | |
|---|---|---|---|---|---|---|---|---|---|---|---|
| | Metrics | MSE | MAE | MSE | MAE | MSE | MAE | MSE | MAE | MSE | MAE |
| **IID** | Air Quality | **0.446** | **0.430** | 0.454 | 0.432 | 1.83% | 0.27% | 0.5463 | 0.4729 | 18.34% | 8.96% |
| | Mobility | **0.315** | **0.317** | 0.367 | 0.356 | 13.96% | 11.07% | 0.3884 | 0.3669 | 18.78% | 13.60% |
| | M5 | **3.501** | **0.849** | 3.985 | 0.898 | 12.14% | 5.44% | 4.4909 | 0.9489 | 22.04% | 10.49% |
| | Wiki-20k | **10.273** | **0.302** | 12.521 | 0.356 | 17.95% | 15.13% | 12.0434 | 0.3511 | 14.70% | 13.94% |
| **OOD** | Measles | **0.010** | **0.042** | 0.019 | 0.055 | 50.72% | 23.79% | 0.0215 | 0.0629 | 55.57% | 32.76% |
| | SIRS | **0.007** | **0.052** | 0.371 | 0.373 | 98.05% | 86.06% | 0.2188 | 0.3059 | 96.70% | 82.98% |

However, because both models treat channels independently and cannot model channel correlation, they remain weaker than U-Cast across all datasets. This highlights that there is a clear need for efficient foundation models that can handle high-dimensional time series. We believe this points to a meaningful direction for future research.

## D CLARIFICATION ON "DIMENSIONALITY"

In conventional time series forecasting, the "dimensionality" can represent either different variables of a single object or variables collected from multiple related objects. We categorize the widely used datasets as follows:

(1) **Datasets that "dimensionality" refers to variables of a single object:**
   (a) ETT (Zhou et al., 2021): refer to 7 attributes of a single electricity transformer.
   (b) Weather (Wu et al., 2021): refer to 21 meteorological indicators.
(2) **Datasets that "dimensionality" refers to variables of multiple objects:**
   (a) ECL (Wu et al., 2021): refer to electricity consumption of 321 clients.
   (b) Traffic (Wu et al., 2021): refer to road occupancy rates recorded by 862 sensors.
   (c) Exchange (Lai et al., 2018): refer to daily exchange rates of 8 different countries.
   (d) Solar (Lai et al., 2018): refer to the solar power production records from 137 PV plants.
   (e) ILI (Wu et al., 2021): refer to the ratio of patients seen with influenzalike illness and the number of patients across 7 regions in US.
   (f) PEMS03/04/07/08 Liu et al. (2024a): refer to traffic flow records from 358/307/883/170 sensors, respectively.
   (g) Market Liu et al. (2024a): refer to the server load of Alipay online transactions of 285 to 759 servers.

Conventional time series forecasting includes both types of settings. In fact, datasets where dimensions correspond to multiple objects (such as sensors, regions, or clients) form the majority of commonly used datasets. Because the number of channels in single-object datasets is usually small and difficult to scale, our high-dimensional setting naturally focuses on the second category, which allows systematic study of forecasting under a truly high-dimensional setup.

## E FUTURE OPPORTUNITY

**Scaling Forecasting Models to High-Dimensional Time Series.** Most existing forecasting models are developed and benchmarked on low-dimensional datasets, which restricts their practical relevance in high-dimensional applications. As the number of variables increases, computational and memory bottlenecks become more prominent, especially for models employing attention mechanisms or dense channel-wise operations. Future research can prioritize the development of architectures that maintain accuracy while achieving sub-quadratic complexity in the number of channels. In particular, scalable attention mechanisms, parameter-efficient representations, and adaptive channel selection strategies are promising directions.

**Improving the Ability of Models to Capture Inter-Channel Correlations.** High-dimensional time series often exhibit rich and structured inter-channel dependencies, including spatial, semantic, and hierarchical patterns. While U-Cast demonstrates that learning latent hierarchical structures

can be effective, there remains room to improve in two aspects: (1) dynamic correlation modeling that adapts to temporal shifts in inter-variable relationships, and (2) domain-aware inductive biases that can leverage prior knowledge (e.g., spatial topology, sectoral information) to guide the learning of channel relationships. Better capturing these correlations can lead to models that are both more interpretable and more robust to distributional changes.

**Benchmarking and Standardization for High-Dimensional Forecasting.** Despite the introduction of the TIME-HD benchmark, the evaluation of forecasting models in high-dimensional regimes remains underdeveloped. Future work should explore more comprehensive evaluation protocols, including robustness to missing channels, adaptation to distribution drift, and generalization to unseen domains. Additionally, standardized metrics for scalability and memory efficiency, beyond predictive accuracy, are needed to holistically compare models in real-world deployment scenarios.

## F  BROADER IMPACT

This work reframes time series forecasting (TSF) around high-dimensional TSF (HDTSF), where models must reason over thousands of channels and their structure. By curating TIME-HD across more than ten domains and large channel counts, we move beyond the past practice of training and testing only on small, low-dimensional benchmarks. This shift makes evaluation more reliable, reveals failure modes that only appear at scale, and brings assessment closer to real deployments in sensor networks, markets, and geoscience. It also encourages the community to measure real gains rather than benchmark-specific gains.

TIME-HD and TIME-HD-LIB are released to lower the barrier to entry and raise reproducibility. The dataset suite standardizes timestamps, horizons, and metrics; the library offers a distributed train–validate–test pipeline with common preprocessing and reference configs. Together, they let researchers and practitioners start quickly, scale experiments without bespoke infrastructure, and add new datasets and tasks that broaden HDTSF coverage. This shared stack supports fair comparisons and faster iteration on modeling ideas.

Methodologically, U-CAST serves as a strong baseline for future work on HDTSF. Its design targets cross-channel structure and efficiency, yielding competitive accuracy with favorable speed and memory use. We recommend using U-CAST as a starting point for ablations and new architectures, reporting uncertainty alongside point forecasts, and disaggregating results by domain. Users should watch for dataset bias and compute cost; the released pipeline helps by making runs reproducible and efficient. Overall, the task, resources, and baseline aim to help the TSF community build and evaluate models that scale to realistic settings.

## G  DATASETS DESCRIPTION

**Temp** (Global Temp) and **Wind** (Global Wind). We adapt this dataset from Corrformer (Wu et al., 2023). Global Temp&Wind dataset is from the National Centers for Environmental Information (NCEI) [3]. This dataset contains the hourly averaged wind speed and hourly temperature of 3,850 stations (7,700 dimensions) around the world from 01/01/2019 to 12/31/2020, 17,544 timesteps in total. This dataset is well established. However, we find there are some constant columns in the datasets, which means the temp or wind speed never change. We remove these columns.

**Solar** (NREL Solar Power). NREL Solar Power is collected from National Renewable Energy Laboratory (NREL) [4]. One of the most popular datasets for time series forecasting is Solar-Energy (Lai et al., 2018), which is sourced from NREL. However, it includes data from only 137 solar photovoltaic (PV) plants in Alabama, limiting the dataset to 137 dimensions. To increase the dimensionality, we collect data from 5,166 PV plants across 47 U.S. states, excluding Alaska, Hawaii, and North Dakota, using data from NREL. The data source is clean so we don't conduct further preprocessing on it.

**Meters** (Smart meters in London). We process half-hourly smart meter energy consumption data by loading data from multiple blocks, where each block represents a subset of meters, and aggregating

---

[3]https://www.ncei.noaa.gov/
[4]https://www.nrel.gov/grid/solar-power-data.html

energy consumption data for each meter. The energy unit is kWh/hh. The data is sorted chronologically and merged into a single dataset, consisting of *5,567 meters and 40,405 timesteps*. **First**, to address missing values introduced by the staggered installation of meters, we filter out records before 07/14/2012, as meters were installed at different times starting from 11/23/2011. We also ensure that only records with exact half-hour intervals are retained, resulting in a dataset of *5,567 meters and 28,512 timesteps*. **Second**, we remove meters that became operational after 07/14/2012 or stopped reporting before 02/28/2014, reducing the dataset to *4,213 meters while maintaining 28,512 timesteps*. **Third**, we count missing values and remove meters with more than 10 missing values. The remaining missing values are replaced with zero, yielding a final dataset of *2,899 meters and 28,512 timesteps*. The cleaned and processed dataset is then saved for further analysis.

**Traffic-CA** (LargeST-CA), **Traffic-GLA** (LargeST-GLA), **Traffic-GBA** (LargeST-GBA). The source of this dataset is the Caltrans Performance Measurement System (PeMS)[5], and we utilize its preprocessed version(Liu et al., 2023). **LargeST-CA** encompassing a total of 8,600 sensors in California and each sensor contains five years of traffic flow data (from 2017 to 2021) with a 5-minute interval, resulting in a total of 525,888 time frames. **LargeST-CA consists of three subsets: (1) LargeST-GLA** includes 3,834 sensors installed across five counties in the Greater Los Angeles (GLA) area: Los Angeles, Orange, Riverside, San Bernardino, and Ventura. **(2) LargeST-GBA** contains 2,352 sensors distributed across 11 counties in the Greater Bay Area (GBA): Alameda, Contra Costa, Marin, Napa, San Benito, San Francisco, San Mateo, Santa Clara, Santa Cruz, Solano, and Sonoma. **(3) LargeST-SD**, the smallest subset, consists of 716 sensors located exclusively in San Diego County. Since LargeST-SD has only 716 dimensions, we do not include it in our benchmark. For the 525,888 original time frames, we aggregate the data into 1-hour intervals, resulting in 43,824 time frames. We then remove sensors with more than 100 missing (NaN) values, and apply linear interpolation to fill the remaining missing values.

**Neurolib.** Simulations made use of the Neurolib code (Cakan et al., 2021). We utilize the brain structure in ABIDE dataset (Craddock et al., 2013).

**SIRS.** Simulations extend the classical SIR model (Cai et al., 2015) by incorporating temporary immunity, resulting in dynamic transitions among Susceptible, Infected, and Recovered groups within a single region. To enhance realism, parameter noise, and seasonal variation are included. For multiple regions, each compartment has a fixed transfer rate between regions to simulate commuting and spatial transmission dynamics.

**Air Quality** (Chinese Air Quality). This dataset includes 7 air quality indicators: $AQI$, $CO$, $NO_2$, $O_3$, $PM10$, $PM2.5$, and $SO_2$ of 786 distinct cities in China. We remove dimensions with more than 100 missing values and interpolate other dimensions linearly, resulting in 1,106 dimensions.

**SP500** (S&P 500 Index). The S&P 500 is a stock market index maintained by S&P Dow Jones Indices [6]. It consists of 503 common stocks issued by 500 large-cap companies traded on American stock exchanges. We use yfinance [7], a Python package for financial data retrieval, to download market data for these stocks from Yahoo Finance [8]. We select the latest S&P 500 company list (as of 03/14/2025) and extract daily market data spanning the past 30 years (7,553 days). To ensure data consistency, we retain only the companies that were publicly traded 30 years ago, reducing the dataset to 295 companies. For each company, we extract five key market variables: Open, Close, High, Low, and Volume, resulting in a total of 1,475 dimensions.

**M5.** We adapt this dataset from a Kaggle competition (Howard et al., 2020). It covers stores in three U.S. states, i.e. California, Texas, and Wisconsin, and includes item-level sales data, department and product category details, and store information. The dataset records daily sales for each item from 07/01/2015 to 06/30/2022 (a total of 1,947 days) across different departments and stores, resulting in 30,490 time series. Due to its sparsity (i.e., a large number of zero values), we aggregate sales by summing each item's sales across all departments and stores, reducing the dataset to 3,049 aggregated items.

---

[5]https://pems.dot.ca.gov/

[6]https://www.spglobal.com/spdji/en/

[7]https://github.com/ranaroussi/yfinance

[8]https://finance.yahoo.com/

**Atec.** The dataset consists of 1589 traffic data collected every 10 minutes for different zones of different applications. We remove the dimensions that include more than 10 missing values and conduct interpolation for the remaining dimensions.

**Measles** (Measles England). The measles dataset (Madden et al., 2024; Liu et al., 2025b; 2024b), contains biweekly measles infections from 387 regions across England and Wales from 1944–1965. For each region, the dataset includes three features: inferred infections, population, and suspected cases, resulting in 1,161 dimensions.

**Wiki-20k** (Wikipedia Web Traffic). The dataset is adapted from the Monash Time Series Forecasting Repository [9]. It comprises 145,063 time series representing web traffic for various Wikipedia pages from July 1, 2015, to June 30, 2022 (2,557 days). This dataset is an extended version of the one used in the Kaggle Wikipedia Web Traffic Forecasting Competition [10]. Due to data sparsity, we remove the Wikipedia pages containing NaN values, resulting in a final dataset of 112,333 Wikipedia pages, each corresponding to one dimension. Due to the original dataset's dimension is too large and is not correlated, we select the first 20k dimensions to construct Wiki-20k dataset for evaluation.

**Mobility** (Google Community Mobility). The Community Mobility dataset (Google, 2020) aims to assist in analyzing the impact of COVID-19 on community movement patterns. The dataset covers movement trends over time (from February 15, 2020, to October 15, 2022, spanning 974 days) across 4,334 regions in different countries. It categorizes mobility data into **six** place types: **retail and recreation** (restaurants, cafes, shopping centers, theme parks, museums, libraries, and movie theaters), **groceries and pharmacies** (grocery stores, food warehouses, farmers markets, specialty food shops, drug stores, and pharmacies), **parks** (local and national parks, public beaches, marinas, dog parks, plazas, and public gardens), **transit stations** (subway, bus, and train stations), **workplaces** (places of employment), and **residential** (places of residence). The dataset initially consists of 26,004 dimensions, derived from the six place categories across 4,334 regions. However, due to data sparsity, we exclude regions containing NaN values, reducing the dataset to 971 regions and 5,826 dimensions. The values in the dataset represent changes in visit frequency and duration of stay at various locations relative to the baseline (i.e. the median value for the same day of the week during the five-week period from January 3 to February 6, 2020.). These changes are computed using aggregated and anonymized data, similar to the methodology employed in Google Maps to estimate popular visit times.

**Prediction Length** $S$   Earlier benchmarks commonly evaluated the same four horizons (96, 192, 336, 720) across all datasets, regardless of sampling frequency. This practice can yield scenarios with limited practical relevance, for example, requiring a weather model to predict 720 days ahead. In contrast, we align the prediction horizon with the data resolution so that each task corresponds to a *realistic operational need*:

1 ms–30 min data $\to$ $S = 336$ steps (0.34 s to 28 h)

1 h data $\to$ $S = 168$ steps (7 days)

6 h data $\to$ $S = 28$ steps (7 days)

1 day data $\to$ $S = 7$ steps (1 week)

These settings, listed under "*Pred Length*" in Table 3, reflect forecasting horizons commonly adopted by domain experts (e.g., weekly planning in retail or epidemiology).

## H   ANALYSIS OF CHANNEL CORRELATION IN TIME-HD

We follow TFB (Qiu et al., 2024a) to compute channel correlation. In multivariate time series, different variables often share common temporal patterns due to underlying shared factors. To quantify this, we extract 22 features per channel using Catch22 (Lubba et al., 2019). These features serve as fixed-length representations of each time series channel. We then compute the pairwise Pearson correlation coefficients (Cohen et al., 2009) between all channel representations and report the mean and variance.

---

[9]https://zenodo.org/records/7370977

[10]https://www.kaggle.com/c/web-traffic-time-series-forecasting/data

The results reported in the "*Correlation*" column of Table 3 show that all datasets exhibit **high average channel correlation**, reflecting strong *inter-variable dependencies* and a dominant *global component* shared across channels. While this global component is highly predictable, it provides little additional information for forecasting, since it largely repeats patterns common to all variables. When such shared trends dominate, they can overshadow the subtle yet critical channel-specific dynamics that ultimately determine predictive accuracy. Consequently, a model that focuses primarily on global trends will miss the fine-grained deviations that differentiate one channel from another. These findings motivate the need for models that explicitly disentangle and separate the global component from channel-specific signals while also capturing structured inter-channel dependencies.

**Modeling Strategy in U-CAST.** U-CAST addresses this challenge through an architecture that separates the modeling of unique channel dynamics from that of shared global trends (see Figure 1). Specifically:

- **Modeling Hierarchical Inter-Channel Dependency:** The *Hierarchical Latent Query Network* compresses the input into a low-dimensional latent space, where the *Full-Rank Regularization* (Section 4.2) enforces disentanglement and removes redundancy. The subsequent *Temporal Alignment* module then captures the essential channel-specific temporal patterns within this compact space.

- **Modeling Shared Trends:** After unique inter-channel dynamics are learned, the *Hierarchical Upsampling Network* and final *Output Projection* layer (Equation 6) reconstruct the forecasts for all channels. With a residual connection to the original embeddings, the projection layer re-applies the simple global trends on top of the channel-specific predictions from the latent space.

In essence, U-CAST decomposes the problem: it compresses and disentangles redundant global components to focus on informative inter-channel dynamics, while leveraging the output projection to efficiently recover global trends. This separation is not a loss of information but a deliberate design choice that enables accurate and robust forecasting in high-dimensional, highly correlated time series.

We also want to highlight that, in the real world, there indeed exist time series collections with **low inter-channel correlations**. However, merging such weakly related time series into a single dataset is usually not meaningful because they do not provide complementary predictive information to each other. In contrast, the datasets in TIME-HD are specifically collected from scenarios with strong inter-channel correlations, where exploiting these correlations can meaningfully improve forecasting performance.

# I  IMPLEMENTATION DETAILS

**Metric Details.** We use Mean Square Error (MSE) and Mean Absolute Error (MAE) as our evaluation metrics, following (Liu et al., 2024a; Luo & Wang, 2024; Wang et al., 2024a; Wu et al., 2022; 2021; Zhou et al., 2021; Zeng et al., 2023):

$$\text{MSE} = \frac{1}{S \times C} \sum_{i=1}^{S} \sum_{j=1}^{C} (\mathbf{Y}_{ij} - \hat{\mathbf{Y}}_{ij})^2, \tag{8}$$

$$\text{MAE} = \frac{1}{S \times C} \sum_{i=1}^{S} \sum_{j=1}^{C} |\mathbf{Y}_{ij} - \hat{\mathbf{Y}}_{ij}|. \tag{9}$$

Here, $\mathbf{Y} \in \mathbb{R}^{S \times C}$ represents the ground truth, and $\hat{\mathbf{Y}} \in \mathbb{R}^{S \times C}$ represents the predictions. $S$ denotes the future prediction length, $C$ is the number of channels, and $Y_{ij}$ indicates the value at the $i$-th future time point for the $j$-th channel.

**Experiment Details.** All experiments are implemented in PyTorch (Paszke, 2019) and executed on a cluster equipped with 8 NVIDIA A100 GPUs and 16 NVIDIA H100 GPUs, requiring 3000+ GPU hours in total (mainly for performing hyperparameter searching for all models across all datasets). The baselines are trained using their default configurations as reported in their respective papers, and with further hyperparameter searching (Learning Rate in {0.01, 0.001, 0.0001}, Input Sequence

Length $T$ in $\{3 \times S, 4 \times S, 5 \times S\}$), where $S$ is prediction length. ADAM optimizer (Kingma & Ba, 2014) is used with MSE loss for the training of all the models. We apply early stopping with a patience value of 5 epochs. The batch size is initially set to 32. If an out-of-memory (OOM) error occurs, the batch size is automatically halved until the issue is resolved. For U-CAST, we default to using non-stationary (Liu et al., 2022) normalization for all datasets. We default to set the number of layers $L$ to 2, the hidden dimension $d$ to 512, and the reduction ratio $r$ to 16 for all the datasets. Following existing work (Qiu et al., 2024a; Bergmeir, 2024), we avoid using the drop-last operation to ensure fairness.@ Additional detailed U-CAST model configuration information is presented in Table 10. We conduct hyperparameter sensitivity analysis in Appendix T. For robustness, we perform five-seed runs on all datasets and calculate the standard deviation. As shown in Table 9, U-CAST trains reliably across all cases.

Table 9: Performance of U-CAST (five-seed runs) on all datasets. Results are reported as mean $\pm$ standard deviation.

| Dataset | MSE | MAE |
|---|---|---|
| Atec | $0.290 \pm 0.001$ | $0.278 \pm 0.002$ |
| Air Quality | $0.420 \pm 0.005$ | $0.411 \pm 0.004$ |
| Temp | $0.263 \pm 0.002$ | $0.384 \pm 0.001$ |
| Wind | $1.078 \pm 0.003$ | $0.686 \pm 0.002$ |
| Mobility | $0.317 \pm 0.004$ | $0.324 \pm 0.003$ |
| Traffic-CA | $0.060 \pm 0.001$ | $0.134 \pm 0.002$ |
| Traffic-GBA | $0.059 \pm 0.000$ | $0.128 \pm 0.000$ |
| Traffic-GLA | $0.059 \pm 0.000$ | $0.135 \pm 0.001$ |
| M5 | $3.472 \pm 0.001$ | $0.843 \pm 0.000$ |
| Measles | $0.014 \pm 0.005$ | $0.050 \pm 0.004$ |
| Neurolib | $1.747 \pm 0.005$ | $0.349 \pm 0.006$ |
| Solar | $0.159 \pm 0.001$ | $0.220 \pm 0.002$ |
| SIRS | $0.006 \pm 0.000$ | $0.048 \pm 0.001$ |
| Meters | $0.947 \pm 0.011$ | $0.550 \pm 0.009$ |
| SP500 | $0.501 \pm 0.008$ | $0.305 \pm 0.006$ |
| Wiki-20k | $10.275 \pm 0.005$ | $0.302 \pm 0.004$ |

Table 10: U-CAST hyperparameters used for each dataset.

| Dataset | Learning Rate | Input Sequence Length $T$ | $\alpha$ |
|---|---|---|---|
| Atec | 0.0005 | $3 \times S$ | 0.001 |
| Air Quality | 0.0005 | $3 \times S$ | 0.100 |
| Temp | 0.001 | $3 \times S$ | 0.001 |
| Wind | 0.0005 | $4 \times S$ | 0.010 |
| Mobility | 0.0005 | $4 \times S$ | 0.010 |
| Traffic-CA | 0.002 | $4 \times S$ | 0.010 |
| Traffic-GBA | 0.002 | $4 \times S$ | 0.010 |
| Traffic-GLA | 0.002 | $4 \times S$ | 0.010 |
| M5 | 0.001 | $4 \times S$ | 10.000 |
| Measles | 0.0005 | $4 \times S$ | 0.010 |
| Neurolib | 0.0005 | $4 \times S$ | 0.010 |
| Solar | 0.001 | $4 \times S$ | 0.010 |
| SIRS | 0.001 | $4 \times S$ | 0.001 |
| Meters | 0.0005 | $4 \times S$ | 0.010 |
| SP500 | 0.002 | $4 \times S$ | 0.001 |
| Wiki-20k | 0.001 | $4 \times S$ | 0.001 |

## J    THEOREM 1 PROOF

*Proof.* Our goal is to forecast $z_t = \begin{pmatrix} z_t^{(1)} \\ z_t^{(2)} \end{pmatrix}$. For simplicity, we focus on forecasting $z_{t+1}^{(1)}$; forecasting $z_{t+1}^{(2)}$ is analogous. In a single forward process, the CI forecaster observes only $x = z_t^{(1)}$ and does **not** observe $w = z_t^{(2)}$. The optimal forecast under squared-error loss is the conditional expectation $\mathbb{E}[z_{t+1}^{(1)} \mid x]$. From the Equation 1, $z_{t+1}^{(i)} = a_{11}x + a_{12}w + \varepsilon$, where $\varepsilon = \varepsilon_{t+1}^{(1)} \sim \mathcal{N}(0, \sigma_{11})$. The conditional distribution of $w$ given $x$ is Gaussian. The optimal forecast is:

$$\mathbb{E}[z_{t+1}^{(1)} \mid x] = \mathbb{E}[a_{11}x + a_{12}w + \varepsilon \mid x] = a_{11}x + a_{12}\mathbb{E}[w \mid x] + \mathbb{E}[\varepsilon \mid x].$$

Since $\varepsilon$ is independent of $x$ (and $w$), $\mathbb{E}[\varepsilon \mid x] = \mathbb{E}[\varepsilon] = 0$ (as $\varepsilon$ has zero mean). Thus,

$$\mathbb{E}[z_{t+1}^{(1)} \mid x] = a_{11}x + a_{12}\mathbb{E}[w \mid x].$$

The CI Bayes risk is the expected squared error of this optimal forecast:

$$\begin{aligned} R_{\text{CI}} &= \mathbb{E}[(z_{t+1}^{(1)} - \mathbb{E}[z_{t+1}^{(1)} \mid x])^2] \\ &= \mathbb{E}[(a_{11}x + a_{12}w + \varepsilon - (a_{11}x + a_{12}\mathbb{E}[w \mid x]))^2] \\ &= \mathbb{E}[(a_{12}(w - \mathbb{E}[w \mid x]) + \varepsilon)^2] \\ &= \mathbb{E}[a_{12}^2(w - \mathbb{E}[w \mid x])^2] + \mathbb{E}[\varepsilon^2] + 2a_{12}\mathbb{E}[(w - \mathbb{E}[w \mid x])\varepsilon]. \end{aligned}$$

Again, using the independence of $\varepsilon$ from $w$ and $x$ (and thus from $\mathbb{E}[w \mid x]$), the cross-term expectation is zero: $\mathbb{E}[(w - \mathbb{E}[w \mid x])\varepsilon] = \mathbb{E}[w - \mathbb{E}[w \mid x]]\mathbb{E}[\varepsilon] = 0$. The first term is $a_{12}^2$ times the definition of the conditional variance, $\text{Var}(w \mid x) = \mathbb{E}[(w - \mathbb{E}[w \mid x])^2]$, since $w$ has zero mean (due to normalization) and $\mathbb{E}[w \mid x]$ is the conditional mean. Therefore,

$$\begin{aligned} R_{\text{CI}} &= a_{12}^2\text{Var}(w \mid x) + \mathbb{E}[\varepsilon^2] \\ &= \sigma_{11} + a_{12}^2\text{Var}(w \mid x). \end{aligned}$$

Note that our assumption that the time series are normalized (zero mean) simplifies the interpretation but the structure of the result holds more generally.

In a single forward process, the CD forecaster observes **both** $x = z_t^{(1)}$ and $w = z_t^{(2)}$. The optimal forecast is $\mathbb{E}[z_{t+1}^{(1)} \mid x, w]$.

$$\mathbb{E}[z_{t+1}^{(1)} \mid x, w] = \mathbb{E}[a_{11}x + a_{12}w + \varepsilon \mid x, w] = a_{11}x + a_{12}w + \mathbb{E}[\varepsilon \mid x, w].$$

Since $\varepsilon = \varepsilon_{t+1}^{(1)}$ is independent of both $x = z_t^{(1)}$ and $w = z_t^{(2)}$, $\mathbb{E}[\varepsilon \mid x, w] = \mathbb{E}[\varepsilon] = 0$. Thus,

$$\mathbb{E}[z_{t+1}^{(1)} \mid x, w] = a_{11}x + a_{12}w.$$

The CD Bayes risk is:

$$\begin{aligned} R_{\text{CD}} &= \mathbb{E}[(Y - \mathbb{E}[z_{t+1}^{(1)} \mid x, w])^2] \\ &= \mathbb{E}[(a_{11}x + a_{12}w + \varepsilon - (a_{11}x + a_{12}w))^2] \\ &= \mathbb{E}[\varepsilon^2] = \sigma_{11}. \end{aligned}$$

Comparing the two risks, we see that the reduction in risk from using the CD compared to the CI one is:

$$R_{\text{CI}} - R_{\text{CD}} = (\sigma_{11} + a_{12}^2\text{Var}(w \mid x)) - \sigma_{11} = a_{12}^2\text{Var}(w \mid x).$$

This difference is non-negative since $a_{12}^2 \geq 0$ and the conditional variance $\text{Var}(w \mid x) \geq 0$. The risk reduction is strictly positive if $a_{12} \neq 0$ and $\text{Var}(w \mid x) > 0$. This occurs when the second channel ($w$) has a non-zero direct influence on the first channel in the next period ($a_{12} \neq 0$) and when the second channel carries some information not already present in the first channel (i.e., $w$ is not perfectly predictable from $x$, ensuring $\text{Var}(w \mid x) > 0$). If $a_{12} = 0$, or if $w$ is perfectly predictable from $x$, the risks are equal. Similarly, if we focus on forecasting $z_{t+1}^{(2)}$, we have:

$$R_{\text{CI}} - R_{\text{CD}} = (\sigma_{22} + a_{21}^2\text{Var}(w \mid x)) - \sigma_{22} = a_{21}^2\text{Var}(x \mid w).$$

Thus, $\sum_i^2 (R_{\text{CI}} - R_{\text{CD}}) = a_{12}^2 \text{Var}(w \mid x) + a_{21}^2 \text{Var}(x \mid w) > 0$ This example demonstrates that, under squared-error loss for a linear Gaussian VAR model, using the CD leads to a Bayes risk that is always less than or equal to the Bayes risk obtained using only the CI.

$\square$

## K    THEOREM 2 PROOF

*Proof.* Let $\mathcal{F}_p = \sigma(z_t^{(1)}, \ldots, z_t^{(p)})$ denote the sigma-field generated by the first $p$ channels at time $t$. The Bayes risk under squared-error loss using information set $\mathcal{F}_p$ is:

$$R_p = \mathbb{E}\left[ (Y - \mathbb{E}[Y \mid \mathcal{F}_p])^2 \right].$$

Since conditional expectation minimizes mean squared error, and since the information sets are nested $(\mathcal{F}_1 \subseteq \cdots \subseteq \mathcal{F}_P)$, we have:

$$R_1 \geq R_2 \geq \cdots \geq R_P = \mathbb{E}[(\varepsilon_{t+1}^{(1)})^2] = \sigma_{11},$$

where the last equality holds because $\mathcal{F}_P$ fully determines the conditional mean of $Y$, and the only uncertainty is from the innovation $\varepsilon_{t+1}^{(1)}$.

Define the risk reduction relative to the univariate forecast as:

$$\Delta_p = R_1 - R_p.$$

The monotonicity of $R_p$ implies that $\Delta_p$ is non-decreasing:

$$0 = \Delta_1 \leq \Delta_2 \leq \cdots \leq \Delta_P = R_1 - \sigma_{11}.$$

To characterize when the inequality is strict, note that $\Delta_p > \Delta_{p-1}$ if and only if:

$$R_p < R_{p-1}.$$

This occurs if the $p$-th channel $z_t^{(p)}$ provides additional information about $Y$ beyond what is already available from $z_t^{(1)}, \ldots, z_t^{(p-1)}$. In the VAR(1) context, this requires:

- $a_{1p} \neq 0$, so that $z_t^{(p)}$ directly affects $Y$;

- $\text{Var}(z_t^{(p)} \mid z_t^{(1)}, \ldots, z_t^{(p-1)}) > 0$, so that $z_t^{(p)}$ is not a deterministic function of the previous channels.

If either condition fails, then $z_t^{(p)}$ does not improve the forecast of $Y$, and $R_p = R_{p-1}$. Therefore, each strictly decreasing step in Bayes risk corresponds to the inclusion of a new, informative, and non-redundant channel. $\square$

## L    THEOREM 3 PROOF

*Proof.* Write the compact singular–value decomposition of $\mathbf{H}^{(\ell-1)}$ as $\mathbf{H}^{(\ell-1)} = \mathbf{U}\,\boldsymbol{\Sigma}\,\mathbf{V}^\top$, where $\mathbf{U} \in \mathbb{R}^{C \times r}$, $\boldsymbol{\Sigma} \in \mathbb{R}^{r \times r}$ is diagonal with positive entries, $\mathbf{V} \in \mathbb{R}^{d \times r}$, and the columns of $\mathbf{U}$ form an orthonormal basis of the row-space of $\mathbf{H}^{(\ell-1)}$. Because $r \geq C'$ by assumption, we can pick a matrix $\mathbf{Q} \in \mathbb{R}^{C' \times C}$ of full row rank whose rows are any $C'$ independent linear combinations of the rows of $\mathbf{U}^\top$. Formally, let $\mathbf{S} \in \mathbb{R}^{C' \times C}$ select those $C'$ rows (so $\mathbf{S}\mathbf{S}^\top = \mathbf{I}_{C'}$), and set $\mathbf{Q} = \mathbf{S}$. Then

$$\mathbf{H}^{(\ell)} = \mathbf{Q}\mathbf{H}^{(\ell-1)} = \mathbf{S}\mathbf{U}\,\boldsymbol{\Sigma}\,\mathbf{V}^\top.$$

Since $\mathbf{S}\mathbf{U} \in \mathbb{R}^{C' \times r}$ has full row rank $C'$,

$$\text{rank}(\mathbf{H}^{(\ell)}) = \text{rank}(\mathbf{S}\mathbf{U}) = C',$$

which proves the first claim. Assume further that $r \geq d$ and $C' \geq d$, so the maximal attainable row rank is $d$. Define $\boldsymbol{\Sigma}^{(\ell)} = \frac{1}{d}\mathbf{H}^{(\ell)}\mathbf{H}^{(\ell)\top} \in \mathbb{R}^{C' \times C'}$. Adding the penalty $\mathcal{L}_{\text{cov}} = -\frac{1}{C'}\log\det(\boldsymbol{\Sigma}^{(\ell)} + \varepsilon\mathbf{I}_{C'})$ in training maximises $\det(\boldsymbol{\Sigma}^{(\ell)}) = \prod_{i=1}^{C'} \sigma_i^2$, where $\sigma_i$ are the singular values of $\mathbf{H}^{(\ell)}$. Because the product of the singular values grows when any $\sigma_i$ that is near zero increases, gradient descent on $-\log\det(\cdot)$ pushes all $\sigma_i$ away from zero, thereby driving $\mathbf{H}^{(\ell)}$ toward full row rank $\min(C', d)$.

**Compatibility with HLQN attention.** In HLQN the $\ell$-th block produces $\mathbf{H}^{(\ell)} = W_o\big(\text{softmax}(W_q\mathbf{Q}_\ell\,(W_k\mathbf{H}^{(\ell-1)})^\top/\sqrt{d_h})\,W_v\mathbf{H}^{(\ell-1)}\big)$. Write the attention weights as $\mathbf{A}(\mathbf{H}^{(\ell-1)}) = \text{softmax}\big(W_q\mathbf{Q}_\ell\,(W_k\mathbf{H}^{(\ell-1)})^\top/\sqrt{d_h}\big) \in \mathbb{R}^{C_\ell \times C}$. Then

$$\mathbf{H}^{(\ell)} = \underbrace{W_o\mathbf{A}(\mathbf{H}^{(\ell-1)})W_v}_{\mathbf{Q}(\mathbf{H}^{(\ell-1)}) \in \mathbb{R}^{C_\ell \times C}}\ \mathbf{H}^{(\ell-1)}.$$

Hence each forward pass is algebraically identical to left-multiplying $\mathbf{H}^{(\ell-1)}$ by a sample-dependent matrix $\mathbf{Q}(\mathbf{H}^{(\ell-1)})$. As long as $\mathbf{Q}(\mathbf{H}^{(\ell-1)})$ has full row rank $C_\ell$—a property encouraged by the LogDet regulariser—the rank analysis above applies verbatim: $\text{rank}\big(\mathbf{H}^{(\ell)}\big) = C_\ell$ and the gradient of the regulariser still pushes every singular value of $\mathbf{H}^{(\ell)}$ away from zero. Therefore the conclusions of Theorem 3 remain valid for the Hierarchical Latent Query Network attention block.

$\square$

## M   THEOREM OF INFORMATION REDUCTION

**Theorem 4** (Entropy Monotonicity of the Covariance Loss). *Assume the rows of $\mathbf{H}^{(\ell)} \in \mathbb{R}^{C' \times d}$ follow an i.i.d. zero-mean Gaussian distribution $\mathcal{N}\big(\mathbf{0}, \boldsymbol{\Sigma}^{(\ell)}\big)$ with $\boldsymbol{\Sigma}^{(\ell)} = \frac{1}{d}\mathbf{H}^{(\ell)}\mathbf{H}^{(\ell)\top} \succ \mathbf{0}$. Define the differential entropy*

$$h^{(\ell)} = \frac{1}{2}\log\Big((2\pi e)^{C'}\det(\boldsymbol{\Sigma}^{(\ell)})\Big).$$

*Let*

$$\mathcal{L}_{\text{cov}}^{(\ell)} = -\frac{1}{C'}\log\det\big(\boldsymbol{\Sigma}^{(\ell)} + \varepsilon I_{C'}\big), \qquad \varepsilon > 0.$$

*Then, along any differentiable optimisation trajectory $t \mapsto \mathbf{H}^{(\ell)}(t)$ that decreases $\mathcal{L}_{\text{cov}}^{(\ell)}$, the entropy satisfies*

$$\frac{\mathrm{d}}{\mathrm{d}t}h^{(\ell)}(t) = -\frac{C'}{2}\frac{\mathrm{d}}{\mathrm{d}t}\mathcal{L}_{\text{cov}}^{(\ell)}(t) \geq 0,$$

*with equality if and only if $\nabla_{\mathbf{H}^{(\ell)}}\mathcal{L}_{\text{cov}}^{(\ell)} = \mathbf{0}$. Hence every strict descent step on $\mathcal{L}_{\text{cov}}^{(\ell)}$ strictly increases the Shannon differential entropy of the latent channel distribution.*

*Proof.* For brevity drop the layer index $(\ell)$ and write $\boldsymbol{\Sigma} = \boldsymbol{\Sigma}^{(\ell)}(t)$. Since $\boldsymbol{\Sigma} \succ \mathbf{0}$ and $\varepsilon > 0$, $\boldsymbol{\Sigma} + \varepsilon I_{C'} \succ \mathbf{0}$. $\frac{\mathrm{d}}{\mathrm{d}t}\log\det\boldsymbol{\Sigma} = \text{tr}\big(\boldsymbol{\Sigma}^{-1}\dot{\boldsymbol{\Sigma}}\big)$, we obtain

$$\dot{h} = \frac{1}{2}\text{tr}\big(\boldsymbol{\Sigma}^{-1}\dot{\boldsymbol{\Sigma}}\big). \tag{S1}$$

By the same identity,

$$\dot{\mathcal{L}}_{\text{cov}} = -\frac{1}{C'}\text{tr}\Big((\boldsymbol{\Sigma} + \varepsilon I_{C'})^{-1}\dot{\boldsymbol{\Sigma}}\Big). \tag{S2}$$

Because $(\boldsymbol{\Sigma} + \varepsilon I_{C'})^{-1} \preceq \boldsymbol{\Sigma}^{-1}$ in the Löwner order (for $\varepsilon > 0$), we have

$$\text{tr}\Big((\boldsymbol{\Sigma} + \varepsilon I_{C'})^{-1}\dot{\boldsymbol{\Sigma}}\Big) \leq \text{tr}\big(\boldsymbol{\Sigma}^{-1}\dot{\boldsymbol{\Sigma}}\big). \tag{S3}$$

Combining (S1)–(S3) yields $\dot{h} \geq -\frac{C'}{2}\dot{\mathcal{L}}_{\text{cov}}$.

If the optimisation rule ensures $\dot{\mathcal{L}}_{\text{cov}} \leq 0$ (e.g. gradient descent or any monotone line search), then the inequality gives $\dot{h} \geq 0$. Equality holds only when $\dot{\mathcal{L}}_{\text{cov}} = 0$, which, under differentiability, implies the gradient of the loss is zero. Therefore entropy increases strictly whenever the loss decreases strictly. $\square$

## N   DETAILED EXPERIMENTAL SETUP FOR EMPIRICAL ANALYSIS

This appendix provides additional details on the synthetic experiments described in Section 3.2, including data generation, model architectures, training procedure, and evaluation.

## N.1 SYNTHETIC DATA GENERATION

We generate multivariate time series using a Vector Autoregression process:

$$z_{t+1} = A z_t + \varepsilon_{t+1}, \quad \varepsilon_{t+1} \overset{i.i.d.}{\sim} \mathcal{N}(\mathbf{0}, I),$$

where $z_t \in \mathbb{R}^C$, and the coefficient matrix $A \in \mathbb{R}^{C \times C}$ controls the temporal and inter-channel dependencies.

We consider the following dependency structures:

- **Independent**: $A$ is a diagonal matrix with entries drawn independently from $[0.8, 1.0]$. Each channel evolves independently without influence from others.
- **Anti-Self**: $A$ has zeros on the diagonal and off-diagonal entries drawn from $[0.5, 1.0]$. This ensures each channel is fully dependent on other channels but not on itself.

To ensure stability, we scale $A$ so that its spectral radius is less than 1. Time series are simulated with $T = 100$ time steps and dimensionality $C \in \{100, 250, 2000\}$. The first 10 steps are discarded to reduce initialization bias.

## N.2 FORECASTING TASK AND DATA PREPARATION

We define an autoregressive forecasting task where the model observes the past $L = 4$ time steps and predicts the next $H = 4$ steps:

$$\{z_{t-L+1}, \ldots, z_t\} \mapsto \{z_{t+1}, \ldots, z_{t+H}\}.$$

For each sequence, we construct overlapping sliding windows. The dataset is split into $80\%$ training and $20\%$ testing samples.

Before training, data is normalized using a channel-wise z-score computed over the training set:

$$\tilde{z}_t^{(i)} = \frac{z_t^{(i)} - \mu_i}{\sigma_i},$$

where $\mu_i$ and $\sigma_i$ are the mean and standard deviation of channel $i$.

## N.3 MODEL ARCHITECTURES

We compare the following two models:

- **CI**: Each channel is modeled independently using a shared linear model. Input shape is $[B, L, C]$, and output shape is $[B, H, C]$. Each channel uses a shared linear layer: $\mathbb{R}^L \to \mathbb{R}^H$.
- **CD**: Each channel is first forecasted independently (as in CI), and the resulting output is then transformed using a shared linear layer across the channel dimension: $\mathbb{R}^{H \times C} \to \mathbb{R}^{H \times C}$.

Both models are implemented in PyTorch and trained using the Adam optimizer.

## N.4 TRAINING DETAILS

Each model is trained for 100 epochs with a learning rate of 0.01 and batch size 32. We use mean squared error (MSE) as the loss function. To stabilize training, gradients are clipped with a maximum norm of 5. Models are trained on a single V100 GPU.

## N.5 EVALUATION METRICS

We report the mean squared error (MSE) averaged over all channels and forecast steps:

$$\text{MSE} = \frac{1}{HCN} \sum_{n=1}^{N} \sum_{h=1}^{H} \sum_{c=1}^{C} \left( \hat{z}_{t+h}^{(c)} - z_{t+h}^{(c)} \right)^2,$$

where $N$ is the number of test samples.

## O  COMPUTATIONAL COMPLEXITY

**Notation**    $C$ – number of channels (tokens in the *channel* dimension); $T$ – look-back length (time steps); $d$ – model width; $h$ – number of heads; $r > 1$ – reduction ratio; $L = \lceil \log_r C \rceil$ – number of encoder/decoder stages. Before the first attention layer, both UCast and iTransformer apply a *single* linear map $\mathbf{W}_{\text{in}} \in \mathbb{R}^{T \times d}$ that compresses each length-$T$ series to a $d$-dimensional embedding. Consequently, attention thereafter depends only on *channel counts*, not on $T$.

**Hierarchical Latent Query Network.**    At stage $\ell$ the query length is $M_\ell = C/r^{\,\ell}$ and the key/value length is $C_{\ell-1} = C/r^{\,(\ell-1)}$. Scaled-dot-product attention therefore costs

$$\mathcal{T}_{\text{attn}}^{(\ell)} = \mathcal{O}\big(M_\ell\,C_{\ell-1}\,d/h\big) = \mathcal{O}\Big(\frac{C^2 d}{h\,r^{\,2\ell-1}}\Big).$$

Summing the geometric series over $\ell = 1, \dots, L$ gives the encoder cost

$$\mathcal{T}_{\text{enc}} = \mathcal{O}\Big(\frac{C^2 d}{h\,r}\Big).$$

**Hierarchical Upsampling Network.**    The decoder is symmetric, so $\mathcal{T}_{\text{dec}} = \mathcal{O}\big(C^2 d/(h\,r)\big)$. Hence the *overall* time complexity of UCast is

$$\mathcal{T}_{\text{UCast}} = \mathcal{O}\Big(\tfrac{C^2 d}{h\,r}\Big).$$

Peak attention memory occurs in the first stage with $M_1 \times C = C^2/r$ score entries per head:

$$\mathcal{M}_{\text{UCast}} = \mathcal{O}\Big(\tfrac{C^2}{r}\Big).$$

**iTransformer**    iTransformer also compresses the time axis to width $d$ and then applies full self-attention *once* over the $C$ channel tokens:

$$\mathcal{T}_{\text{iTrans}} = \mathcal{O}\Big(\tfrac{C^2 d}{h}\Big), \qquad \mathcal{M}_{\text{iTrans}} = \mathcal{O}(C^2).$$

**Comparison**    The reduction ratio $r$ governs the savings:

$$\frac{\mathcal{T}_{\text{UCast}}}{\mathcal{T}_{\text{iTrans}}} = \frac{1}{r}, \qquad \frac{\mathcal{M}_{\text{UCast}}}{\mathcal{M}_{\text{iTrans}}} = \frac{1}{r}.$$

With a typical choice $r = 16$, UCast lowers both time and memory by a factor of 16 compared with a flat channel Transformer, yet still retains the expressive power of attention through its latent-query hierarchy.

## P  EMPIRICAL EVIDENCE OF HIERARCHICAL STRUCTURES

Table 11: CPCC values of hierarchical clustering applied to real-world datasets. Higher values (closer to 1) indicate stronger hierarchical structure.

| **Dataset** | M5 | Meter | Temp | Wind | Atec | Mobility | SP500 | Air Quality | Measles |
|---|---|---|---|---|---|---|---|---|---|
| **CPCC** | 0.7852 | 0.7705 | 0.9057 | 0.7478 | 0.8866 | 0.8338 | 0.9484 | 0.7429 | 0.8950 |

To empirically evaluate the hierarchical structure of real-world datasets, we apply hierarchical clustering to the data and assess the resulting structure using the **cophenetic correlation coefficient (CPCC)** (Saraçli et al., 2013). A CPCC value closer to 1 indicates that the dendrogram generated by hierarchical clustering preserves the original pairwise distances between data points more accurately (Kumar & Toshniwal, 2016), which implies that the features in the dataset exhibit a **stronger hierarchical structure**.

Table 12: CPCC values of hierarchical clustering on the latent representations $H^{(l)}$ for different encoder layers $l$. Higher CPCC indicates a stronger hierarchical structure.

| Dataset | Model | Layer | CPCC |
|---------|-------|-------|------|
| Measles | U-Cast | $L = 1$ | 0.88 |
| Measles | U-Cast | $L = 2$ | 0.91 |
| Measles | w/o $\mathcal{L}_{\text{cov}}$ | $l = 1$ | 0.65 |
| Measles | w/o $\mathcal{L}_{\text{cov}}$ | $l = 2$ | 0.40 |
| SIRS | U-Cast | $L = 1$ | 0.88 |
| SIRS | U-Cast | $L = 2$ | 0.93 |
| SIRS | w/o $\mathcal{L}_{\text{cov}}$ | $l = 1$ | 0.51 |
| SIRS | w/o $\mathcal{L}_{\text{cov}}$ | $l = 2$ | 0.48 |

## Q  HOW FULL-RANK REGULARIZATION IMPROVES HIERARCHICAL LEARNING

Full-rank regularization is essential for learning a consistent hierarchical structure across layers. We provide both the causal explanation and a quantitative analysis that connects the regularization term $\mathcal{L}_{\text{cov}}$ to reduced overfitting and improved latent hierarchy formation.

**Causal Explanation**  Full-rank regularization reduces the redundancy among channels in the latent representation. Models trained without this constraint often memorize co-linear channel interactions in the data, which produces latent features with high dependence. This dependence prevents the encoder from constructing a stable multi-level hierarchy. When $\mathcal{L}_{\text{cov}}$ is applied, the latent variables become disentangled. The disentanglement reduces the overlap between channels and enforces a representation that separates independent sources of variation. As stated in Theorem 3, removing linear redundancy is sufficient to reveal the hierarchical latent channel structure that is otherwise hidden inside correlated features. Theorem 4 shows that the regularization promotes full-rank latent covariance, which is the required condition for such disentanglement.

**Quantitative Evidence**  To measure the quality of the learned hierarchy, we compute the **cophenetic correlation coefficient (CPCC)** (Saraçli et al., 2013) for the latent representations $H^{(l)}$ at each encoder layer $l$. The CPCC score evaluates the consistency between the hierarchical clustering dendrogram and the pairwise distances of the encoded features. A CPCC value closer to 1 indicates a clearer and more coherent hierarchical structure (Kumar & Toshniwal, 2016). Table P reports the CPCC values for U-CAST and the ablated variant without $\mathcal{L}_{\text{cov}}$ on the Measles and SIRS datasets. The U-CAST model produces progressively stronger hierarchical structure as the depth increases, while the model without the regularization exhibits a collapse of the hierarchy at deeper layers.

Table 13: Redundancy analysis of latent representations using MAOC (lower is better) and NPR (higher is better). Results are reported for each encoder layer.

| Dataset | Model | Layer | MAOC $\downarrow$ | NPR $\uparrow$ |
|---------|-------|-------|--------|-----|
| Measles | U-CAST | $L = 1$ | 0.05 | 0.90 |
| Measles | U-CAST | $L = 2$ | 0.03 | 0.92 |
| Measles | w/o $\mathcal{L}_{\text{cov}}$ | $L = 1$ | 0.28 | 0.35 |
| Measles | w/o $\mathcal{L}_{\text{cov}}$ | $L = 2$ | 0.31 | 0.31 |
| SIRS | U-CAST | $L = 1$ | 0.02 | 0.95 |
| SIRS | U-CAST | $L = 2$ | 0.01 | 0.96 |
| SIRS | w/o $\mathcal{L}_{\text{cov}}$ | $L = 1$ | 0.35 | 0.28 |
| SIRS | w/o $\mathcal{L}_{\text{cov}}$ | $L = 2$ | 0.38 | 0.25 |

## R    QUANTITATIVE ANALYSIS ON DISENTANGLEMENT EFFECT

To quantitatively validate the disentanglement effect introduced by the full-rank regularization, we conduct a new experiment that directly compares the full U-CAST model (with $\mathcal{L}_{\text{cov}}$) against the ablated variant without $\mathcal{L}_{\text{cov}}$ (trained only with MSE). We evaluate the redundancy of the latent representations $H^{(l)}$ at each hierarchical layer ($l = 1, 2$) using two metrics:

**MAOC (Mean Absolute Off-Diagonal Correlation).** This metric measures the linear redundancy among latent channels. A lower value indicates reduced correlation and less redundancy.

**NPR (Normalized Participation Ratio).** This metric evaluates the effective rank of the representation space. A higher value indicates a representation with higher effective dimensionality and better disentanglement.

The results in Table P show that the full U-CAST model achieves consistently lower MAOC values across all layers, indicating substantially reduced channel redundancy. At the same time, U-CAST maintains near-ideal NPR values, demonstrating that the latent representations preserve high effective rank and avoid the representation collapse observed in the ablated model.

This analysis provides direct quantitative evidence that the full-rank regularization systematically reduces redundancy and prevents rank deficiency at each hierarchical layer. These findings confirm that the regularization is the key factor driving the disentanglement effect observed throughout the hierarchy.

## S    EMPIRICAL VALIDATION FOR CHANNEL DEPENDENCY FINDINGS

This section provides further empirical validation of the channel dependency findings by extending the analysis beyond synthetic settings. We examine both real-world data and controlled synthetic experiments to study how channel interactions influence the performance of channel-independent (CI) and channel-dependent (CD) models.

Table 14: MAE performance of CI model DLinear and CD model U-CAST on Wiki datasets with different channel subsets.

|           | Channels | DLinear | U-CAST |
|-----------|----------|---------|--------|
| Wiki-20k  | 20,000   | 0.394   | **0.302** |
| Wiki-10k  | 10,000   | 0.489   | **0.385** |
| Wiki-2k   | 2,000    | 0.669   | **0.659** |
| Wiki-0.2k | 200      | **0.697** | 0.782  |

**Real-World Validation**    To bridge the synthetic analysis to a real-world setting, we conducted an additional experiment. To reduce the effect of dataset-specific bias, we fixed the dataset to Wiki-20k, which is a **real-world** dataset, and randomly selected different numbers of channels to construct Wiki-10k, Wiki-2k, and Wiki-0.2k. We then evaluated the MAE performance of the CI model DLinear and the CD model U-CAST on these datasets. As shown in Table 14, CI models perform better when the dimensionality is **low**, but as the dimensionality increases, CD models gradually **outperform** CI models. This observation is consistent with the conclusion in Section 3.2.

Table 15: MSE for PAttn and U-CAST under different sythetic settings.

| Model       | Independent 100 | Anti-Self 100 | Anti-Self 250 | Anti-Self 500 |
|-------------|-----------------|---------------|---------------|---------------|
| PAttn (CI)  | 0.0039          | 0.0048        | 0.0049        | 0.0048        |
| U-Cast (CD) | 0.0052          | 0.0011        | 0.0010        | 0.0009        |

**Synthetic Validation with Advanced Models**    The preliminary analysis in Section 3.2 is based on two linear models. To give a more complete view, we add experiments using U-CAST (CD) and PAttn (CI) on the same synthetic datasets. The results are shown below. These results follow the

same trend as the linear models. When channels are independent, CI performs better; when channels have structured interactions, CD improves as the number of channels grows and outperforms CI.

## T  HYPERPARAMETER SENSITIVITY ANALYSIS

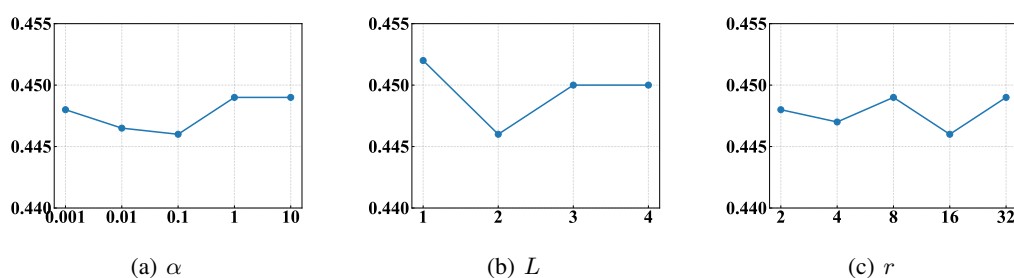

Figure 5: Performance (MSE) on Air Quality dataset under different hyperparameters.

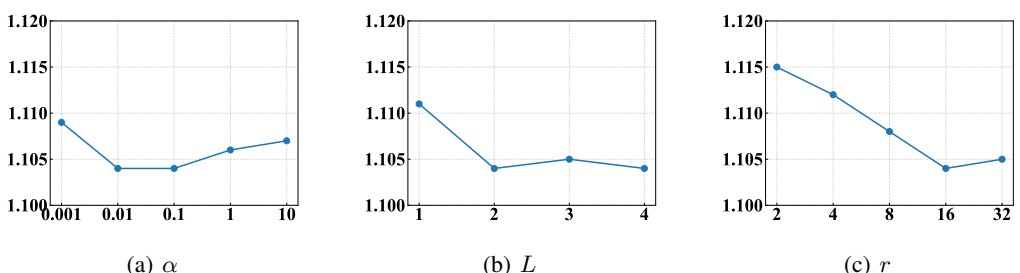

Figure 6: Performance (MSE) on Wind dataset under different hyperparameters.

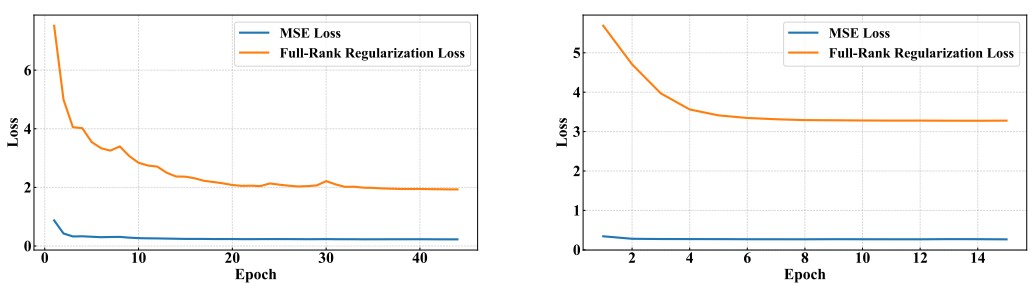

Figure 7: Comparison of loss scales between the MSE loss $\mathcal{L}_{mse}$ and the full-rank regularization loss $\mathcal{L}_{cov}$. **Left:** Atec, **Right:** Temp.

**Sensitivity Analysis of Full-Rank Regularization Strength** $\alpha$. We investigate how the regularization coefficient $\alpha$ influences the performance of U-CAST by varying it across several orders of magnitude. As shown in Figure 5(a) and 6(a), the model demonstrates high robustness to this hyperparameter. Even when $\alpha$ is varied across five orders of magnitude (from 0.001 to 10), the performance fluctuation is minimal across benchmarks. For instance, on the Air Quality dataset, the MSE only varies between 0.446 (best) and 0.449 (worst); on the Wind dataset, the range is similarly tight between 1.104 (best) and 1.109 (worst). This minimal variation across different benchmarks and wide parameter ranges confirms the robustness of the approach. The best performance is consistently achieved when $\alpha$ is set to a small value (0.01–0.1). This behavior can be explained by *the imbalance in the scales of the two losses*: the full-rank regularization loss $\mathcal{L}_{cov}$ is roughly an order of magnitude larger than the MSE loss $\mathcal{L}_{mse}$ (see Figure 7). If $\alpha$ is too large, the optimization is dominated by $\mathcal{L}_{cov}$, which may

overconstrain the latent channel representations and reduce the model's ability to capture informative structures. Using a smaller $\alpha$ helps balance the two objectives, stabilizes gradient updates, and leads to better forecasting accuracy.

**Sensitivity Analysis of Number of Hierarchical Layers** $L$. We evaluate the effect of the number of hierarchical latent query layers $L$ on forecasting performance, where a lower mean squared error (MSE) indicates better accuracy. As shown in Figure 5 (b), increasing $L$ from 2 to 3 reduces the MSE, indicating worse performance. This suggests that deeper hierarchies may not capture more useful structure and could lead to redundancy or overfitting. Based on this analysis, we use a unified setting of layer number $L = 2$ across all 16 datasets from different domains, which consistently achieves a stable trade-off between accuracy and efficiency without the need for per-dataset hyperparameter tuning.

**Sensitivity Analysis of Reduction Ratio** $r$. We investigate how the reduction ratio $r$, which determines the compression of channel representations at each layer, influences performance. As shown in Figure 5 (c), the model achieves relatively stable performance across a wide range of $r$ values. However, extremely large or small values can lead to slight degradation. For instance, very low reduction (e.g., $r = 2$) leads to higher computational cost without performance gain, while aggressive reduction (e.g., $r = 32$) may discard useful information. We find that $r = 16$ offers a favorable trade-off between efficiency and accuracy. Thus, we use a unified setting of layer number $r = 16$ across all 16 datasets.

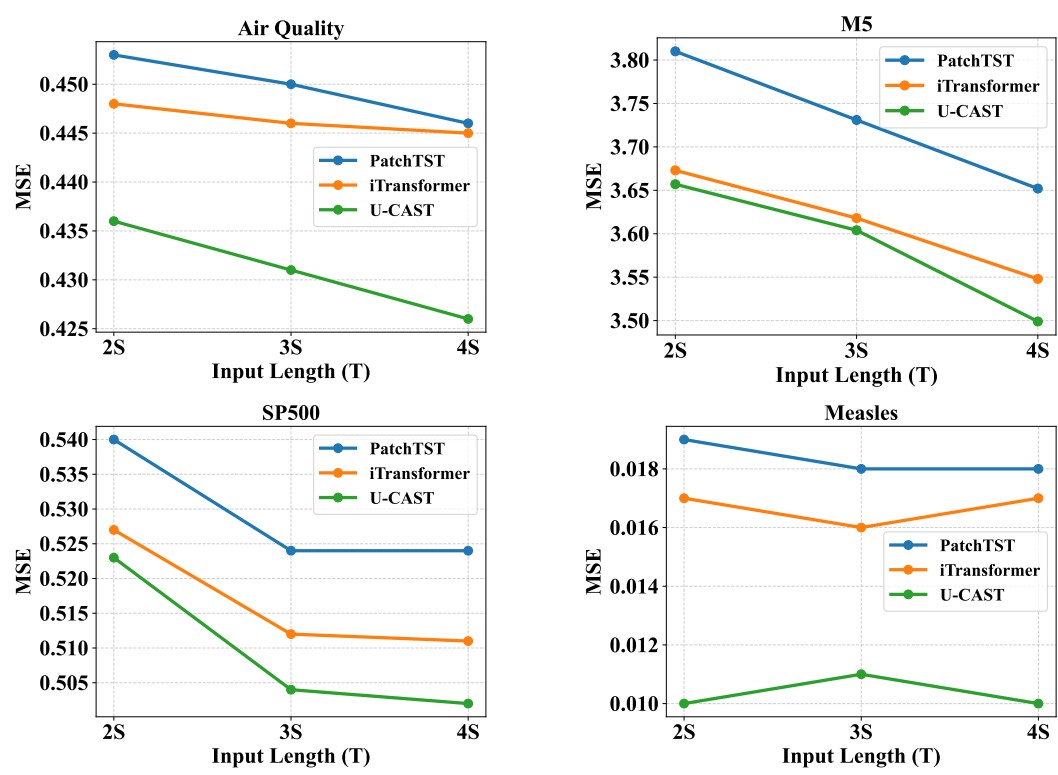

Figure 8: Comparison of performance (MSE) among U-CAST, PatchTST, and iTransformer on three datasets (Air Quality, M5, SIRS) under varying input lengths $T$.

**Sensitivity Analysis of Input Length** $T$. To directly assess the impact of input length and to further validate the findings from our TIME-HD benchmark, we conduct an extended experiment. We evaluate PatchTST, iTransformer, and U-CAST on three datasets across input lengths set to $2\times$, $3\times$, and $4\times$ the prediction horizon ($S$). The results are presented in Figure 8. These experiments lead to three main observations. **First**, longer input sequences generally improve performance in both CI and CD models on most datasets. **Second**, the effect varies by dataset. For instance, in the SIRS dataset, all models achieve their best results with the shortest input length, likely because its short-term epidemiological dynamics make longer histories add noise. This variability highlights

Table 16: Ablation study on different components of U-CAST. Each model is evaluated on 16 datasets using MSE and MAE. The best performance is highlighted in **red**, and the second-best is underlined.

| Dataset | U-CAST | | w/o $\mathcal{L}_{cov}$ | | w/o Hierarchical | | w/o Latent Query | | w/o Upsampling | |
|---|---|---|---|---|---|---|---|---|---|---|
| | MSE | MAE | MSE | MAE | MSE | MAE | MSE | MAE | MSE | MAE |
| Atec | **0.287** | **0.280** | 0.311 | 0.297 | 0.319 | 0.306 | 0.308 | 0.296 | 0.306 | 0.292 |
| Air Quality | **0.446** | **0.430** | 0.461 | 0.444 | 0.447 | 0.431 | 0.451 | 0.434 | 0.451 | 0.434 |
| Temp | **0.262** | **0.383** | 0.273 | 0.395 | 0.272 | 0.392 | 0.280 | 0.398 | 0.294 | 0.410 |
| Wind | **1.104** | **0.692** | 1.115 | 0.701 | 1.110 | 0.695 | 1.106 | **0.692** | 1.109 | 0.693 |
| Mobility | **0.315** | **0.317** | 0.336 | 0.332 | 0.316 | 0.319 | 0.326 | 0.323 | 0.327 | 0.324 |
| Traffic-CA | **0.061** | **0.131** | 0.072 | 0.144 | 0.062 | 0.138 | 0.062 | 0.136 | 0.062 | 0.133 |
| Traffic-GBA | **0.059** | **0.126** | 0.070 | 0.137 | 0.060 | 0.130 | 0.060 | 0.130 | 0.060 | 0.127 |
| Traffic-GLA | **0.060** | **0.132** | 0.071 | 0.144 | 0.061 | 0.136 | 0.061 | 0.135 | 0.061 | 0.134 |
| M5 | **3.501** | **0.849** | 3.520 | 0.861 | 3.566 | 0.854 | 3.516 | 0.852 | 3.597 | 0.860 |
| Measles | **0.010** | **0.042** | 0.022 | 0.061 | 0.019 | 0.054 | 0.011 | 0.051 | 0.014 | 0.063 |
| Neurolib | **1.750** | **0.350** | 1.770 | 0.362 | 1.818 | 0.370 | 1.813 | 0.358 | 1.756 | 0.355 |
| Solar | 0.172 | 0.246 | 0.182 | 0.242 | 0.173 | 0.239 | 0.175 | 0.237 | **0.167** | **0.222** |
| SIRS | **0.007** | **0.052** | 0.038 | 0.119 | 0.008 | 0.056 | 0.008 | 0.057 | 0.035 | 0.126 |
| Meters | **0.943** | 0.551 | 0.957 | 0.563 | 0.950 | 0.558 | 0.945 | **0.552** | 0.946 | 0.550 |
| SP500 | **0.502** | **0.301** | 0.561 | 0.338 | 0.562 | 0.333 | 0.566 | 0.335 | 0.597 | 0.349 |
| Wiki-20k | **10.273** | **0.302** | 10.520 | 0.321 | 10.459 | **0.309** | 10.509 | 0.310 | 10.527 | 0.310 |

Table 17: Stepwise ablation study from MLP to U-CAST. The best performance is highlighted in **red**, and the second-best is underlined.

| Dataset | MLP | | MLP+Attn | | MLP+QAttn | | MLP+QAttn+FR | | MLP+HQAttn | | MLP+HQAttn+FR | |
|---|---|---|---|---|---|---|---|---|---|---|---|---|
| | MSE | MAE | MSE | MAE | MSE | MAE | MSE | MAE | MSE | MAE | MSE | MAE |
| Atec | 0.323 | 0.319 | 0.347 | 0.321 | 0.317 | 0.303 | 0.319 | 0.306 | 0.311 | 0.297 | **0.287** | **0.280** |
| Air Quality | 0.456 | 0.453 | 0.449 | 0.432 | 0.471 | 0.453 | 0.447 | 0.431 | 0.461 | 0.444 | **0.446** | **0.430** |
| Temp | 0.276 | 0.397 | 0.267 | 0.388 | 0.279 | 0.402 | 0.272 | 0.392 | 0.273 | 0.395 | **0.262** | **0.383** |
| Wind | 1.146 | 0.708 | 1.114 | 0.699 | 1.137 | 0.715 | 1.110 | 0.695 | 1.115 | 0.701 | **1.104** | **0.692** |
| Mobility | 0.349 | 0.365 | 0.314 | 0.316 | 0.343 | 0.339 | 0.316 | 0.319 | 0.336 | 0.332 | **0.315** | **0.317** |
| Traffic-CA | 0.064 | 0.143 | 0.274 | 0.393 | 0.073 | 0.147 | 0.062 | 0.138 | 0.072 | 0.144 | **0.061** | **0.131** |
| Traffic-GBA | 0.063 | 0.140 | 0.066 | 0.144 | 0.071 | 0.139 | 0.060 | 0.130 | 0.070 | 0.137 | **0.059** | **0.126** |
| Traffic-GLA | 0.063 | 0.145 | 0.068 | 0.148 | 0.072 | 0.146 | 0.061 | 0.136 | 0.071 | 0.144 | **0.060** | **0.132** |
| M5 | 3.744 | 0.884 | 3.534 | 0.853 | 3.590 | 0.878 | 3.566 | 0.854 | 3.520 | 0.861 | **3.501** | **0.849** |
| Measles | 0.130 | 0.257 | 0.014 | 0.051 | 0.023 | 0.062 | 0.019 | 0.054 | 0.022 | 0.061 | **0.010** | **0.042** |
| Neurolib | 1.718 | 0.387 | 1.713 | 0.349 | 1.806 | 0.369 | 1.818 | 0.370 | 1.770 | 0.362 | **1.750** | **0.350** |
| Solar | 0.177 | 0.260 | 0.345 | 0.428 | 0.186 | 0.247 | 0.173 | 0.239 | 0.182 | 0.242 | **0.172** | **0.246** |
| SIRS | 0.059 | 0.171 | 0.032 | 0.116 | 0.038 | 0.122 | 0.007 | 0.052 | 0.038 | 0.119 | **0.007** | **0.052** |
| Meters | 0.959 | 0.558 | 0.948 | 0.556 | 0.976 | 0.574 | 0.950 | 0.558 | 0.957 | 0.563 | **0.943** | **0.551** |
| SP500 | 0.639 | 0.373 | 0.512 | 0.308 | 0.572 | 0.345 | 0.562 | 0.333 | 0.561 | 0.338 | **0.555** | **0.328** |
| Wiki20000 | 10.901 | 0.400 | 10.882 | 0.407 | 10.731 | 0.327 | 10.459 | 0.309 | 10.520 | 0.321 | **10.351** | **0.309** |
| Average | 1.317 | 0.372 | 1.305 | 0.369 | 1.293 | 0.348 | 1.263 | 0.332 | 1.267 | 0.341 | **1.243** | **0.326** |

the usefulness of TIME-HD as a benchmark for testing robustness under diverse temporal structures. **Third**, U-CAST remains competitive or superior across most datasets and horizons, and CD models (U-CAST and iTransformer) as a group continue to perform strongly when cross-channel structure provides informative signals.

## U   ABLATION STUDY

As U-CAST integrates two core components, the Hierarchical Latent Query Network and the Hierarchical Upsampling Network, we assess their individual contributions through ablation studies. Specifically, we evaluate the impact of: (1) **w/o hierarchical** by retaining only a single layer for dimensionality reduction, (2) **w/o latent query** by setting $\mathbf{Q}_\ell$ requires_grad=False, and (3)**w/o up-sampling** by using a simple linear projection to restore channel dimension. In addition, we examine the role of the covariance full-rank regularisation by **w/o** $\mathcal{L}_{cov}$, i.e., setting $\alpha = 0$.

Table 16 shows that removing any component generally degrades U-CAST's performance, confirming their necessity. **First**, *w/o latent query* still performs well on some datasets (e.g., NREL Solar Power), likely due to high inter-variable correlation (0.998), where random projections suffice. **Second**, *w/o upsampling* unexpectedly improves performance on NREL, suggesting that encoder representations are already sufficient and further decoding may introduce redundancy or overfitting. **Third**, removing

Table 18: Comparison among U-CAST, U-CAST-MLP, and U-CAST-Conv across datasets.

| Dataset | U-CAST | | U-CAST-MLP | | U-CAST-Conv | |
|---|---|---|---|---|---|---|
| | MSE | MAE | MSE | MAE | MSE | MAE |
| Atec | **0.287** | **0.280** | 0.323 | 0.315 | 0.337 | 0.329 |
| Air Quality | **0.446** | **0.430** | 0.520 | 0.502 | 0.498 | 0.481 |
| Mobility | **0.315** | **0.317** | 0.354 | 0.356 | 0.370 | 0.372 |
| M5 | **3.501** | **0.849** | 4.034 | 0.984 | 3.858 | 0.941 |
| Measles | **0.010** | **0.042** | 0.018 | 0.056 | 0.018 | 0.054 |
| SIRS | **0.007** | **0.052** | 0.015 | 0.064 | 0.015 | 0.067 |
| SP500 | **0.555** | **0.328** | 0.645 | 0.384 | 0.617 | 0.368 |
| Wiki-20k | **10.351** | **0.309** | 10.875 | 0.331 | 11.082 | 0.337 |

the covariance regularization has the largest impact on structured datasets like SIRS (MSE rises from 0.007 to 0.038), showing its importance in enforcing feature decorrelation.

**Stepwise ablation from MLP to U-Cast.** As shown in Table U, we further perform a stepwise ablation starting from a simple MLP and progressively adding each proposed component. Each variant adds one new design element, allowing us to isolate its contribution. To clarify the meaning of each model variant in the ablation table, we describe the role of every column:

(1) **MLP**: A three-layer MLP applied along the temporal dimension, without any channel interaction modules.
(2) **MLP+Attention (MLP+Attn)**: Adds two standard attention layers along the channel dimension, forming a `Linear → Attention → Linear → Attention → Linear` structure.
(3) **MLP+Query Attention (MLP+QAttn)**: Replaces standard attention with Query Attention, which performs channel up/down sampling and reduces redundant interactions.
(4) **MLP+Query Attention + Full-Rank Regularization (MLP+QAttn+FR)**: Adds full-rank regularization with $\ell = 1$, encouraging a more expressive channel mixing matrix.
(5) **MLP+Hierarchical Query Attention (MLP+HQAttn)**: Uses hierarchical query attention with $\ell = 2$ and $\alpha = 0$, enabling multi-level channel aggregation without regularization.
(6) **MLP+Hierarchical Query Attention + Full-Rank Regularization (MLP+HQAttn+FR)(U-Cast)**: Combines hierarchy and full-rank regularization, corresponding to the full U-Cast design.

**First**, *MLP+Attention* improves over the plain MLP on most datasets, which indicates that modeling channel correlation is important for high-dimensional forecasting. **Second**, *MLP+Query Attention* outperforms MLP+Attention, showing that query-guided aggregation reduces redundant interactions while improving accuracy and computational efficiency. **Third**, *MLP+Query Attention+Full Rank Regularization* performs slightly better than *MLP+Hierarchical Query Attention*, which shows that promoting full-rank structure has a stronger effect than adding hierarchy alone. **Finally**, the complete U-Cast model yields the best overall performance, demonstrating that hierarchical query attention and full-rank regularization contribute complementary benefits that jointly lead to the final gains.

**Substituting Attention with MLP or Convolution.** We choose not to use convolutions or MLP for up/down sampling channels because earlier work shows that attention is more effective than MLP for modeling channel correlation (see Table 3 in iTransformer Liu et al. (2024a)). In addition, channel ordering does not carry meaning. Convolutions are sensitive to ordering and restricted by a local receptive field, which makes them unsuitable for capturing channel correlation, while attention is order-invariant and therefore fits better.

To provide an empirical comparison between attention, convolution, and MLP for performance, we evaluate U-CAST and two variants, U-CAST-MLP and U-CAST-Conv, where "Query Attention" is replaced by an MLP and a 1D convolution, respectively. The results are shown in Table U. Across above eight datasets, U-CAST achieves lower MSE and MAE than both U-CAST-MLP and U-CAST-Conv. These results indicate that attention captures global channel dependencies more accurately than MLP and convolution, which rely on fixed local structures. The improvements are stable, suggesting that attention is a robust choice for modeling channel correlation in high-dimensional time series.

## V EFFICIENCY

We compare the training time per batch with recent state-of-the-art models in Figure 9 and Figure 10. In Figure 10, the Atec, Temp, and Wind datasets are removed to provide a clearer view of the efficiency trend at higher dimensions. The x-axis corresponds to different datasets, arranged from left to right in increasing order of dimensionality.

The results indicate that: **(1)** U-CAST consistently maintains favorable training efficiency across different dimensionalities; **(2)** Its efficiency advantage over other non-linear methods grows as dimensionality increases; **(3)** The efficiency of CI-based methods (e.g., PAttn, PatchTST) is sensitive to both channel and temporal dimensions. In Figure 10, when channel dimensionality is very large, as in Wiki-20k, their efficiency drops sharply; similarly, in Figure 9, longer input lengths (Atec, Temp, Wind) also lead to significant degradation; **(4)** As shown in Figure 10, the efficiency of CD-based methods (iTransformer, U-CAST) is mainly affected by the number of input channels. However, U-CAST is more robust than iTransformer, showing more stable efficiency as dimensionality increases.

The detailed efficiency comparison of both the training and inference phases is provided in Appendix Y. In both stages, U-CAST achieves the best trade-off between performance and efficiency.

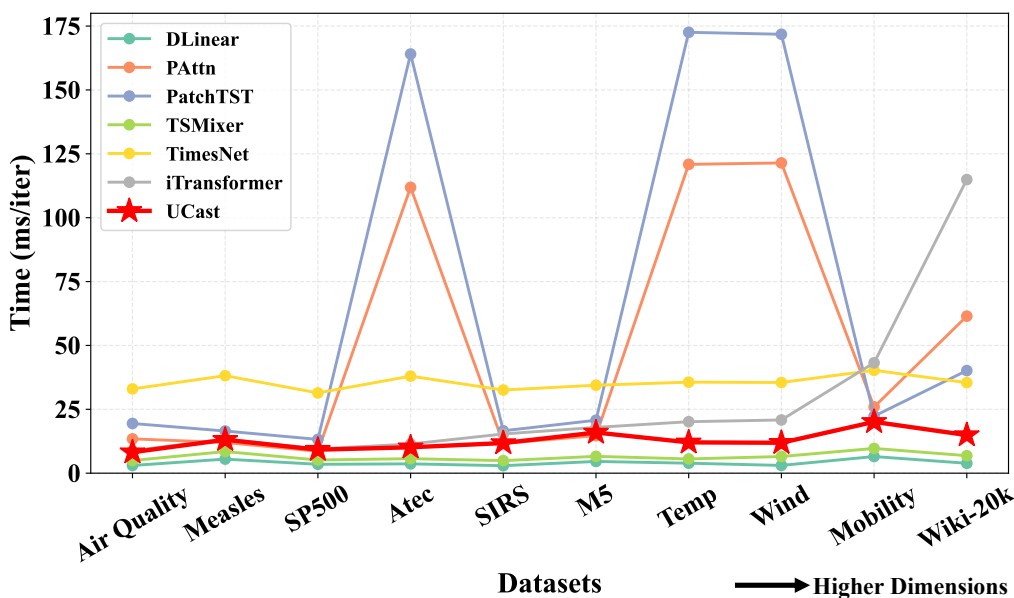

Figure 9: Training time across datasets. The x-axis represents different datasets, arranged from left to right in order of increasing dimensionality.

## W ADDITIONAL BASELINES

Table 19: Preliminary forecasting results of MCANN and NECPlus on selected datasets. Lower values indicate better performance.

|             | MCANN (MSE) | MCANN (MAE) | NECPlus (MSE) | NECPlus (MAE) |
|-------------|-------------|-------------|---------------|---------------|
| **Measles** | 1.727 | 0.851 | 1.160 | 0.722 |
| **Air Quality** | 0.466 | 0.457 | 0.447 | 0.449 |
| **Mobility** | 3.471 | 1.367 | 1.815 | 0.899 |
| **M5** | 17.031 | 1.441 | 16.842 | 1.387 |

We additionally include two methods, **MCANN** Li & Anastasiu (2025) and **NECPlus** Li et al. (2023), which further broaden the coverage of our benchmark. Both methods have been integrated into our

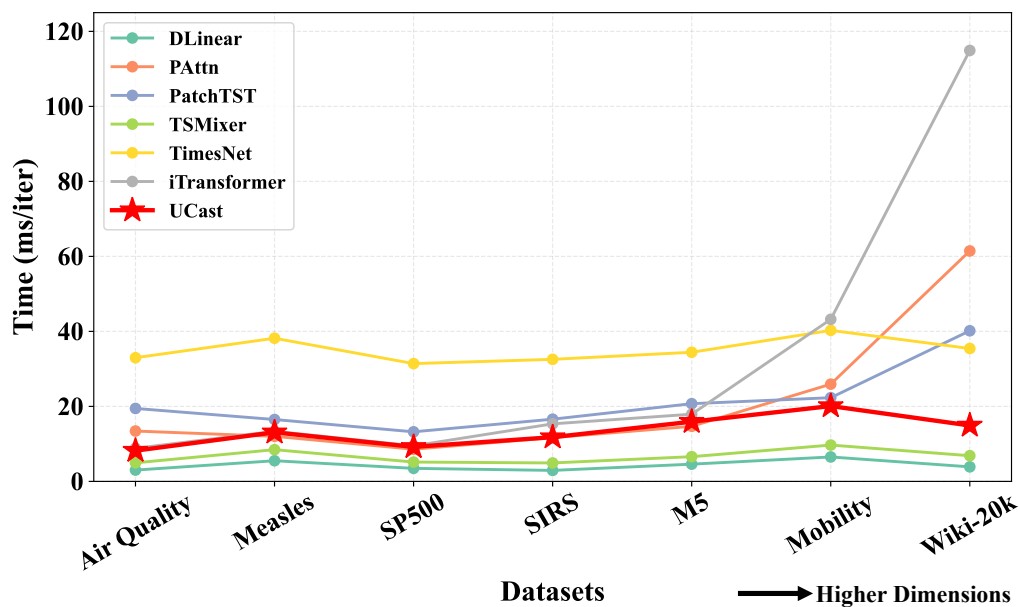

Figure 10: Training time across datasets (**w/o Atec, Temp, and Wind datasets**). The x-axis represents different datasets, arranged from left to right in order of increasing dimensionality.

**Time-HD-Lib**. For NECPlus, we adopt the standard model implementation. Preliminary results are presented in Table 19.

## X    SHOWCASES

For clear comparison, we present test set showcases in Appendix Z, where U-CAST shows better performance.

# Y    DETAILED EFFICIENCY COMPARISON

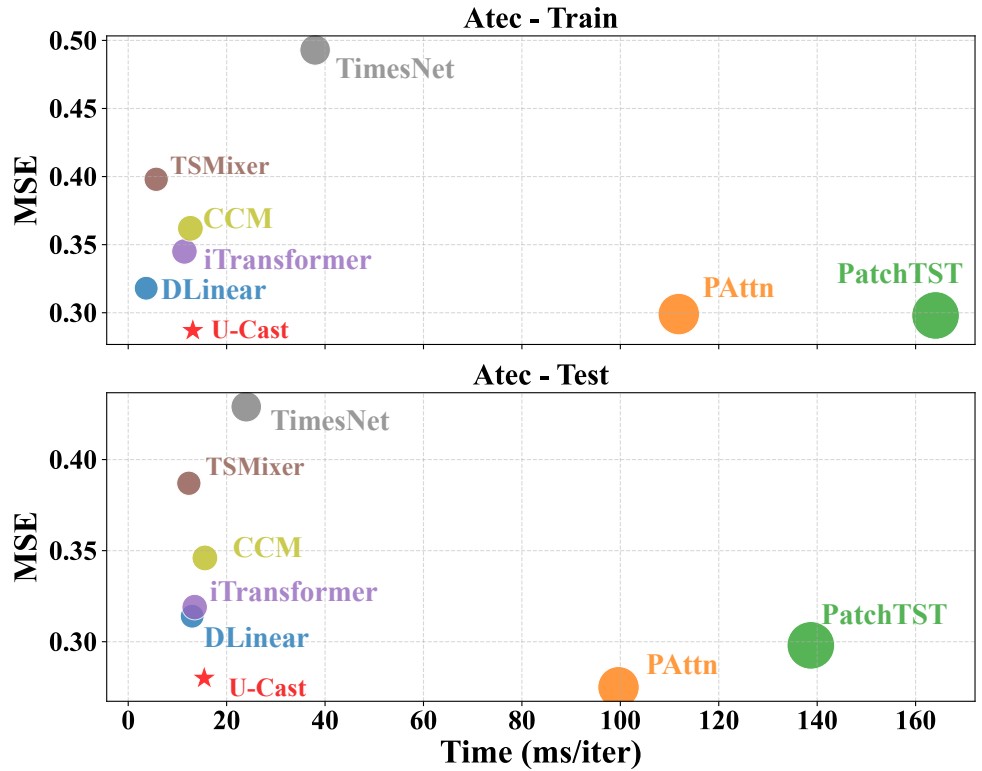

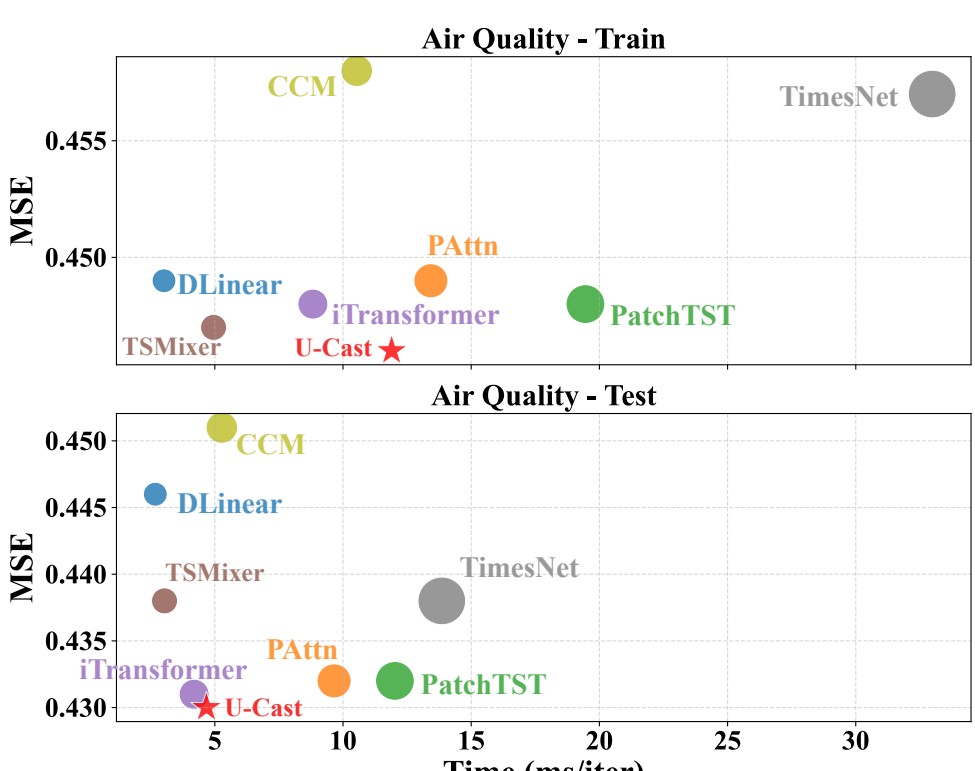

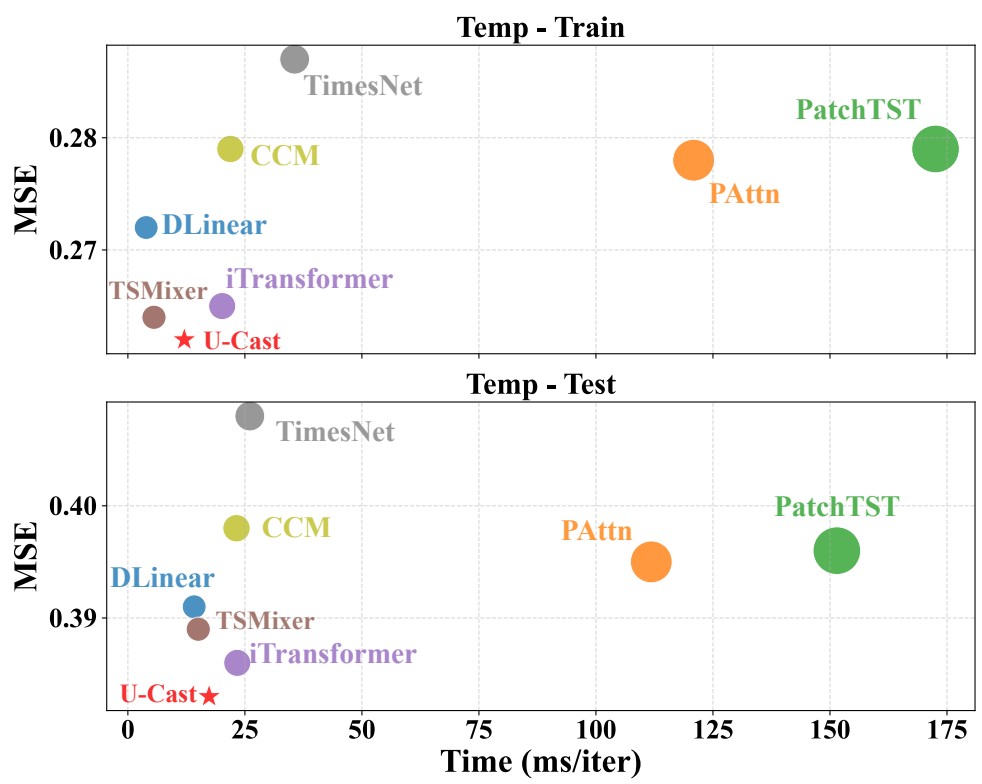

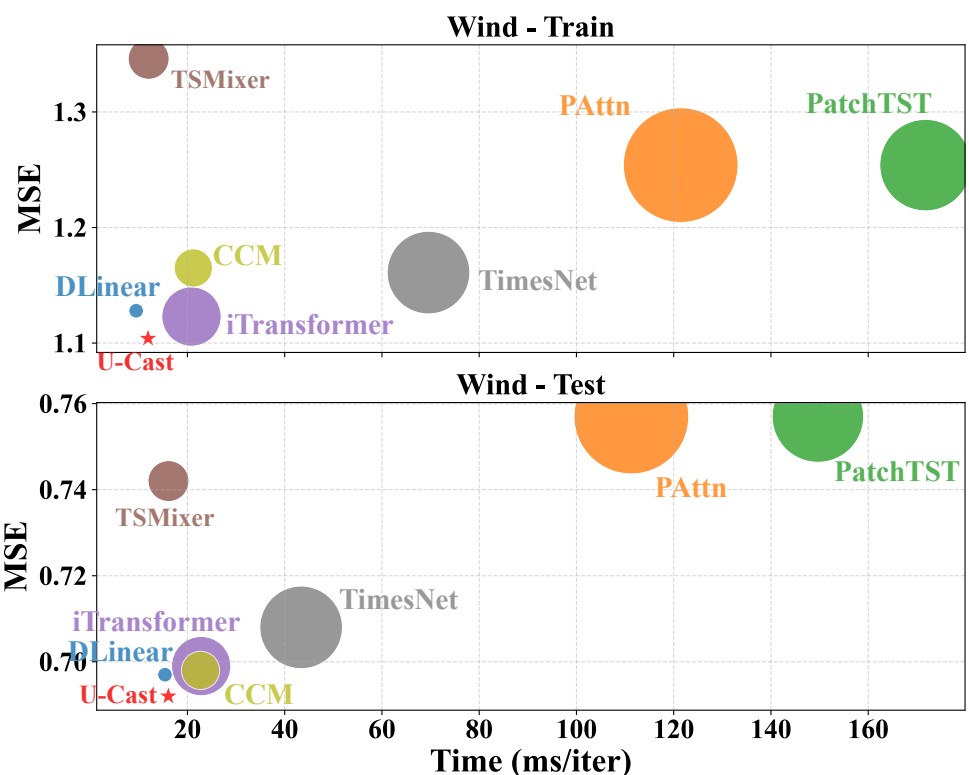

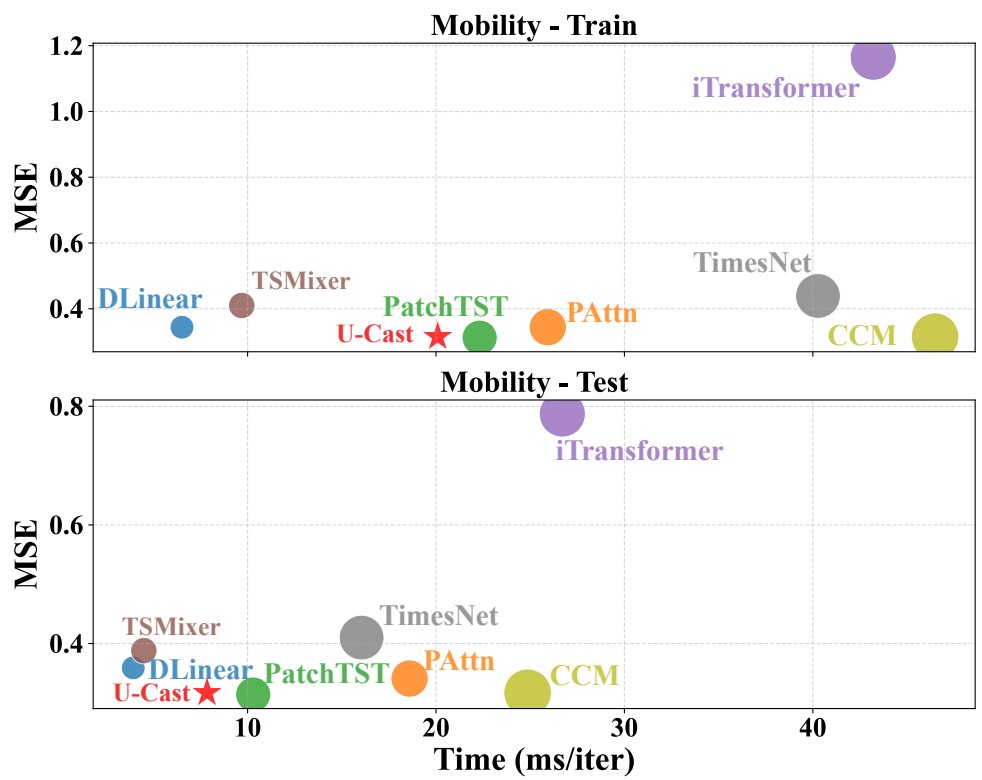

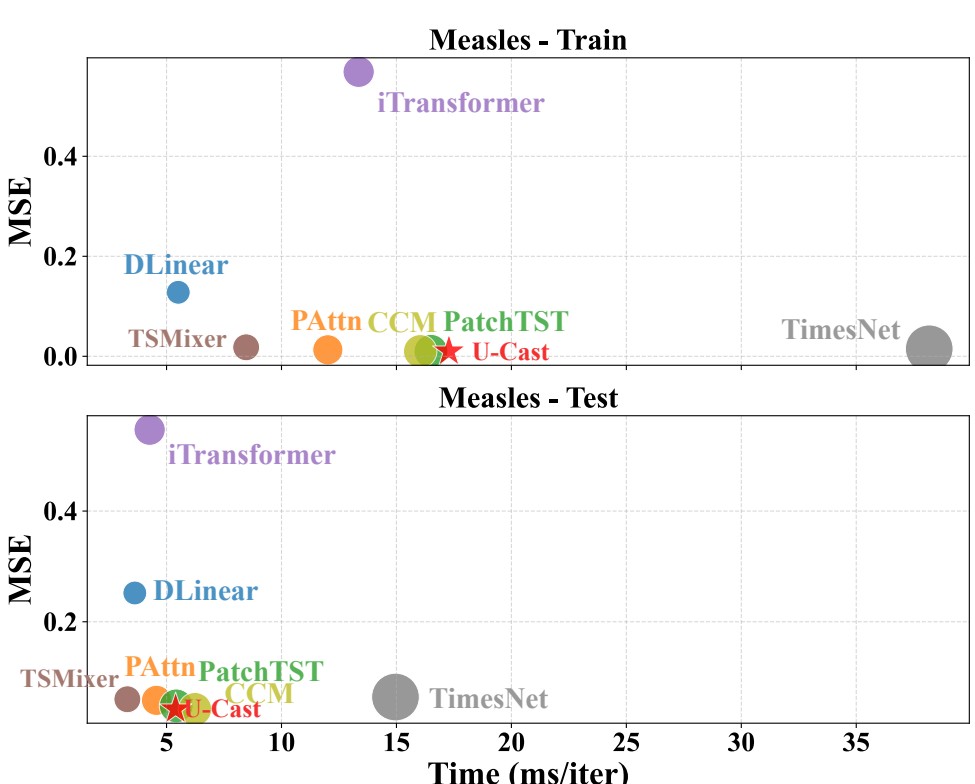

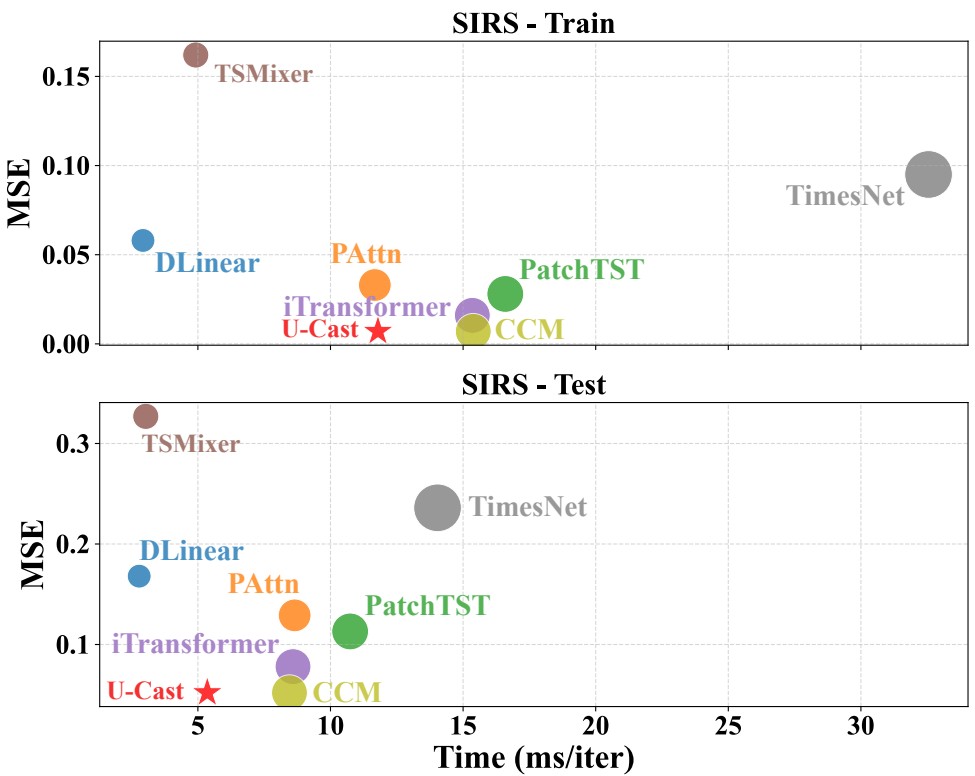

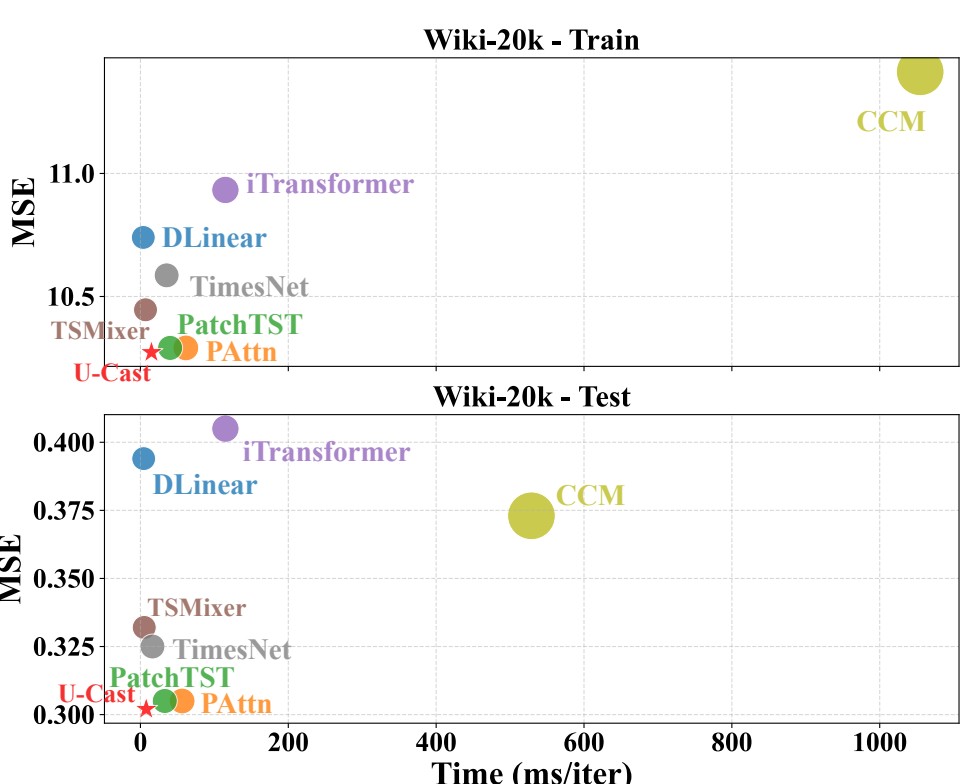

# Z SHOWCASES

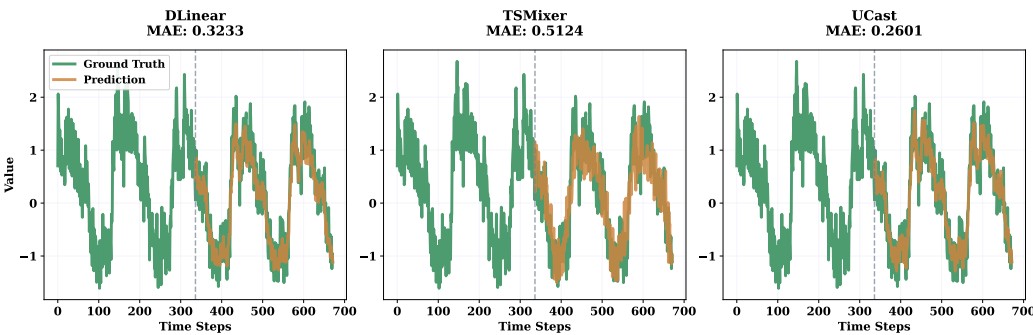

Figure 11: Prediction cases from Atec by different models under the input-S-predict-S settings.

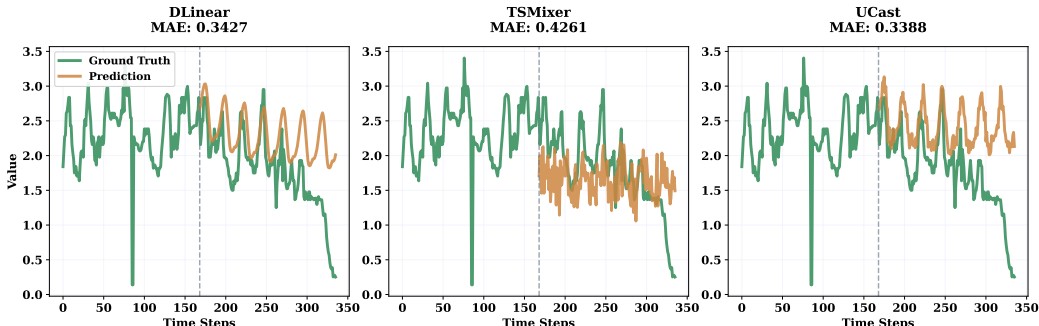

Figure 12: Prediction cases from Temp by different models under the input-S-predict-S settings.

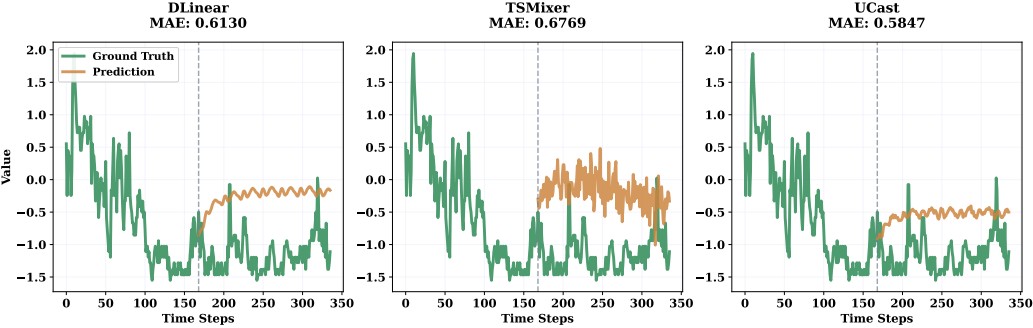

Figure 13: Prediction cases from Wind by different models under the input-S-predict-S settings.

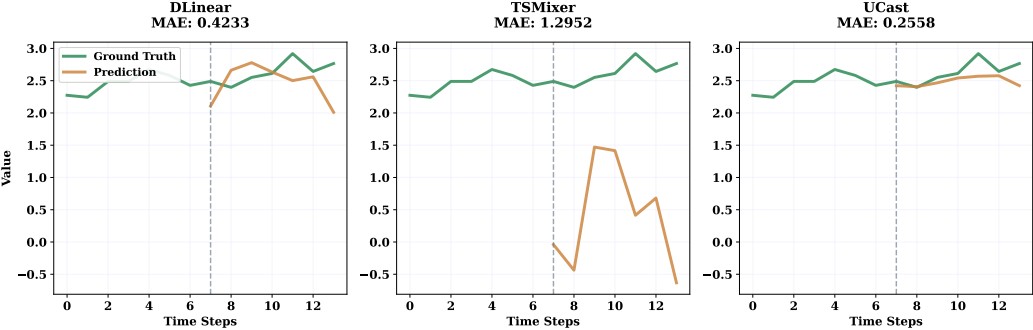

Figure 14: Prediction cases from Mobility by different models under the input-S-predict-S settings.

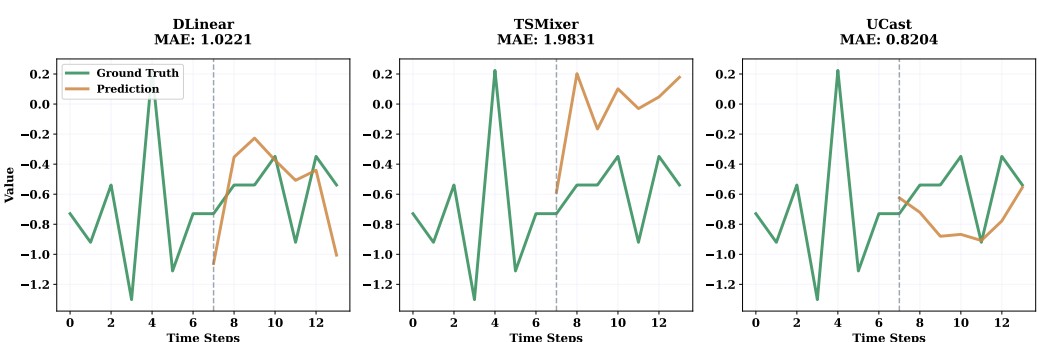

Figure 15: Prediction cases from M5 by different models under the input-S-predict-S settings.

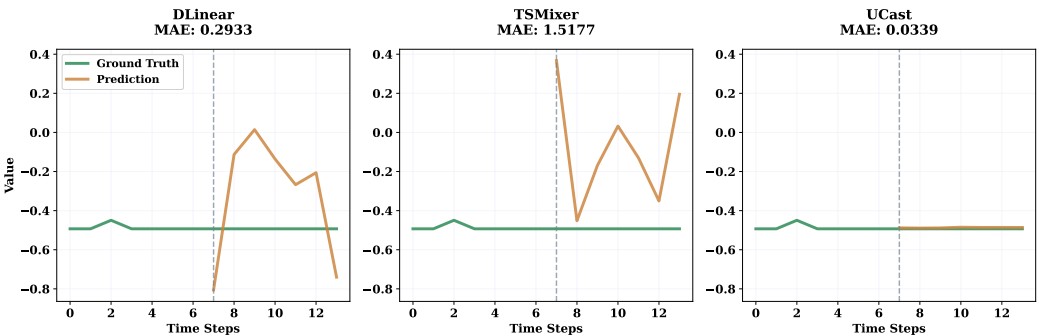

Figure 16: Prediction cases from Measles by different models under the input-S-predict-S settings.

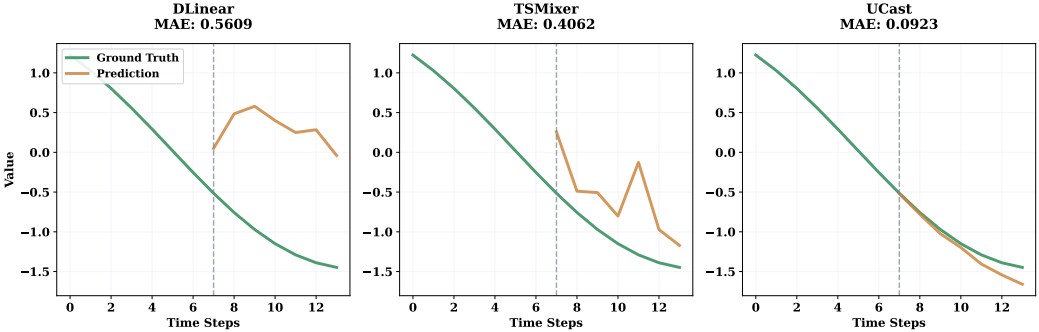

Figure 17: Prediction cases from SIRS by different models under the input-S-predict-S settings.

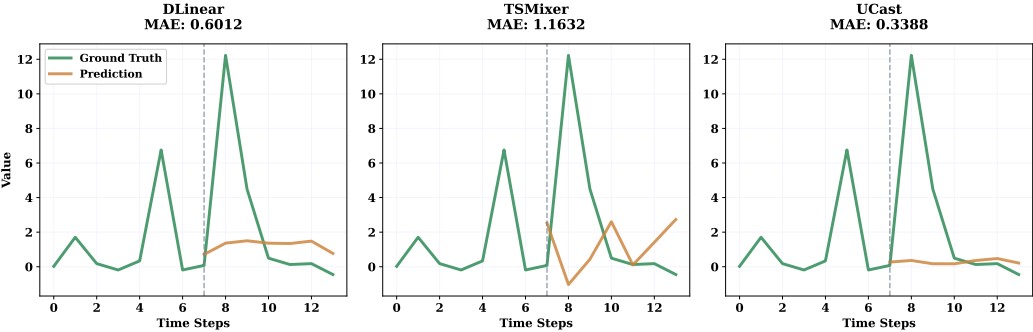

Figure 18: Prediction cases from Wiki-20k by different models under the input-S-predict-S settings.

