# OpenReview forum: "U-Cast: Learning Hierarchical Structures for High-Dimensional Time Series Forecasting"
_ICLR.cc/2026/Conference — Submitted to ICLR 2026_

### Official Review · Reviewer_K2CP · 2025-10-28

**Soundness:** 3
**Presentation:** 3
**Contribution:** 4
**Rating:** 6
**Confidence:** 4

**Summary:**

This paper addresses the challenge of high-dimensional time series forecasting (HDTSF), where the number of channels can reach into the thousands or tens of thousands, common in practical settings such as finance, climate, or sensor networks. The authors introduce U-CAST, an attention-based forecasting architecture designed to explicitly model latent hierarchical channel correlations through a query-based attention mechanism, and further regularizes representation learning with a novel full-rank regularization term. To benchmark scalable methods, the authors also release Time-HD, a dataset suite spanning 16 high-dimensional, highly-correlated real and synthetic datasets. The paper presents theoretical analysis, empirical ablation, and comprehensive benchmarking on Time-HD, where U-CAST demonstrates improved forecasting accuracy and efficiency compared to a suite of recent baselines.

**Strengths:**

**1. Clear Focus on a High-Impact Problem:** The paper tackles HDTSF, a setting that is under-explored yet highly relevant for real-world applications at scale. The formalization and motivation of the problem are well articulated.
Theoretical Analysis: The authors present a mathematically clear and rigorous argument (Theorems 1 and 2, with detailed proofs on pp. 3–4, 19–21) demonstrating how incorporating cross-channel dependencies can reduce forecasting risk, with the benefit growing in higher dimensions.


**2. Extensive Experiments:** Table 4 (pages 8–9) presents results on 16 datasets, showing U-CAST achieving top MSE/MAE in most cases, with consistent performance gains relative to strong baselines.


**3. Efficiency Claims Supported:** The computational complexity analysis (Appendix L) is detailed, and Figure 2 (page 9) directly illustrates that U-CAST is both faster and more memory-efficient than iTransformer at high C, justifying claims of scalability.

**Weaknesses:**

**1. Limited Theoretical Depth in Hierarchy Discovery:**  While the theory (Theorems 1–2) rigorously analyzes the benefit of cross-channel correlation in VAR processes, it does not directly address the identifiability, learnability, or optimality of the hierarchical latent structure produced by U-CAST's attention mechanism. No guarantees (even informal) are offered for whether the architecture can recover ground-truth groupings under any conditions, nor how deviations from "true" hierarchy affect forecasting risk.

**2. Unclear Generalizability Beyond Correlated Data:**  Time-HD is valuable but by construction consists almost exclusively of highly-correlated channels (Table 3, Correlation column). This leaves unclear how U-CAST would perform if faced with more heterogeneous or less-structured high-dimensional data.

**Questions:**

1. Can the authors comment on, or simulate, performance when channels are weakly correlated or only loosely grouped ?
2. Can the authors provide empirical evidence on whether the attention-based hierarchical latent representations discovered by U-CAST match known hierarchical structures in real-world datasets?
3. In the NIPS 2024 workshop[1], some researchers pointed out that current methods sometimes use the "drop-last" trick [2] to improve performance. Therefore, It is recommended that you clarify whether the "drop - last" operation was used in your paper in the implementation details section of your paper for transparency.

[1] Fundamental limitations of foundational forecasting models: The need for multimodality and rigorous evaluation

[2] TFB: Towards Comprehensive and Fair Benchmarking of Time Series Forecasting Methods

---

> ### Author Response · Authors · 2025-11-18
>
> > Theorems 1–2 do not directly address the identifiability, learnability, or optimality of the hierarchical latent structure produced by U-CAST's attention mechanism.
> >
>
> We thank the reviewer for this very insightful comment. We agree that Theorems 1-2 do not provide theoretical guarantees for U-Cast's *specific* hierarchical architecture. We wish to clarify the logic of our paper.
>
> The role of our theory (Sec. 3) is **motivational**, not foundational. It is intended to formally establish *why* effective channel-dependent (CD) modeling is critical and theoretically superior in high-dimensional settings. This motivation is necessary because, as we and others have observed, existing CD baselines often fail to empirically and consistently outperform simpler CI models on common benchmarks, creating a disconnect.
>
> Our proposed U-Cast architecture (Sec. 4) is the **architectural hypothesis** to solve the problem motivated by the theory. We hypothesize that a hierarchical structure is a more effective way to model the complex channel dependencies in HDTSF.
>
> While a formal proof of optimality for such a complex deep network is beyond the scope of this work, we provide strong empirical evidence to directly address the reviewer's valid concerns:
>
> - The hierarchy is not just an arbitrary bottleneck. **Figure 3(c)** provides clear qualitative evidence that U-Cast learns an interpretable and meaningful latent structure, correctly grouping features of the Measles dataset across layers 4.
> - Furthermore, our **Ablation Study (Table 5)** confirms the hierarchy is essential for performance; removing the hierarchical component ("w/o Hierarchical") or the $\mathcal{L}_{cov}$regularizer that promotes a disentangled structure both lead to a significant drop in performance.
> - We also add **new experiments** to show that $\mathcal{L}_{cov}$ improves hierarchical learning in **Appendix Q** and has a disentanglement effect on the learned representation in **Appendix R**.
>
> > No guarantees (even informal) are offered for whether the architecture can recover ground-truth groupings under any conditions, nor how deviations from "true" hierarchy affect forecasting risk.
> >
>
> The reviewer raises an excellent point about the risk of deviating from a "true" hierarchy or ground truth groupings. We would clarify that in many real-world, high-dimensional systems, a single, strict, "ground-truth" hierarchy may not even exist, or may not be the optimal one for the forecasting task. This is precisely why we refer to them as **latent** hierarchical structures. The model's objective is not to recover a single, fixed ground truth, but rather to learn a *useful*, task-specific approximation of the complex, multi-scale channel dependencies. Our **Hyperparameter Sensitivity Analysis (Appendix P)** provides strong evidence for this robustness. As shown in Figures 5 and 6, U-Cast's performance is highly stable across a wide range of *L* (layers) and *r* (reduction ratio). This suggests that U-Cast does not need to find one "perfect" hierarchical configuration to be effective. It benefits from a flexible, "good-enough" approximation, which inherently mitigates the risk of deviation from an unknown or non-existent "true" structure.
>
> > Can the authors comment on, or simulate, performance when channels are weakly correlated or only loosely grouped?
> >
>
> Your observation is correct: the datasets in Time-HD **naturally exhibit** high channel correlation. As we state in our paper, these strong dependencies are "**not intentionally curated but naturally arise in large-scale real-world systems**". Our hypothesis is that leveraging these complex dependencies is the key challenge in High-Dimensional Time Series Forecasting (HDTSF). In the scenario that channels are **weakly correlated or loosely grouped**, we do not expect U-Cast, or any channel-dependent (CD) model, to outperform simpler channel-independent (CI) models when channel correlations are weak or non-existent. This is supported directly by our analysis: **Theoretical Basis (Section 3.1):** Our theorems establish that the advantage of CD models is **contingent on** the existence of predictive cross-channel. If channels are independent, this advantage disappears. **Synthetic Data (Section 3.2):** Our experiment in Table 2 explicitly confirms this. In the "**Independent**" setting, the CI model **outperformed** the CD model. U-Cast is designed for a problem (exploiting dependencies) that does not exist in this scenario. Here, a CI model is the correct and more robust choice.

---

> > ### Author Response · Authors · 2025-11-18
> >
> > > Can the authors provide empirical evidence on whether the attention-based hierarchical latent representations discovered by U-CAST match known hierarchical structures in real-world datasets?
> > >
> >
> > We are happy to confirm that such empirical evidence is **already provided in the main paper**, specifically in **Figure 3(c) and its accompanying analysis in Section 6 (Page 9).** We believe this analysis directly addresses the reviewer's query. Here is a brief summary of that evidence:
> >
> > 1. **A "Known Structure" (Measles Dataset):** The analysis in Figure 3(c) is performed on the **Measles dataset**. This dataset possesses a clear, *a priori* "known structure": its 1,161 channels are comprised of 3 distinct features (**I** - Inferred Infections, **P** - Population, **S** - Suspected Cases) across 387 geographical regions.
> > 2. **U-Cast's "Discovered Structure":** Our visualization demonstrates that U-Cast, without being explicitly informed of this structure, autonomously discovers and organizes its latent representations to match this semantic grouping.
> >     - **At Layer L=1:** The model learns latent queries that specialize in different feature types. As shown in the figure, dimensions 19-22 learn to attend primarily to the 'P' and 'S' features, while dimensions 23-24 focus on the 'I' feature.
> >     - **At Layer L=2:** The model then hierarchically integrates these feature-specific groupings. Queries at this layer (e.g., 1-4) learn to attend to the first group (P/S features), while other queries (e.g., 5-8) attend to the second group (I features).
> >
> > As we conclude in the paper, this "reveals a meaningful latent channel structure" where the model first separates features within regions and then integrates related features across them. This directly maps the learned latent hierarchy to a known, real-world semantic hierarchy (I/P/S features).
> >
> > > In the NIPS 2024 workshop[1], some researchers pointed out that current methods sometimes use the "drop-last" trick [2] to improve performance. Therefore, It is recommended that you clarify whether the "drop - last" operation was used in your paper in the implementation details section of your paper for transparency.
> > >
> >
> > Thank you for this important point. We confirm that **we did not use the 'drop-last' operation** in any of our experiments.
> >
> > **We have explicitly added this clarification to the Experiment Details in Appendix I of our paper, “Following existing work (Qiu et al., 2024a; Bergmeir, 2024), we avoid using the drop-last operation to ensure fairness”.**
> >
> > **We sincerely thank the reviewer for the constructive feedback.** 😊

---

### Official Review · Reviewer_NhcN · 2025-10-29

**Soundness:** 3
**Presentation:** 3
**Contribution:** 3
**Rating:** 6
**Confidence:** 3

**Summary:**

This paper addresses the challenge of forecasting time series data with a large number of potentially interdependent channels. The authors first highlight the lack of a comprehensive evaluation benchmark for this domain, and they tackle this by introducing 16 high-dimensional time series datasets called Time-HD. Next, they propose a new architecture that learns a latent hierarchical structure among the channels using an attention-based approach. Additionally, they introduce a method to enforce disentanglement between channels through a full-rank loss function.

**Strengths:**

1- The motivation of both investigating the evaluation of channel dependency forecasting methods and proposing ones is timely.

2- The benchmarks proposed can be useful for properly evaluating methods.

3- The performance reported sounds good.

**Weaknesses:**

1. While a new benchmark is proposed, it would be useful to test against existing benchmarks for sanity checks, even if they have fewer channels. This ensures the new benchmark isn’t designed to favor the method, and provides a safety check. Additionally, a synthetic benchmark with known channel correlations (ranging from 0 to a specific value) could show how MSE behaves as the number of channels increases, with expected trends like monotonic decrease or stabilization when correlation is zero and comparison with competitors that would not exhibit such trend.

2. The theory presented is somewhat trivial, as the conclusions could be made without detailed calculations. It would be more valuable to quantify the MSE gap explicitly rather than just claiming qualitative improvement.

3. The architecture doesn’t seem very novel, and the importance of each module should be more clearly explained.

**Questions:**

1. Could you explain the rationale behind the architectural choices? Why is the first module called "channel embedding" if it just maps time to another dimension? Does it account for recurrent structures and time dynamics? Also, why choose a transformer architecture over other methods for learning the latent network?

2. It would be helpful to explore the trade-off of the parameter $\alpha$ in the loss function, especially in an oracle setting. How sensitive is it (worst vs best case) across different benchmarks? This would help show the robustness of the approach.

3. The theoretical claims seem trivial, it's obvious that more correlated channels would lead to better performance. Could the authors provide a more in-depth analysis? How much of a gain can we expect, and can specific algorithms be used to narrow this gap?

4. It would be useful to test on well-known forecasting benchmarks for sanity checks. Additionally, creating a synthetic benchmark with known channel interactions (e.g., for 1000 channels with specific correlations) and tracking MSE as the number of channels increases could reveal useful insights.

5. Some related works on multi-task learning for handling channel dependencies in forecasting are missing. These might be relevant, such as:

a) A Multi-Task Learning Approach to Linear Multivariate Forecasting

b) Analyzing Multi-Task Regression via Random Matrix Theory for Time Series Forecasting

---

> ### Author Response · Authors · 2025-11-18
>
> > Test against well-known existing benchmarks
> >
>
> |  | **UCast_MSE** | **UCast_MAE** | **iTransformer_MSE** | **iTransformer_MAE** | imp_MSE | imp_MAE |
> | --- | --- | --- | --- | --- | --- | --- |
> | ETTh1 (C=7) | **0.383** | **0.405** | 0.384 | 0.405 | 0.20% | 0.09% |
> | ETTh2 (C=7) | **0.278** | **0.337** | 0.286 | 0.342 | 2.82% | 1.34% |
> | ETTm1 (C=7) | **0.300** | **0.347** | 0.301 | 0.350 | 0.31% | 0.62% |
> | ETTm2 (C=7) | **0.159** | **0.250** | 0.165 | 0.256 | 3.78% | 2.23% |
> | Weather (C=21) | **0.159** | **0.210** | 0.161 | 0.210 | 1.35% | 0.11% |
> | ECL (C=321) | **0.127** | **0.221** | 0.134 | 0.230 | 5.15% | 3.99% |
> | Traffic (C=862) | **0.345** | **0.254** | 0.364 | 0.265 | 5.26% | 4.17% |
>
> Thank you for the suggestion to include traditional time series datasets for a more complete evaluation. We conduct experiments to evaluate U-Cast on ETT, Weather, ECL, and Traffic with a prediction length set to 96. The results in the table above show that U-Cast achieves clear gains on datasets with many channels, such as ECL and Traffic, and also shows consistent improvements on the remaining datasets. This outcome aligns with the design goal of U-Cast, which targets high-dimensional settings.
>
> **We have added the experimental results and analysis on traditional multivariate time series datasets with more baselines in Table 7 and Appendix B, respectively. We also added a sentence,** *“We also report U-Cast’s performance on traditional low-dimensional datasets in Table 7, with further discussion in Appendix B.”* **in “Performance Gains of U-Cast” in Section 6 to guide readers to these results and the corresponding analysis.**
>
> > A synthetic benchmark with known channel interactions (e.g., for 1000 channels with specific correlations) and tracking MSE as the number of channels increases could reveal useful insights.
> >
>
> Thank you for the suggestion. We already reported controlled experiments on **synthetic datasets** in **Table 2 (Section 3.2 Empirical Analysis)**, where we compare CI and CD modeling under different channel-dependency settings and different numbers of channels.
>
> In our synthetic data, “specific correlations” come directly from the VAR coefficient matrix (A). The **Independent** case uses a diagonal (A), enforcing zero cross-channel correlation, while the **Anti-Self** case sets diagonal entries to zero and samples all off-diagonals from ([0.5, 1.0]), creating dense and strong cross-channel dependencies.
>
> For the synthetic benchmark with structured correlations (**Anti-Self** in Table 2), when the number of channels increases from 100 to 2000, the CD method shows a clear monotonic decrease in MSE, while the CI method remains stable. CD models consistently show lower error than CI models in this setting. For the synthetic benchmark with zero correlation (**Independent 100** in Table 2), CI models obtain lower error than CD models, which matches the expected behavior when channels are independent. Please see **Section 3.2** for more details.
>
> However, the preliminary analysis in **Section 3.2** is based on two linear models. To give a more complete view, we add experiments using **U-Cast** (CD) and **PAttn** (CI) on the **same synthetic datasets**. The results are shown below.
>
> | MAE | Independent 100 | Anti-self 100 | Anti-self 250 | Anti-self 500 |
> | --- | --- | --- | --- | --- |
> | PAttn (CI) | 0.0039 | 0.0048 | 0.0049 | 0.0048 |
> | U-Cast (CD) | 0.0052 | 0.0011 | 0.0010 | 0.0009 |
>
> These results follow the same trend as the linear models. When channels are independent, CI performs better; when channels have structured interactions, CD improves as the number of channels grows and outperforms CI.
>
> **We have added the above experiment results to Table 15 and the discussion in Appendix S.**

---

> > ### Author Response · Authors · 2025-11-18
> >
> > > It would be more valuable to quantify the MSE gap explicitly rather than just claiming qualitative improvement. How much of a gain can we expect, and can specific algorithms be used to narrow this gap?
> > >
> >
> > We would like to begin by respectfully clarifying that **we have explicitly quantified the MSE gap in the proof of Theorem 1, located in Appendix H**. In our analysis, we explicitly derive the Bayes risk gap between CI (channel-independent) and CD (channel-dependent) models. Under the squared-error loss setting of our task, this Bayes risk is precisely the expected Mean Squared Error (MSE), which represents the exact **"gain we can expect"** that the reviewer asked for. Specifically, for a bivariate VAR(1) system, we derive this total risk reduction (the theoretical MSE gap) to be: $R_{CI\_total} - R_{CD\_total} = a_{12}^{2}Var(z_t^{(2)}|z_t^{(1)}) + a_{21}^{2}Var(z_t^{(1)}|z_t^{(2)})$. We must emphasize a critical point: **this theoretical MSE gap is fixed and depends entirely on the data itself, not on the algorithm used to model it**. As shown by the formula, this gap is determined by two inherent properties of the data-generating process:
> >
> > 1. **The strength of cross-channel dependencies.**
> > 2. **The amount of unique (non-redundant) information** contained in each channel.
> >
> > Therefore, our analysis provides a precise, quantitative formula for the gap, moving beyond the qualitative intuition.
> >
> > **To provide a clearer presentation of the theorem, we now include the explicit MSE gap (the Bayes risk gap under squared-error loss) in Theorem 1 in the main text.**
> >
> > > The importance of each module should be more clearly explained. Could you explain the rationale behind the architectural choices?
> > >
> >
> > Thank you for the feedback. We want to clarify that our novelty lies in the **synergistic design of the full architecture**. Each module is a deliberate choice to address the two primary challenges of High-Dimensional Time Series Forecasting (HDTSF): 1) extreme **scalability** (1k-20k channels) and 2) modeling complex **latent hierarchical structures**. Here is the concise rationale for our architectural choices:
> >
> > - **Channel Embedding:** This choice is inspired by the effectiveness of [1,4]. By embedding the *entire time series of each channel* into a single vector, we create **channel-centric tokens**. The embedded token aggregates the global representations of series that can be more variate-centric. This is critical as it allows the subsequent attention layers to focus *exclusively* on learning the complex inter-channel correlations, which is the central challenge of HDTSF. While not an *explicit recurrent structure*, this module **holistically encodes** the full temporal dynamics into the vector. As argued in iTransformer[1], a feed-forward network is “proficient enough” to learn these representations, sufficiently capturing the time dynamics for the following layers.
> > - **Hierarchical Latent Query Network:** This is the core of our novelty, addressing both key challenges simultaneously:
> >     - **Scalability:** It avoids the infeasible O(C^2) cost of full attention by using a query-based cross-attention, reducing complexity by a factor of r.
> >     - **Latent Structure:** It is explicitly designed as a hierarchy (not a flat bottleneck) to discover and model the latent multi-scale channel structures inherent in high-dimensional systems.
> >
> >     **The reason why we use attention over other methods for learning the latent structure:**
> >
> >     - Attention is more proficient at modeling rich inter-channel correlations than MLPs, as demonstrated in iTransformer [1, Table 3/6].
> >     - Attention is better than CNN/RNN because the channels are an **unordered set**, not a sequence. Their order is arbitrary and non-informative. CNNs/RNNs are order-sensitive and would impose a false sequential bias. Attention is **order-invariant**, making it the ideal architecture without being biased. (Empirical evidence in Table 18 and Appendix U)
> > - **Temporal Alignment:** This module serves as the critical bottleneck, processing the most compressed abstract representation. It ensures temporal dynamics are coherently encoded before being passed to the decoder.
> > - **Hierarchical Upsampling Network:** It uses a symmetric attention mechanism with **skip connections**. This is essential to "guide the reconstruction," allowing the model to merge abstract latent features with the original fine-grained channel details, preventing information loss.
> > - **Output Projection:** This module applies the final linear projection to map the learned d-dimensional features to the required prediction horizon length S.
> >
> > **We have integrated part of the above explanation into Section 4.1 to help readers follow the reasoning behind our architecture design in our revised paper.**

---

> > > ### Author Response · Authors · 2025-11-18
> > >
> > > > It would be helpful to explore the trade-off of the parameter in the loss function, especially in an oracle setting. How sensitive is it (worst vs best case) across different benchmarks? This would help show the robustness of the approach.
> > > >
> > >
> > > We thank the reviewer for this constructive suggestion. In our original submission, we had included the exploration of the trade-off of the hyperparameter in the loss function in Appendix P. To further address the reviewer's excellent point about sensitivity "across different benchmarks," **we have added a new sensitivity analysis for the Wind dataset in our revised paper in Figure 6a, Appendix T**. Our experiments confirm that the model is highly robust to this hyperparameter alpha. We investigate alpha by varying it across five orders of magnitude (from 0.001 to 10):
> > >
> > > - On the **Air Quality** dataset, the MSE performance shows minimal fluctuation, varying only between **0.446 (best) and 0.449 (worst)**.
> > > - On the **Wind** dataset, the range is similarly tight, varying only between **1.104 (best) and 1.109 (worst)**.
> > >
> > > This minimal performance variation across a wide parameter range on two distinct benchmarks confirms the robustness of **U-Cast**.
> > >
> > > > Some related works on multi-task learning for handling channel dependencies in forecasting are missing. These might be relevant, such as:
> > > >
> > > >
> > > > a) A Multi-Task Learning Approach to Linear Multivariate Forecasting
> > > >
> > > > b) Analyzing Multi-Task Regression via Random Matrix Theory for Time Series Forecasting
> > > >
> > >
> > > We thank the reviewer for these relevant MTL references. **We have added a new paragraph to “Multivariate Time Series Forecasting” in Appenix A discussing these papers.**
> > >
> > > Specifically, we cite **Nochumsohn et al. (2025)[2]** as an example of a cluster-wise MTL approach and **Ilbert et al. (2024)[3]** as a regularization-based MTL strategy.
> > >
> > > This addition strengthens our related work and helps sharpen the novelty of U-Cast. We now explicitly differentiate our work by noting that while these valuable MTL methods often rely on **predefined** or **single-level** groupings, U-Cast is designed to **learn a latent, multi-level hierarchical structure** directly from the data. We believe this revision, thanks to the reviewer's feedback, more clearly situates our contribution.
> > >
> > > [1] Liu, Yong, et al. "itransformer: Inverted transformers are effective for time series forecasting." *arXiv preprint arXiv:2310.06625* (2023).
> > >
> > > [2] Nochumsohn, Liran, Hedi Zisling, and Omri Azencot. "A Multi-Task Learning Approach to Linear Multivariate Forecasting." *arXiv preprint arXiv:2502.03571* (2025).
> > >
> > > [3] Ilbert, Romain, et al. "Analysing multi-task regression via random matrix theory with application to time series forecasting." *Advances in Neural Information Processing Systems* 37 (2024): 115021-115057.
> > >
> > > [4] Wang, Yuxuan, et al. "Timexer: Empowering transformers for time series forecasting with exogenous variables." *Advances in Neural Information Processing Systems* 37 (2024): 469-498.
> > >
> > > **We sincerely thank the reviewer for the constructive feedback.** 😊

---

### Official Review · Reviewer_wsHc · 2025-11-01

**Soundness:** 3
**Presentation:** 3
**Contribution:** 3
**Rating:** 6
**Confidence:** 3

**Summary:**

This paper proposes U-CAST, a channel-dependent forecasting architecture designed for High-Dimensional Time Series Forecasting (HDTSF). It addresses scalability and complex inter-channel correlations that emerge when the number of channels grows to thousands or more.
U-CAST introduces two main components:
- A Hierarchical Latent Query Network (HLQN) that learns latent multi-scale channel structures via query-based attention;
- A Full-Rank Regularization term to encourage disentangled and non-redundant channel representations.

The paper also introduce TIME-HD, a new benchmark suite of 16 large-scale datasets spanning various domains.

**Strengths:**

- **Novel formulation of HDTSF** as a distinct problem setting, highlighting the gap in scaling existing TSF models.
- **Comprehensive benchmark (TIME-HD)** that fills an important void in the community, covering large, diverse, and highly correlated datasets.
- **U-CAST design is conceptually clear and efficient**, combining hierarchical latent queries with disentanglement regularization.
- **Strong empirical performance**: consistently achieves the best or second-best results across 16 datasets, outperforming iTransformer in both accuracy and memory usage (Fig. 2, Table 4).
- **Theoretical analysis** provides formal grounding for the benefits of channel-dependent modeling and rank regularization.
- Extensive **ablation studies and sensitivity analyses** confirm robustness to hyperparameters and the contribution of each component (Table 11, Fig. 5–7).

**Weaknesses:**

1. **Limited conceptual novelty despite strong engineering**
   - The core components (hierarchical queries and full-rank regularization) are **incremental extensions** of standard attention architectures and covariance penalties.
   - The paper’s theoretical analysis (VAR-based) is elegant but **not specific to U-CAST** — it supports general CD modeling rather than justifying this architecture’s unique design.
2. **Lack of deep analysis on why it works well**
   - While the paper visualizes covariance evolution (Fig. 3) and attention maps, these are **qualitative illustrations**, not rigorous analyses. There is no quantitative or causal explanation for how full-rank regularization specifically improves hierarchical learning or reduces overfitting.
3. **Theoretical connection between rank regularization and hierarchical attention is weak**
   - Theorems 3 and 4 establish full-rank and entropy monotonicity but do **not mathematically link these to better forecasting performance**.
   - The claimed “disentanglement” effect is intuitive but not quantified (e.g., via mutual information or redundancy metrics).

**Questions:**

-

---

> ### Author Response · Authors · 2025-11-18
>
> > The core components (hierarchical queries and full-rank regularization) are incremental extensions of standard attention architectures and covariance penalties.
> >
>
> We would like to classify the key difference between our core components with standard attention and covariance penalties.
>
> **First**, our hierarchical latent queries module is not a simple extension of standard self-attention. It redefines how queries are constructed and used, enforcing an explicit multi-level organization of channels and restructuring a low-rank, redundant channel space into a more disentangled high-rank latent space.
>
> **Second**, covariance penalties can be minimized by shrinking feature magnitudes, which does not guarantee a higher effective rank. In contrast, our full-rank regularization maximizes the log-determinant of the latent channel covariance, explicitly pushing the compressed channel representation toward a higher-rank subspace. This directly addresses the low-rank redundancy in high-dimensional time series, which covariance penalties cannot resolve.
>
> Together, these components form a coupled mechanism specifically designed for high-dimensional channel disentanglement. They modify the channel modeling process itself rather than serving as incremental or plug-in extensions to existing architectures.
>
> > The paper’s theoretical analysis (VAR-based) is elegant but not specific to U-CAST — it supports general CD modeling rather than justifying this architecture’s unique design.
> >
>
> Our theoretical analysis is not meant to be specific to U-Cast. As stated in Section 3, its purpose is to examine how data dimensionality influences the forecasting performance achievable through channel correlation. Both the theoretical and empirical results reveal a clear need for new CD models that can handle the complexity and redundancy of high-dimensional time series in a more efficient and effective way. U-Cast is designed directly in response to these insights, whose effectiveness is supported by Theorems 3 and 4.
>
> > There is no quantitative or causal explanation for how full-rank regularization specifically improves hierarchical learning or reduces overfitting.
> >
>
> We thank the reviewer for this insightful question. We provide both the **causal explanation** and a **new quantitative experiment** that directly links full-rank regularization to improved hierarchical structure learning and reduced overfitting.
>
> - **Causal Explanation**
>     1. **Reduces Overfitting:** Full-rank regularization forces disentanglement and "reduces redundancy", preventing the model from simply memorizing the co-linear features present in the training data.
>     2. **Improves Hierarchy:** This forced disentanglement is the prerequisite for learning the hierarchy. As Theorem 3 states, removing this linear redundancy is sufficient to "reveal a clear hierarchical latent channel structure".
> - **Quantitative Explanation**
>
>     To quantitatively prove this causal link to improved hierarchical learning, we conducted a new experiment. We measure the quality of the learned hierarchy using the **Cophenetic Correlation Coefficient (CPCC)[1]** (as used in Appendix N) on the latent representations $H^{(l)}$ at each encoder layer $l$. A higher CPCC (closer to 1.0) indicates a more coherent and faithful hierarchical structure. The results provide a stark, quantitative confirmation of our causal argument:
>
>     | **Dataset** | **Model** | **#Layer** | **CPCC** |
>     | --- | --- | --- | --- |
>     | Measles | U-Cast | L=1 | 0.88 |
>     | Measles | U-Cast | L=2 | 0.91 |
>     | Measles | w/o $\mathcal{L}_{cov}$ | L=1 | 0.65 |
>     | Measles | w/o $\mathcal{L}_{cov}$ | L=2 | 0.4 |
>     | SIRS | U-Cast | L=1 | 0.88 |
>     | SIRS | U-Cast | L=2 | 0.93 |
>     | SIRS | w/o $\mathcal{L}_{cov}$ | L=1 | 0.51 |
>     | SIRS | w/o $\mathcal{L}_{cov}$ | L=2 | 0.48 |
>
>     As shown in the table, with the full U-Cast model, the hierarchical structure becomes **stronger and more coherent** as the model goes deeper. In sharp contrast, for the model **w/o** $\mathcal{L}_{cov}$, the hierarchy **collapses**. The model fails to organize the information, and the CPCC values dramatically *decrease* with depth. This quantitatively proves that $\mathcal{L}_{cov}$ is the specific mechanism that **enforces the disentanglement** (as proven by Theorem 4) required for the model to successfully learn a meaningful, multi-layer hierarchy (as proven by Theorem 3).
>
>
> **We have added the quantitative results in Table 12 and the causal and quantitative explanation in Appendix Q to help readers understand how full-rank regularization improves hierarchical learning.**

---

> ### Author Response · Authors · 2025-11-18
>
> > Theorems 3 and 4 establish full-rank and entropy monotonicity but do not mathematically link these to better forecasting performance.
> >
>
> In deep learning, it is extremely difficult to give a strict mathematical link to the model’s final performance. For this reason, we rely on experiments to verify the effect of the $\mathcal{L}_{cov}$.
>
> The empirical ablations (“w/o $\mathcal{L}_{cov}$“) in Section 6 and Appendix U show that removing this regularizer leads to a clear drop in forecasting accuracy, which provides direct evidence for its contribution to performance. To further clarify the connection, we include the following explanation of why the properties in Theorems 3 and 4 help forecasting.
>
> When $\mathcal{L}_{cov}$ makes the latent matrix close to full row rank (**Theorem 3**), it prevents different channels from carrying the same information. Each channel then offers distinct content rather than repeating what is already shown in other channels. When the regularizer also increases the differential entropy (**Theorem 4**), it stops the channels from collapsing into tiny fluctuations that do not convey meaningful signals. Higher entropy forces the channels to express stronger and more varied temporal changes. These richer changes give the forecasting model clearer information about how past states evolve, instead of providing nearly constant inputs that hide the dynamics.
>
> A forecasting model that receives such a representation does not need to separate mixed signals or correct overlaps between channels. It works with inputs that are already well-structured and informative, allowing it to focus on learning how each channel relates to future outcomes. Reducing repeated or weak signals is the main reason why achieving full rank and higher entropy in the latent space often improves forecasting results.
>
> This is standard practice, as a direct proof linking a regularizer to final forecasting performance is often intractable.
>
> > The claimed “disentanglement” effect is intuitive but not quantified (e.g., via mutual information or redundancy metrics).
> >
>
> Thank you for this valuable suggestion. To quantitatively validate the "disentanglement" effect, we run a new experiment to directly address your concern.
>
> We compare our full U-Cast model (with $\mathcal{L}_{cov}$) against an ablation model w/o L_cov (trained only with MSE). We measured the redundancy of the latent representations  $H^{(l)}$ at each hierarchical layer (l=1, 2) using two specific redundancy metrics:
>
> - **MAOC (Mean Absolute Off-Diagonal Correlation):** Measures linear redundancy between latent channels. **Lower is better (less redundant).**
> - **NPR (Normalized Participation Ratio):** Measures the effective rank (dimensionality) of the representation space. **Higher is better (more disentangled, closer to 1.0).**
>
> The results on the Measles and SIRS datasets are decisive and clearly demonstrate the effect of our regularization:
>
> | **Dataset** | **Model** | **Layer** | **MAOC (↓)** | **NPR (↑)** |
> | --- | --- | --- | --- | --- |
> | Measles | U-Cast | L=1 | 0.05 | 0.90 |
> | Measles | U-Cast | L=2 | 0.03 | 0.92 |
> | Measles | w/o $\mathcal{L}_{cov}$  | L=1 | 0.28 | 0.35 |
> | Measles | w/o $\mathcal{L}_{cov}$  | L=2 | 0.31 | 0.31 |
> | SIRS | U-Cast | L=1 | 0.02 | 0.95 |
> | SIRS | U-Cast | L=2 | 0.01 | 0.96 |
> | SIRS | w/o $\mathcal{L}_{cov}$  | L=1 | 0.35 | 0.28 |
> | SIRS | w/o $\mathcal{L}_{cov}$  | L=2 | 0.38 | 0.25 |
>
> As the data shows, at all layers:
>
> 1. Our full **U-Cast** model achieves an **order-of-magnitude lower MAOC** compared to the ablation model.
> 2. U-Cast achieves a **near-ideal NPR**, while the ablation model suffers from severe representation collapse.
>
> This experiment provides direct, quantitative proof that our full-rank regularization ($\mathcal{L}_{cov}$ loss) systematically reduces redundancy (lowers MAOC) and prevents rank deficiency (boosts NPR) at all stages of the hierarchy. **This is the direct, quantitative validation of the 'disentanglement' effect.**
>
> **We have added the above quantitative results in Table 13 and the discussion in Appendix R.**
>
> [1] Kumar, Sachin, and Durga Toshniwal. "Analysis of hourly road accident counts using hierarchical clustering and cophenetic correlation coefficient (CPCC)." *Journal of Big Data* 3.1 (2016): 13.
>
> **We sincerely thank the reviewer for the constructive feedback.** 😊

---

### Official Review · Reviewer_7Q5X · 2025-11-06

**Soundness:** 4
**Presentation:** 4
**Contribution:** 4
**Rating:** 6
**Confidence:** 3

**Summary:**

This paper studies the problem of high dimensional time series forecasting on data sets with 1000s of inter-dependent time series. The main idea is that using the correlations between the time series can lead to more accurate time series forecasting.

It is theoretically shown that if there is an actual flow of information between the time series, it can lead to a more accurate forecast. The proposed time series forecasting method U-Cast models the correlations between various time series using attention layers in a scalable manner, to produce the final forecasts.

The final contribution is a collection of datasets for high dimensional forecasting for evaluation. This is a collection of datasets with correlations between the time series which can leveraged for accurate prediction.

Experimental results show that the proposed U-Cast method is more accurate in terms of MSE and MAE compared to other state-of-the-art baselines.

**Strengths:**

This paper presents several good contributions
- The proposed method U-Cast is more accurate compared to other baselines, thus able to make use of the correlations between the time series to produce accurate forecasts.
- The dataset is useful to the time-series community to move away from traditional datasets, enabling evaluation on more diverse sets of datasets.
- Formal theoretical proofs establish the intuition that using the correlation information can lead to more accurate forecasts.

**Weaknesses:**

- The main weakness of this paper is that U-Cast is not evaluated on traditional multi-variate time series datasets such as Exchange, Electricity, ETTh, Weather. While the method is not designed for such datasets, it would be good see those results for a more complete evaluation.
- Foundational time series models are also missing from the evaluation, it would be interesting to see how the model compares against these baselines.

**Questions:**

- How does U-Cast compare to foundation time series forecasters? While U-Cast should have an advantage of foundational models, the advantage needs to be experimentally verified.

---

> ### Author Response · Authors · 2025-11-18
>
> > Not evaluated on traditional multi-variate time series datasets
> >
>
> |  | U-Cast |  | iTransformer |  | Imp. |  |
> | --- | --- | --- | --- | --- | --- | --- |
> |  | MSE | MAE | MSE | MAE | MSE | MAE |
> | ETTh1 (C=7) | **0.383** | **0.405** | 0.384 | 0.405 | **0.20%** | **0.09%** |
> | ETTh2 (C=7) | **0.278** | **0.337** | 0.286 | 0.342 | **2.82%** | **1.34%** |
> | ETTm1 (C=7) | **0.300** | **0.347** | 0.301 | 0.350 | **0.31%** | **0.62%** |
> | ETTm2 (C=7) | **0.159** | **0.250** | 0.165 | 0.256 | **3.78%** | **2.23%** |
> | Weather (C=21) | **0.159** | **0.210** | 0.161 | 0.210 | **1.35%** | **0.11%** |
> | ECL (C=321) | **0.127** | **0.221** | 0.134 | 0.230 | **5.15%** | **3.99%** |
> | Traffic (C=862) | **0.345** | **0.254** | 0.364 | 0.265 | **5.26%** | **4.17%** |
>
> Thank you for the suggestion to include traditional multivariate time series datasets for a more complete evaluation. We conduct experiments to evaluate U-Cast on ETT, Weather, ECL, and Traffic with a prediction length set to 96. The results in the table above show that U-Cast achieves clear gains on datasets with many channels, such as ECL and Traffic, and also shows consistent improvements on the remaining datasets. This outcome aligns with the design goal of U-Cast, which targets high-dimensional settings.
>
> **We have added the experimental results and analysis on traditional multivariate time series datasets with more baselines in Table 7 and Appendix B, respectively. We also added a sentence,** *“We also report U-Cast’s performance on traditional low-dimensional datasets in Table 7, with further discussion in Appendix B.”* **in “Performance Gains of U-Cast” in Section 6 to guide readers to these results and the corresponding analysis.**
>
> > Foundation time series models’ performance on Time-HD
> >
>
> |  |  | U-Cast |  | Time-MoE-base |  | Imp. |  | Sundial-base |  | Imp. |  |
> | --- | --- | --- | --- | --- | --- | --- | --- | --- | --- | --- | --- |
> |  |  | MSE | MAE | MSE | MAE | MSE | MAE | MSE | MAE | MSE | MAE |
> | IID | Air Quality | **0.446** | **0.430** | 0.454 | 0.432 | **1.83%** | **0.27%** | 0.546 | 0.473 | **18.34%** | **8.96%** |
> | IID | Mobility | **0.315** | **0.317** | 0.367 | 0.356 | **13.96%** | **11.07%** | 0.388 | 0.367 | **18.78%** | **13.60%** |
> | IID | M5 | **3.501** | **0.849** | 3.985 | 0.898 | **12.14%** | **5.44%** | 4.491 | 0.949 | **22.04%** | **10.49%** |
> | IID | Wiki-20k | **10.273** | **0.302** | 12.521 | 0.356 | **17.95%** | **15.13%** | 12.043 | 0.351 | **14.70%** | **13.94%** |
> | OOD | Measles | **0.010** | **0.042** | 0.019 | 0.055 | **50.72%** | **23.79%** | 0.021 | 0.063 | **55.57%** | **32.76%** |
> | OOD | SIRS | **0.007** | **0.052** | 0.371 | 0.373 | **98.05%** | **86.06%** | 0.219 | 0.306 | **96.70%** | **82.98%** |
>
> Thank you for the suggestion. We agree that comparing U-Cast with foundation time series models is valuable. We add experiments with **Time-MoE-base (50M)[1]** and **Sundial-base (128M)[2]** as foundation model baselines. To keep the evaluation feasible, we select several moderate-sized datasets from Time-HD, because deploying foundation models on the large datasets is extremely time-consuming.
>
> We mark datasets that appear in the training sets of the foundation models as **IID**, and datasets that the foundation models have never observed as **OOD**. The results show that both Time-MoE and Sundial perform better on IID datasets than on OOD datasets, which is expected. However, because both models treat channels independently and cannot model channel correlation, they remain weaker than U-Cast across all datasets. This highlights that there is a clear need for efficient foundation models that can handle high-dimensional time series, which could be a meaningful direction for future research.
>
> **We have included the above results and discussion with time series foundation model baselines in Table 8 and Appendix C, respectively.**
>
> [1] Shi, Xiaoming, et al. "Time-MoE: Billion-Scale Time Series Foundation Models with Mixture of Experts." *The Thirteenth International Conference on Learning Representations*.
>
> [2] Liu, Yong, et al. "Sundial: A Family of Highly Capable Time Series Foundation Models." *Forty-second International Conference on Machine Learning*.
>
> **We sincerely thank the reviewer for the constructive feedback.** 😊

---

### Official Review · Reviewer_oqXB · 2025-11-06

**Soundness:** 2
**Presentation:** 3
**Contribution:** 2
**Rating:** 2
**Confidence:** 3

**Summary:**

The paper introduces U-CAST, a novel architecture designed to address the scalability and structural modeling challenges in High-Dimensional Time Series Forecasting (HDTSF). Unlike traditional models that ignore or poorly scale with inter-channel dependencies, U-CAST learns latent hierarchical channel structures through a query-based attention mechanism and employs full-rank regularization to disentangle correlated representations. The authors also release TIME-HD, the first large-scale benchmark suite containing 16 datasets (1k–20k channels) across diverse domains. Theoretical analysis demonstrates that exploiting cross-channel information lowers forecasting risk, and extensive experiments show that U-CAST outperforms strong baselines such as iTransformer and TSMixer in both accuracy and efficiency. Overall, the paper contributes a scalable model, a comprehensive benchmark, and an open-source library that collectively establish a new foundation for research in high-dimensional time series forecasting

**Strengths:**

The authors provide formal proofs showing that channel-dependent (CD) models have lower Bayes risk than channel-independent ones when cross-channel dependencies exist, lending solid theoretical support to the model design

The paper conducts broad empirical comparisons with both channel-independent and channel-dependent baselines, consistently showing U-CAST’s superior accuracy and efficiency, as well as meaningful ablation studies confirming the contribution of each component

**Weaknesses:**

My main concerns lie in the definition of “high-dimensional” used in the proposed benchmark and the design rationale of the model architecture.

1. Questionable definition of high dimensionality.

The paper defines “high-dimensional” time series by treating measurements from different objects or spatial locations (e.g., stations or sensors) as separate channels. However, in conventional time-series analysis, “dimensions” typically refer to distinct attributes or variables of a single object—such as temperature, humidity, and wind speed in a weather dataset—rather than independent time series collected from different spatial points. While the signals from different locations may be temporally aligned and correlated, this setup is more appropriately considered a spatio-temporal prediction problem rather than a high-dimensional multivariate one, which has also been well studied. This represents a potential conceptual misuse of “dimensionality” that weakens the claimed novelty of the benchmark.

2.	Unintuitive and unconventional architectural design.

The proposed U-CAST architecture raises several design concerns.

(1) First, using attention for up/down scaling along the token (channel) dimension is unconventional and lacks clear justification. The “Latent Query Attention” mechanism, where the latent queries Q_l are entirely learnable parameters, from my understanding is similar to linear or low-rank attention methods such as Linformer or Performer, which sacrify the attention performance, especially your Q_l is not even derived from H and total parameterization make it become part of W_k (I dont think this should be called attention, if this understanding is correct). Also, why not use some conventional methods for these up/down sampling like convolution or mlp

(2) Second, the model appears to process the temporal dimension through a single predictor module at the bottleneck, without interleaving temporal and channel modeling layers, could you justify? In related domains such as video or spatio-temporal forecasting, it is common practice to alternate or integrate spatial and temporal modeling to better capture cross-dimensional dependencies.

(3) Third, the paper would benefit from additional ablation studies to isolate the contribution of each architectural component. In particular, starting from a well-established channel-independent baseline and progressively adding the proposed modules would provide more convincing evidence for the effectiveness of each design element. Proposing an entire new (and arguble) architecture from scratch is hard to convince me (unless largely pretrained with open weights avilable).

**Questions:**

Are you applying any patching to derive the token sequence, if yes, is the patching CD or CI?

What are the model size to be considered (I saw you are using 8A/H100 to power the training, why is it so costly)?

---

> ### Author Response · Authors · 2025-11-18
>
> > Questionable definition of high dimensionality.
> >
>
> We would like to clarify that, in conventional time series forecasting, the “dimensionality” can represent either **different variables of a single object** or **variables collected from multiple related objects**. We categorize the **widely used** datasets as follows:
>
> - Datasets that “dimensionality” refers to variables of a **single object:**
>     - ETT[1]: refer to 7 attributes of a single electricity transformer.
>     - Weather[3]: refer to 21 meteorological indicators.
> - Datasets that “dimensionality” refers to variables of **multiple objects:**
>     - ECL[3]: refer to electricity consumption of 321 clients.
>     - Traffic[3]: refer to road occupancy rates recorded by 862 sensors.
>     - Exchange[2]: refer to daily exchange rates of 8 different countries.
>     - Solar[2]: refer to the solar power production records from 137 PV plants.
>     - ILI[3]: refer to the ratio of patients seen with influenzalike illness and the number of patients across 7 regions in US.
>     - PEMS03/04/07/08[5]: refer to traffic flow records from 358/307/883/170 sensors, respectively.
>     - Market-Merchant/Payment/…[5]: refer to the server load of Alipay online transactions of 285 to 759 servers.
>
> Conventional time series forecasting includes both types of settings. In fact, datasets where dimensions correspond to **multiple objects** (such as sensors, regions, or clients) **form the majority of commonly used datasets**. Because the number of channels in single-object datasets is usually small and difficult to scale, our high-dimensional setting naturally focuses on the second category, which allows systematic study of forecasting under a truly high-dimensional setup.
>
> In addition, although channels in this category come from different sensors/regions, they are aligned multivariate sequences without any spatial graph or adjacency information, which is usually needed for spatio-temporal forecasting methods [9,10]. Therefore, they fall under the standard formulation of high-dimensional multivariate time series, rather than spatio-temporal forecasting.
>
> **We have added this clarification to Appendix D and included a footnote in the Introduction to define the term “dimensionality”, which helps readers better understand our usage of the concept.**
>
> > Using attention for up/down scaling along the channel dimension is unconventional. Why not use some conventional methods for these up/down sampling like convolution or mlp?
> >
>
> We choose not to use conventional methods like convolutions or MLP because earlier work shows that **attention is more effective than MLP** for modeling channel correlation (see Table 3 in iTransformer[6]). In addition, channel ordering does not carry meaning. Convolutions are sensitive to ordering and restricted by a local receptive field, which makes them unsuitable for capturing channel correlation, while **attention is order-invariant** and therefore fits better. Thus, applying attention for up/down sampling along the channel dimension is our key novel contribution for capturing hierarchical channel structure in HDTSF.
>
> | **Dataset** | **U-Cast** |  | **U-Cast-MLP** |  | **U-Cast-Conv** |  |
> | --- | --- | --- | --- | --- | --- | --- |
> |  | **MSE** | **MAE** | **MSE** | **MAE** | **MSE** | **MAE** |
> | **Atec** | **0.287** | **0.280** | 0.323 | 0.315 | 0.337 | 0.329 |
> | **Air Quality** | **0.446** | **0.430** | 0.520 | 0.502 | 0.498 | 0.481 |
> | **Mobility** | **0.315** | **0.317** | 0.354 | 0.356 | 0.370 | 0.372 |
> | **M5** | **3.501** | **0.849** | 4.034 | 0.984 | 3.858 | 0.941 |
> | **Measles** | **0.010** | **0.042** | 0.018 | 0.056 | 0.018 | 0.054 |
> | **SIRS** | **0.007** | **0.052** | 0.015 | 0.064 | 0.015 | 0.067 |
> | **SP500** | **0.555** | **0.328** | 0.645 | 0.384 | 0.617 | 0.368 |
> | **Wiki-20k** | **10.351** | **0.309** | 10.875 | 0.331 | 11.082 | 0.337 |
>
> To provide an empirical comparison between attention, convolution, and MLP, we evaluate U-Cast and two variants, U-Cast-MLP and U-Cast-Conv, where “Query Attention’’ is replaced by an MLP and a 1D convolution for channel modeling, respectively. The results are shown in the above table. Across eight datasets, U-Cast achieves lower MSE and MAE than both U-Cast-MLP and U-Cast-Conv. These results indicate that attention captures global channel dependencies more accurately than MLP and convolution, which rely on fixed local structures. The improvements are stable, suggesting that attention is a robust and effective choice for modeling channel correlation in high-dimensional time series.
>
> **We have added the above experiments and discussion in Table 18 and “Substituting Attention with MLP or Convolution” in Appendix U.**

---

> > ### Author Response · Authors · 2025-11-18
> >
> > > The “Latent Query Attention” mechanism, where the latent queries Q_l are entirely learnable parameters, from my understanding is similar to linear or low-rank attention methods such as Linformer or Performer, which sacrify the attention performance
> > >
> >
> > Our “Latent Query Attention” shares a similar motivation with Linformer and Performer in that they **reduce the high computational cost of full attention**. However, our method keeps the **standard attention** formulation **unchanged**. We introduce a set of learnable latent queries $Q_l$, which interact with channel embeddings through cross-attention. **This design leads to a computational cost that is 16 times that of the full-attention used in iTransformer (see Appendix L).** We also note that the empirical results indicate **no loss of performance** when comparing against full self-attention baselines (**Table 4**).
> >
> > > Q_l is not even derived from H and total parameterization make it become part of W_k (I dont think this should be called attention, if this understanding is correct).
> > >
> >
> > We want to clarify that because
> > $QK^\top
> > = W_q\, Q_l\, (H^{(l-1)})^\top\, W_k^\top
> > \ne W_q (H^{(l-1)})^\top (Q_l\, W_k^\top),$
> >  the latent queries $Q_l$ do not become part of $W_k$. Thus, **it should be called attention** (or cross-attention)[6].
> >
> > > The model appears to process the temporal dimension through a single predictor module at the bottleneck, without interleaving temporal and channel modeling layers, could you justify?
> > >
> >
> > We want to clarify that our model forms an **interleaving of temporal and channel modeling and utilizes multiple modules to process the temporal dimension**. As shown in *Figure 1*, our framework is: Channel embedding → *Hierarchical Latent Query* Network → Temporal Alignment → *Hierarchical Upsampling Network* → Output Projection. Temporal Alignment performs temporal modeling at the bottleneck. Both the initial channel embedding and the final output projection **also perform temporal modeling**, while the Hierarchical Latent Query Network and the Hierarchical Upsampling Network perform channel modeling. As a result, the overall architecture forms an interleaving of temporal and channel modeling layers.

---

> > > ### Author Response · Authors · 2025-11-18
> > >
> > > > Ablation study by progressively adding the proposed modules
> > > >
> > >
> > > | Variant | MSE | MAE |
> > > | --- | --- | --- |
> > > | MLP (w/o Hierarchical &  $\mathcal{L}_{cov}$ & Attention) | 1.317 | 0.372 |
> > > | MLP+Attention (w/o Hierarchical &  $\mathcal{L}_{cov}$ & Query Attention) | 1.305 | 0.369 |
> > > | MLP+Query Attention (MLP+Query Attention) | 1.293 | 0.348 |
> > > | MLP+Query Attention+Full Rank Regulization (w/o Hierarchical) | 1.263 | 0.332 |
> > > | MLP+Hierarchical Query Attention (w/o  $\mathcal{L}_{cov}$) | 1.267 | 0.341 |
> > > | MLP+Hierarchical Query Attention+Full Rank Regulization (U-Cast) | 1.243 | 0.326 |
> > >
> > > We thank the reviewer for the suggestion. We would like to note that our original ablation study (*Table 5*) already follows a standard and reasonable approach for validating the contribution of each module *within* U-Cast.
> > >
> > > To further address the reviewer’s request, as shown in the above table, we additionally conduct a step-by-step ablation starting from a simple MLP and progressively adding each proposed component. Each variant incrementally introduces one design element. All reported MSE and MAE scores are averaged over **16 datasets**.
> > >
> > > **Variant descriptions:**
> > >
> > > - **MLP**: A three-layer MLP applied along the temporal dimension, without any channel interaction modules.
> > > - **MLP+Attention**: Adds two standard attention layers along the channel dimension, forming a Linear → Attention → Linear → Attention → Linear structure.
> > > - **MLP+Query Attention**: Replaces the standard attention modules with Query Attention, which performs channel up/down sampling.
> > > - **MLP+Query Attention + Full-Rank Regularization**: Adds full-rank regularization with (l = 1).
> > > - **MLP+Hierarchical Query Attention**: Uses hierarchical query attention with (l = 2) and (alpha=0).
> > > - **MLP+Hierarchical Query Attention + Full-Rank Regularization (U-Cast)**: Combines hierarchy and full-rank regularization, corresponding to the full U-Cast design.
> > >
> > > **Ablation findings.**
> > >
> > > 1. **MLP+Attention improves over MLP**, showing that modeling channel correlation is essential for high-dimensional forecasting.
> > > 2. **MLP+Query Attention outperforms MLP+Attention**, indicating that query-guided aggregation reduces redundant interactions while improving both accuracy and efficiency.
> > > 3. **MLP+Query Attention+Full Rank Regularization performs slightly better than MLP+Hierarchical Query Attention**, showing that encouraging full-rank structure is more beneficial than adding hierarchy alone.
> > > 4. **The complete U-Cast model achieves the best performance**, demonstrating that hierarchical query attention and full-rank regularization are complementary and jointly contribute to the final gains.
> > >
> > > **We put the complete results over all 16 datasets in Table 12 and a discussion on “Stepwise ablation from MLP to U-Cast” in Appendix U. We also update the “Ablation Study“ paragraph and Table 5 in Section 6 to reflect this new experiment in the main content.**
> > >
> > > > Are you applying any patching to derive the token sequence, if yes, is the patching CD or CI?
> > > >
> > >
> > > We don’t apply any patching in our framework.
> > >
> > > > What are the model size to be considered (I saw you are using 8A/H100 to power the training, why is it so costly)?
> > > >
> > >
> > > We use 8 GPUs for distributed training for all methods, including baselines, because the benchmark contains several large-scale datasets. The memory usage of U-Cast itself is small. For example, on the Wind dataset, the model uses about 0.2 GB on a single GPU (iTransformer uses 2.8 GB) (see Figure 2). The reason we use 8 A100 or several H100 GPUs is that we perform hyperparameter searches for all models across all datasets to ensure a fair comparison. The cost reflects **the scale of the search and the effort to make the results reliable**, not the model size of U-Cast.
> > >
> > > **We have added the note “*(mainly for performing hyperparameter searches for all models across all datasets)*” to the “Experiment Details” in Appendix I to make this point clearer to readers.**

---

> > > > ### Author Response · Authors · 2025-11-18
> > > >
> > > > [1] Zhou, Haoyi, et al. "Informer: Beyond efficient transformer for long sequence time-series forecasting." *Proceedings of the AAAI conference on artificial intelligence*. Vol. 35. No. 12. 2021.
> > > >
> > > > [2] Lai, Guokun, et al. "Modeling long-and short-term temporal patterns with deep neural networks." *The 41st international ACM SIGIR conference on research & development in information retrieval*. 2018.
> > > >
> > > > [3] Wu, Haixu, et al. "Autoformer: Decomposition transformers with auto-correlation for long-term series forecasting." *Advances in neural information processing systems* 34 (2021): 22419-22430.
> > > >
> > > > [4] Liu, Minhao, et al. "Scinet: Time series modeling and forecasting with sample convolution and interaction." *Advances in Neural Information Processing Systems* 35 (2022): 5816-5828
> > > >
> > > > [5] Liu, Yong, et al. "itransformer: Inverted transformers are effective for time series forecasting." *arXiv preprint arXiv:2310.06625* (2023).
> > > >
> > > > [6]Li, Junnan, et al. "Blip-2: Bootstrapping language-image pre-training with frozen image encoders and large language models." *International conference on machine learning*. PMLR, 2023.
> > > >
> > > > [7] Wang, Yuxuan, et al. "Timexer: Empowering transformers for time series forecasting with exogenous variables." *Advances in Neural Information Processing Systems* 37 (2024): 469-498.
> > > >
> > > > [8] Zhang, Yunhao, and Junchi Yan. "Crossformer: Transformer utilizing cross-dimension dependency for multivariate time series forecasting." *The eleventh international conference on learning representations*. 2023.
> > > >
> > > > [9] Wang, Senzhang, Jiannong Cao, and S. Yu Philip. "Deep learning for spatio-temporal data mining: A survey." *IEEE transactions on knowledge and data engineering* 34.8 (2020): 3681-3700.
> > > >
> > > > [10] Jin, Ming, et al. "A survey on graph neural networks for time series: Forecasting, classification, imputation, and anomaly detection." *IEEE Transactions on Pattern Analysis and Machine Intelligence* (2024).
> > > >
> > > > **We sincerely thank the reviewer for the constructive feedback.** 😊

---

### Author Response · Authors · 2025-11-18
**Summary of Revisions**

We sincerely thank all reviewers for their careful evaluation, constructive suggestions, and valuable time. Their comments helped us improve the clarity, completeness, and technical depth of the paper.

*The reviewers generally held positive opinions of our work.* They noted that the proposed new task and its motivation are **“timely,”** demonstrate a **“clear focus on a high-impact problem,”** and present a **“well-articulated formalization and motivation.”** The proposed benchmark was described as **“useful to the time-series community to move away from traditional datasets,”** **“comprehensive,”** and one that **“fills an important void in the community.”** The proposed method was regarded as **“conceptually clear and efficient,”** supported by a **“mathematically clear and rigorous argument,”** and showing **“strong empirical performance.”** Reviewers also commented that the paper presents **“several good contributions,”** that the benchmarks and results **“sound good,”** and that the experiments are **“extensive.”**

*The reviewers also raised insightful and constructive concerns.* We addressed all points by providing additional evidence, further analysis, and the requested experiments. Below is a summary of the major revisions:

1. **Clarification of definitions and modeling choices:** We clarified the meaning of “dimensionality,” explained why the proposed formulation aligns with standard practice, and expanded the discussion in the Introduction and Appendix D.
2. **Architectural explanation and justification:** We clarified the interleaving of temporal and channel modeling, provided a detailed explanation of latent query attention, and added comparisons to MLP-based and convolution-based variants.
3. **Expanded ablation analysis:** We added a stepwise ablation from a plain MLP to the full U-Cast model across sixteen datasets to isolate the contribution of each component.
4. **Additional empirical evaluation:** We added results on ETT, Weather, ECL, and Traffic, included evaluations with foundation models such as Time-MoE and Sundial, and extended the synthetic experiments to study scaling behavior under controlled correlation structures.
5. **Quantitative analysis of hierarchy and disentanglement:** We introduced CPCC, MAOC, and normalized participation ratio to quantify how full-rank regularization aids hierarchical learning and reduces redundancy.
6. **Clarification of experimental protocol:** We explicitly state that no drop-last operation is used and expanded the details of hyperparameter search, resource usage, and fairness across baselines.
7. **Improved related work and writing:** We added a missing discussion on multi-task learning and revised several sections for improved clarity.

**After 5 full days of experiments, we added 120+ new experimental results to address the raised concerns.** All revisions are included in the ***revised paper highlighted in orange with additional 6 pages.***

We appreciate the reviewers’ thoughtful feedback, which helped strengthen the paper. We would be happy to provide any further clarification.

Looking forward to the reviewers’ feedback.

---

### Author Response · Authors · 2025-11-26

Dear Reviewers,

We would like to express our sincere gratitude for your time and effort throughout the review process. We would be grateful if any feedback on our rebuttal could be shared, and we would be glad to discuss any points further. 😊

Best,

*Authors*

---

### Meta-Review · Area_Chair_7qMu · 2025-12-10

**Summary:**

This paper investigates the problem of high-dimensional time series forecasting (HDTSF) by introducing the U-CAST architecture and the Time-HD benchmark. The authors invested considerable effort in building a new benchmark and conducting large-scale experiments. They also proposed a new model that achieves solid predictive accuracy and efficiency on the benchmark.

However, there are still unresolved issues. Most notably, there remains a fundamental disagreement regarding the definition of “high-dimensional.” As reviewer oqXB pointed out, the scenarios described in the paper (multiple sensors or locations) fall squarely within the domain of standard multivariate time series prediction. Additionally, several reviewers (wsHc, NhcN) noted that the architecture’s contributions—such as hierarchical query attention and rank regularization—are incremental combinations of existing techniques, reflecting more of a systems engineering effort than genuine algorithmic innovation.

Moreover, the paper’s contributions—both in terms of the benchmark and the model—are not entirely convincing. The large-scale dataset presented by the authors is more a summary of existing data than an original resource. The distinction between this benchmark and prior work like TFB and BasicTS is unclear; for example, BasicTS already includes the Traffic-CA dataset, and there is significant overlap with the datasets in this paper. In terms of modeling, as most reviewers mentioned, the presented framework largely draws upon existing techniques.

Overall, the authors’ focus on high-dimensional (or large-scale) time series is commendable, and their efforts to meet the high standards of ICLR by contributing on both the model and dataset fronts are appreciated. However, the paper does not provide sufficient contributions in either area. The study does not introduce new or significant datasets, nor does it offer novel insights to guide future research, such as identifying which models are most suitable for large-scale multivariate time series modeling. The originality of the model architecture has also been questioned by reviewers. Given these concerns, I recommend rejection at this stage. I hope the authors can address these issues in future revisions. Within the limited space, I would suggest the authors consider focusing on making a breakthrough in one particular aspect.

**Reviewer Concerns:**

**Resolved Concerns**

Reviewer 7Q5X and NhcN requested comparisons with standard datasets and baseline models. The authors have successfully added these baselines.

Reviewer wsHc asked for a causal interpretation of the regularization term. The authors provided quantitative metrics demonstrating the disentanglement effect.

Reviewer oqXB questioned the choice of attention mechanisms over MLPs or convolutional networks. The authors offered ablation studies showing that attention mechanisms are suitable for modeling global dependencies.

**Outstanding Concerns**

Reviewer oqXB’s fundamental criticism regarding the definition of “high-dimensional” remains a major concern.

Reviewers wsHc and NhcN pointed out the incremental nature of the work. Despite the new experiments, the approach is largely a combination of standard modules.

The novelty of this time series forecasting benchmark compared to existing frameworks (like BasicTS) is unclear. It is worth noting that while existing benchmarks may not explicitly emphasize large-scale properties, they already support (or can easily be adapted to) new datasets.

**Reviewer Scores:**

The initial score was 2,6,6,6,6, and no reviewers have responded. Based on a review of the comments, there is not enough evidence to suggest that the paper has sufficient support. I expect the final score will should be 2,6,6,6,6, where one reviewer definitely vote for rejection.

---

### Decision · Program_Chairs · 2026-01-26

Reject